# Robust Cross-Domain Alignment

**Anish Chakrabarty**
*Theoretical Statistics and Mathematics Unit*
*Indian Statistical Institute, Kolkata*

**Arkaprabha Basu**
*University of Surrey*

**Swagatam Das** *swagatam.das@isical.ac.in*
*Electronics and Communication Sciences Unit*
*Indian Statistical Institute, Kolkata*

**Reviewed on OpenReview:** *https://openreview.net/forum?id=0mchjaZZi4*

## Abstract

The Gromov-Wasserstein (GW) distance is an effective measure of alignment between distributions supported on distinct ambient spaces. Calculating essentially the mutual departure from isometry, it has found vast usage in domain translation and network analysis. It has long been shown to be vulnerable to contamination in the underlying measures. All efforts to introduce robustness in GW have been inspired by similar optimal transport (OT) techniques, which predominantly advocate partial mass transport or unbalancing. In contrast, the cross-domain alignment problem, being fundamentally different from OT, demands specific solutions to tackle diverse applications and contamination regimes. Deriving from robust statistics, we discuss three contextually novel techniques to robustify GW and its variants. For each method, we explore metric properties and robustness guarantees along with their co-dependencies and individual relations with the GW distance. For a comprehensive view, we empirically validate their superior resilience to contamination under real machine learning tasks against state-of-the-art methods.

## 1 Introduction

Aligning unalike objects (images, networks, point clouds, etc.) based on their geometry remains the crux of machine learning challenges such as style transfer, graph correspondence, and shape matching. The first hint of a statistical measure of discrepancy between two such distinct distributions came in the form of Gromov-Wasserstein distance (Mémoli, 2011), quickly finding continual application in data alignment (Demetci et al., 2022), clustering (Chowdhury & Needham, 2021; Gong et al., 2022), and dimensionality reduction (Clark et al., 2024). Emerging as an $L^p$-relaxation of the Gromov-Hausdorff distance, it calculates the minimal distortion between replicates from distributions $\mu$ and $\nu$, themselves defined on spaces $\mathcal{X}$ and $\mathcal{Y}$ respectively. In other words,

$$\inf_{\pi} \big\| d_X(x, x') - d_Y(y, y') \big\|_{L^p(\pi \otimes \pi)},$$

where $\pi$ denotes a coupling between distributions $(\mu, \nu)$ and the spaces are endowed with the respective metrics $d_X$ and $d_Y$, $p \geq 1$. We note that the metric space $(\mathcal{X}, d_X)$ coupled with the measure $\mu$ defines a *metric measure* (mm) space. Resembling the Kantorovich formulation in OT, it immediately inspires a Monge-like upper bound to the distance (namely, Gromov-Monge (GM)), given by

$$\inf_{\phi} \big\| d_X(x, x') - d_Y(\phi(x), \phi(x')) \big\|_{L^p(\mu \otimes \mu)},$$

where the infimum is instead over measure preserving maps $\phi : \operatorname{supp}(\mu) \to \operatorname{supp}(\nu)$. In both cases, the underlying cost, measuring the extent of departure from strong isometry, differentiates the problem from

mass transportation. It is rather the susceptibility to contamination that unites alignment and OT. The value of GW can be arbitrarily perturbed only by implanting an arbitrarily 'outlying' observation. However, the defense against such outliers in the context of alignment turns out to be much more nuanced compared to OT (see Section 4). Its diverse applications, coupled with the objective of aligning geometries, demand unique solutions in different contexts. The very formulation of GW also hints towards several avenues to search for a robust formulation. On the other hand, existing approaches to robustify GW and its progenies all draw insight from similar techniques in OT. While relaxing the optimization following *partial* (Chapel et al., 2020) or *unbalanced* (Séjourné et al., 2021) OT fosters capable solutions, it is perhaps unfounded to expect them to serve every context. For example, relieving the set of feasible couplings from meeting the marginal constraints also takes away metric properties. Moreover, despite showing that image-to-image (I2I) translation architectures such as CycleGAN (Cycle-consistent Generative Adversarial Networks) are indeed special cases of GM-like distances (Zhang et al., 2022), current literature does not provide a pathway to accurate generation under contaminated source data. As remedies, this work analyzes three principal means of robustifying the cross-domain alignment problem. This way, besides meeting diverse requirements of tasks related to alignment, we address the larger landscape of contamination. We refer the reader to Section 4 for a detailed outline of the discussion.

**Contributions:** The key takeaways of our study are as follows.

- The first method introduces penalization to large distortions while calculating GW in the spirit of Tukey and Huber. In context, it gives rise to relaxed GW distances that preserve topologies and usual metric properties (Proposition 1). We show that GW, under Tukey's penalization, becomes robust to Huber contamination (Theorem 1) and promotes resilience to underlying distributions (Corollary 2). Provably, it extends the robustification principle native to Robust OT (ROBOT) to distributions supported on distinct mm spaces —in the same way GW extends the transportation problem (OT) onto otherwise incomparable spaces in the form of alignment. We provide algorithms to calculate the Tukey and Huber GW distances, which, in applications such as shape matching, exhibit superior performance compared to existing techniques, under contamination. We also suggest data-dependent parameter tuning schemes that produce precise levels of robustness.

- Offering a finer control over extreme pairwise distances from either space, the second method rather deploys relaxed metrics that preserve topology. The resultant *locally* robust distance, surrogate to GW, becomes a lower bound to the first formulation (Lemma 3). We prove that solving the same boils down to calculating an OT between truncated observations from $\mu$ and $\nu$ (Theorem 4). We also show that the notion can be generalized to define robust distances over probabilistic mm spaces. This eventually leads to a framework that offers denoising capability to Image translation models, assuming a shared latent space.

- The third approach regularizes the optimization based on 'clean' proxy distributions to achieve robust measure-preserving maps. We show its connection to robust OT formulations (Lemma 6) and the sample complexity that such optimizations demand under contamination. The resultant optimization generalizes the notion of partial alignment, as plans corresponding to the latter can be shown to be an amenable candidate of ours. Based on the same, we propose RRGM, a novel image-to-image translation architecture that exhibits superior denoising capacity while generating handwritten digit images under contamination.

We place all additional discussions along with mathematical proofs in the Supplementary Material. For codes, technical details regarding the implementation of algorithms and datasets, we refer the reader to the repository https://github.com/Thecoder1012/RCDA.

## 2 Related Work

Recovering unperturbed transport plans under contamination poses a significant challenge in cross-domain alignment. In most treatments based on GW (and Sturm's GW (Sturm, 2006)), the *unbalanced* relaxation to the class of underlying couplings is used to ensure robustness (Séjourné et al., 2021; De Ponti & Mondino, 2022). As a result, the 'denoised' solutions in both spaces become merely positive Radon measures. In

case the marginal constraints are imposed using the TV norm (instead of Csiszár or $\phi$-divergence), the idea boils down to transporting only a fraction of the mass under the distributions (Chapel et al., 2020; Bai et al., 2024). UCOOT's (Tran et al., 2023) robust formulation to deter Huber contamination utilizes a similar relaxation additionally on the feature spaces of the domains. While such a mass-trimming approach penalizes outliers, the resultant distance suffers significant deviations from its balanced counterpart (Nguyen et al., 2023). Moreover, the alignment problem, fundamentally different from mass transportation, raises more unanswered questions. For example, in most image-to-image translation problems, only one domain runs the risk of contamination. Unbalancing turns out to be ill-posed to handle such a semi-constrained robustification (Le et al., 2021). Also, the landscape of contamination models stretches way beyond that of Huber's, which the current unbalanced techniques are solely equipped to deal with. The most recent technique offering robustness in GW alignment (Kong et al., 2024) reinforces unbalancing, based on *inlying* surrogate distributions over graphs. This, essentially being an upper bound to UGW, carries all the aforementioned issues. As such, a detailed exploration of robust alignment between distinct domains subject to diverse underlying tasks remains overdue.

**Notations:** Given a Polish space $\mathcal{X}$, we denote by $\mathcal{P}(\mathcal{X})$ and $\mathcal{M}(\mathcal{X})$ the set of Borel probability measures and signed Radon measures defined on it, respectively. For $p \in [1, \infty)$, measures $\rho \in \mathcal{P}(\mathcal{X})$ with finite $p$-th absolute moment, $M_p(\rho) \coloneqq \int \|x\|^p \, \rho(dx) < \infty$ form the space $\mathcal{P}_p(\mathcal{X})$. The Total Variation (TV) norm of $\rho \in \mathcal{M}(\mathcal{X})$ is denoted as $\|\rho\|_{\mathrm{TV}} \coloneqq \frac{1}{2}|\rho|(\mathcal{X})$. The space of measurable functions $f : \mathcal{X} \to \mathbb{R}$ satisfying $\|f\|_{L^p(\rho)} \coloneqq (\int |f|^p d\rho)^{1/p} < \infty$ is denoted by $L^p(\rho)$. The pushforward of $\rho \in \mathcal{P}(\mathcal{X})$ by a measurable map $f$ is defined as $f_{\#}\mu = \mu(f^{-1})$. We define the uniform norm as $\|f\|_\infty \coloneqq \sup_{x \in \mathcal{X}} |f(x)|$. The notation $\odot$ denotes the tensor-matrix multiplication, whereas $\odot$ and $\oslash$ signify element-wise product and division in matrices, respectively. The notation used for the Frobenius norm is $\|\cdot\|_{\mathrm{F}}$. Given $a, b \in \mathbb{R}$, we write $a \vee b = \max\{a, b\}$ and $a \wedge b = \min\{a, b\}$. The uniform $\varepsilon$-covering number of a class of functions $\mathcal{F}$, based on $n$ points $\{x_i\}_{i=1}^n$, with respect to (w.r.t.) the metric $d(f, f') \coloneqq \max_{i \in [n]} |f(x_i) - f'(x_i)|$ is denoted as $N_\infty(\varepsilon, \mathcal{F}, n)$. We also write inequalities, suppressing absolute constants, as $\lesssim$ and $\gtrsim$. In case $a \lesssim b$, we equivalently write $a = O(b)$. Given that the previous relation holds for a polylogarithmic function of $b$ (i.e., $a = O(b \log^{O(1)} b)$), we write $a = \tilde{O}(b)$. If there exists a (strong) isometry between the spaces $\mathcal{X}$ and $\mathcal{Y}$, we write $\mathcal{X} \cong \mathcal{Y}$. For such participating spaces $\mathcal{X}$ and $\mathcal{Y}$, we denote by $M \coloneqq \mathrm{diam}(\mathcal{X}) \vee \mathrm{diam}(\mathcal{Y})$ the effective diameter.

## 3 Preliminaries

Before introducing our robust formulations, we review the basics of transportation and alignment between metric measure spaces.

**Optimal Transport and Entropic Regularization:** Given a Polish space $\mathcal{X}$ endowed with a metric $d(\cdot, \cdot)$, the OT problem between $\mu, \nu \in \mathcal{P}(\mathcal{X})$ is defined as

$$\mathrm{OT}_c(\mu, \nu) \coloneqq \inf_{\pi \in \Pi(\mu, \nu)} \int_{\mathcal{X} \times \mathcal{X}} c(x, y) d\pi(x, y), \tag{1}$$

where $c : \mathcal{X} \times \mathcal{X} \to \mathbb{R}_+$ is the lower semi-continuous transportation cost and $\Pi(\mu, \nu) = \{\pi \in \mathcal{P}(\mathcal{X} \times \mathcal{X}) : \pi(\cdot \times \mathcal{X}) = \mu, \pi(\mathcal{X} \times \cdot) = \nu\}$ is the set of couplings between $\mu$ and $\nu$. We note that (1) is the Kantorovich formulation and can be shown to possess a minimizer. Given $c(x, y) = d(x, y)^p$, $p \geq 1$ it defines the $p$-Wasserstein metric $W_p(\mu, \nu) \coloneqq [\mathrm{OT}_c(\mu, \nu)]^{1/p}$ on the space $\mathcal{P}_p(\mathcal{X})$ and metrizes weak convergence (Villani et al., 2009). OT is a typical linear program, and it is an entropic regularization that makes it strictly convex. Given a parameter $\varepsilon > 0$, reinforcing the marginal constraint under the Kullback-Leibler (KL) divergence yields the primal Entropic OT (EOT) problem:

$$\mathrm{EOT}_c^\varepsilon(\mu, \nu) \coloneqq \inf_{\pi \in \Pi(\mu, \nu)} \int_{\mathcal{X} \times \mathcal{X}} c(x, y) d\pi(x, y) + \varepsilon d_{\mathrm{KL}}(\pi | \mu \otimes \nu). \tag{2}$$

Unlike its unregularized counterpart, the convergence rate corresponding to the empirical EOT cost (towards the population limit) becomes devoid of $\mathrm{dim}(\mathcal{X})$ (Mena & Niles-Weed, 2019). Entropic regularization also enables computing $\delta$-approximate estimates of the transport cost in $\tilde{O}(n^2/\delta)$ time (Blanchet et al., 2024).

Despite computational and theoretical prowess, observe that the EOT cost (also OT) and corresponding potentials can be arbitrarily perturbed if either $\mu$ or $\nu$ (or both) is perturbed the slightest in TV.

**The Gromov-Wasserstein distance:** As mentioned before, we call the triplet $(\mathcal{X}, d, \mu)$ a metric measure space, where $\mu$ has full support, i.e. $\text{supp}(\mu) = \mathcal{X}$. While it is technically convenient to define GW as an extension of OT between two distinct mm spaces, we differentiate them based on their origins in mass transportation and object alignment. Given two Polish mm spaces $(\mathcal{X}, d_X, \mu)$ and $(\mathcal{Y}, d_Y, \nu)$, the GW distance in all its generality is defined as

$$d_{\text{GW}}(\mu, \nu) \coloneqq \left( \inf_{\pi \in \Pi(\mu,\nu)} \int_{\mathcal{X} \times \mathcal{Y}} \int_{\mathcal{X} \times \mathcal{Y}} [\Lambda(d_X(x, x'), d_Y(y, y'))]^p d\pi \otimes \pi \right)^{\frac{1}{p}}, \tag{3}$$

where $\Lambda : \mathbb{R}_+ \times \mathbb{R}_+ \to \mathbb{R}_+$ is a pseudometric measuring the extent of distortion, $1 \le p < \infty$. We also sparingly write $d_{\text{GW}}(X, Y)$. Observe that (3) is essentially the $L^p$-relaxation of the Gromov-Hausdorff distance (Mémoli (2011), Section 4.1), an operation similar to what leads to the Kantorovich-Rubinstein formulation in OT ($W_p$). Now, considering $\Lambda = \Lambda_q(a, b) \coloneqq \frac{1}{2}|a^q - b^q|^{1/q}$, $q < \infty$ one can recover the $(p, q)$-GW distance (Arya et al., 2024), which induces a metric over the class of strongly isomorphic[1] mm spaces with finite $p$-diameter, i.e. $\int_{\mathcal{X} \times \mathcal{X}} [d_X(x, x')]^p \mu_X(dx) \mu_X(dx') < \infty$. Different choices of $d_X, d_Y$ also lead to interesting variants of the GW distance, e.g., considering $d_X = \langle \cdot, \cdot \rangle$ (with $p = 2, q = 1$) and $\|\cdot - \cdot\|$ (with $p = 4, q = 2$) makes the corresponding distances invariant to orthogonal transformations and translations, respectively. Bauer et al. (2024) proposes $\mathcal{Z}$-GW distances by further generalizing $d_X : \mathcal{X} \times \mathcal{X} \to \mathcal{Z}$ (also $d_Y$) as network kernels, given any complete and separable metric space $\mathcal{Z}$. Despite enjoying structural maneuverability, unlike OT, GW distances pose a quadratic assignment problem (QAP) and are, in general, NP-hard to compute. While it is still feasible to determine the exact value of $(4, 2)$-GW between spheres (Arya et al., 2024), given samples from arbitrary distributions, one must resort to entropic regularization to ensure computational tractability (Scetbon et al., 2022; Rioux et al., 2024). Following the setup in (3), the Entropic GW (EGW) distance is defined as

$$\text{EGW}^\varepsilon(\mu, \nu) \coloneqq \inf_{\pi \in \Pi(\mu,\nu)} \int_{\mathcal{X} \times \mathcal{Y}} \int_{\mathcal{X} \times \mathcal{Y}} [\Lambda(d_X(x, x'), d_Y(y, y'))]^p d\pi \otimes \pi + \varepsilon d_{\text{KL}}(\pi | \mu \otimes \nu). \tag{4}$$

This becomes particularly useful in case both the mm spaces are Euclidean with $(\mu, \nu) \in \mathcal{P}_4(\mathcal{X}) \times \mathcal{P}_4(\mathcal{Y})$, as it ties the underlying $(4, 2)$-EGW[2] optimization to EOT with an altered cost. However, the issue regarding uncontrolled perturbation under contamination still persists.

## 4 Robustifying Gromov-Wasserstein

Formulating a mechanism that forestalls the effects of contamination in GW is more elusive compared to OT. Firstly, there is the context of the underlying optimization itself. In OT, the treatment ensuring robustness differs based on the task at hand. For example, cases that prioritize a divergence (e.g., generative models requiring a robust loss) usually call for a robust surrogate to $W_p$ only. As a result, relaxations such as unbalancing or mass truncation (equivalently, addition) are often appropriate (Nietert et al., 2022; 2023). The goal in such cases lies mainly to recover $W_p(\mu, \nu)$ based on a robust proxy $W_p^\epsilon(\hat{\mu}_n, \hat{\nu}_n)$, i.e. $|W_p(\mu, \nu) - W_p^\epsilon(\hat{\mu}_n, \hat{\nu}_n)| \to 0$ in probability, where $\epsilon > 0$ denotes the radius of robustness. This can also be achieved by defining a margin on the extent of allowable perturbation while choosing the surrogate (Raghvendra et al., 2024). While such formulations preserve sample complexity, the resultant transport plans $(\pi_\epsilon^*)$ do not carry robust marginals that are also necessarily probability distributions. This becomes crucial when one is also interested in finding a robust measure-preserving map $(T_\epsilon : \text{supp}(\mu) \to \text{supp}(\nu))$ between the two distributions in the sense of Monge. A surrogate loss ignoring the marginal constraints is bound to result in a map whose deviation from the oracle $(T^*)$ has a non-vanishing lower bound (i.e. there exists $\tau_\epsilon > 0$ such that $\|T_\epsilon - T^*\| \gtrsim \tau_\epsilon$). In this regard, Balaji et al. (2020); Le et al. (2021) (ROT)

---

[1]$(\mathcal{X}, d_X, \mu)$ and $(\mathcal{Y}, d_Y, \nu)$ are said to be *strongly isomorphic* if there exists a measure preserving isometry $\phi : \mathcal{X} \to \mathcal{Y}$ (i.e. $\phi_\# \mu = \nu$ and $d_Y(\phi(x), \phi(x')) = d_X(x, x')$) which is also a bijection.

[2]Essentially the square root of the $(4, 2)$-EGW distance. A more convenient way of realizing it is to assume $\Lambda_q(a, b) \coloneqq \frac{1}{2}|a^q - b^q|$ instead, under which the parameters become $p = 2, q = 2$ (Rioux et al., 2024).

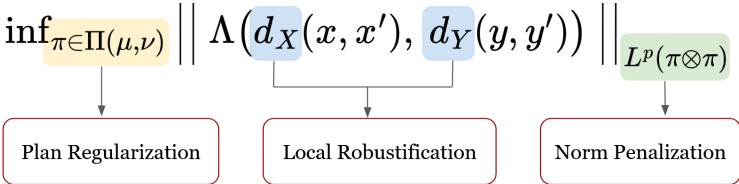

Figure 1: Three disjoint approaches leading to outlier-robustness of different degrees in Gromov-Wasserstein formulations. The forthcoming discussion follows the course: Section 4.1 (■), Section 4.2 (■), and Section 4.3 (■).

maintains a balanced transport by optimizing over proxy distributions instead. While statistical properties of the resulting plans remain unexamined, KL-enforced regularization makes them tractable with comparable efficacy ($\tilde{O}(n^2/\delta)$, where $\delta > 0$ is the error margin and $n$ is the sample size).

Due to its role in alignment (e.g., shape matching) and the involvement of two distinct mm spaces, one needs to be more cautious in approaching the GW problem using similar techniques. Observe that, $d_{\mathrm{GW}}(\mu, \nu)$ calculates the optimal $p$-distortion of a coupling between $\mu$ and $\nu$ (i.e. $\|\Lambda(d_X(x, x'), d_Y(y, y'))\|_{L^p(\pi \otimes \pi)}$). As such, it may become *extremely fragile* (Blumberg et al. (2014), Proposition 4.3) and sustain uncontrolled fluctuation if a single observation from either space is perturbed heavily. Unbalancing readily limits mass allocation to such outliers, resulting in a robust surrogate to $d_{\mathrm{GW}}$. However, unlike OT, it risks sacrificing geometric information contained solely in pairwise distances. This creates significant misalignment between the resultant plan and the isometric benchmark. The latter technique of penalizing the distributions ($\mu$ and $\nu$) themselves, based on robust proxies, also warrants further consideration. For example, when only one of them is contaminated (semi-constrained), the focus must lie on robustifying the pairwise distances, which is not the same as making the law robust. The problem is further confounded if near-isometric robust Monge maps (Dumont et al., 2024) (bidirectional, in case of Reverse Gromov-Monge (Hur et al., 2024)) are sought. On top of these existing ambiguities, the participating mm spaces might suffer different types of contamination with varying intensity. As such, the spectrum of contamination models (Wasserstein, Huber's $\epsilon$-contamination, etc.) needs to be kept in mind.

**Organization.** Based on the varied demands of cross-domain alignment problems, we identify *three* solutions to the robustification problem in GW (see Fig. 1). The *primary* and most immediate way is to arrest the extreme distortions, given pairwise observations $(x, y), (x', y')$. In the process, we introduce Tukey's and Huber's relaxation into the GW metric (Section 4.1). The *second* solution stems from robust surrogates to the metrics $d_X, d_Y$ that limit fluctuations at their nascency while calculating pairwise distances (Section 4.2). Based on the nature of the mm spaces, this approach may propose both structural and optimization-based robustness. In search of robust translation maps, the *third* method advocates relaxing the optimization itself by regularizing the set of plans $\Pi(\mu, \nu)$ (Section 4.3). Intriguing relations to robust OT formulations and duality emerge from this approach.

Notably, the three penalization techniques are tied based on their shared goal of relocation of mass from outliers to inliers. It is the pathway that differs. For *Norm Penalization*, the mass assigned to points from either space that bring about distortion beyond a prespecified threshold ($\tau$) is redistributed. Unlike partial transport, the redistribution in this scheme is not uniform towards the inliers. Pairs of observations that generate about $\tau$ distortion receive the truncated mass. This mechanism makes it immediately appealing for robust recovery of unperturbed GW losses. During *Local Robustification*, a similar truncation is applied to both spaces, and any point contributing to pairwise distances beyond a threshold ($\lambda$) is targeted. Inliers that are around $\lambda$ apart seemingly receive the mass recovered from outliers. This produces a finer penalization regime compared to the previous method. Lastly, our *Plan Regularization* is rooted in the development of robust translation architectures. It enforces a relaxed plan in a Reversible Gromov-Monge-type formulation. Since it readily ties to the partial GW, it risks significant deviation from the unperturbed value. Table 1 summarizes our propositions and the underlying motivations.

Table 1: Comparison of robust GW methods across their objective, robustness guarantee, and utility.

| Criterion | Norm Penalization (TGW/HGW) | Local Robustification (LRGW) | Plan Regularization (RRGM) |
|---|---|---|---|
| *Objective* | Deploy an outlier-robust norm (instead of $L^p$) to measure distortion | Replace the endowed metric of each mm space with robust proxies | Impose mass relaxation in the binding constraint corresponding to the (path-restricted) plan $\pi$ |
| *Robustness Guarantee* | Penalizes the contribution of paired observations from either space that causes $\geq \tau$ distortion | Penalizes contributions of observations from each space that lie $\geq \lambda$ apart | Uniformly redistributes relaxed $\epsilon$ mass from each distribution among respective inliers |
| *Utility* | Robust recovery of GW, shape matching | Cross-domain translation (via Sturm's formulation), shape matching, robust recovery of GW barycenters | Cross-domain translation |

## 4.1 Norm Penalization: Towards Huber's Gromov-Wasserstein

The most recognized contamination model in robust statistics (Huber, 1964) assumes the existence of an arbitrary distribution $\mu_c \in \mathcal{P}(\mathcal{X})$ from which outliers originate. Under the same, it is equivalent to tossing a coin with $\epsilon \in (0,1)$ probability in favor of $\mu_c$ during each independent draw from $\mu$. In case $\mu_c$ admits a density with heavy tails, it implies vastly outlying observations in the sample (see e.g., Figure 2(c)). A similar output may occur if the moments corresponding to $\mu$ and $\mu_c$ differ significantly. In a shape-matching context, this amounts to pronounced disfiguration of shapes. Now, let us recall the definition (3),

$$d_{\mathrm{GW}}(\mu,\nu) = \inf_{\pi \in \Pi(\mu,\nu)} \frac{1}{2}\big\|\Lambda_{X,Y}\big\|_{L^p(\pi \otimes \pi)},$$

where $\Lambda_{X,Y} = 2\Lambda_1(d_X, d_Y)$ in particular. In a sample problem, the $l_p$ norm calculating the departure from isometry is sensitive to unusually large (or small) observations. We employ relaxed $l_p$ norms to curb the effects of such outliers in distortion. Considering the computational complexity of the GW formulation, we first invoke the most intuitive way of implementing a relaxation, namely Tukey's relaxation (Clarkson et al., 2019).

**Definition 1** (Tukey loss function). *Given a threshold $\tau \geq 0$, the p-Tukey loss function for $p \in [1,\infty)$ is defined as*

$$\mathcal{T}_p(x) := \begin{cases} |x|^p & \text{if } |x| \leq \tau \\ \tau^p & \text{otherwise.} \end{cases}$$

Observe that, it is polynomially bounded above[3] with degree $p$. It also induces the corresponding 'norm', $\|f\|_{\mathcal{T}_p(\mu)} = \big(\int_{\mathcal{X}} \mathcal{T}_p(f(x))d\mu(x)\big)^{1/p}$, given $f \in L^p(\mu)$. Though dependent on the parameter $\tau$, it enables one to define

$$\big\|\Lambda_{X,Y}\big\|_{\mathcal{T}_p(\pi \otimes \pi)} := \Big( \int_{\mathcal{X} \times \mathcal{Y}} \int_{\mathcal{X} \times \mathcal{Y}} \mathcal{T}_p\big|d_X(x,x') - d_Y(y,y')\big| \, d\pi \otimes \pi(x,y,x',y') \Big)^{\frac{1}{p}}. \tag{5}$$

We call the quantity $d_{\mathrm{TGW}}(\mu,\nu) := \inf_{\pi \in \Pi(\mu,\nu)} \frac{1}{2}\big\|\Lambda_{X,Y}\big\|_{\mathcal{T}_p(\pi \otimes \pi)}$, *Tukey's GW* (TGW, or specifically $(p,\tau)$-TGW). Clearly, it is a lower bound to the corresponding $d_{\mathrm{GW}}(\mu,\nu)$ and $d_{\mathrm{TGW}} \to d_{\mathrm{GW}}$ as $\tau \to \infty$. It

---

[3]An increasing function $f : \mathbb{R}_+ \to \mathbb{R}_+$ is said to be polynomially bounded above with degree $p$ if $\frac{f(b)}{f(a)} \lesssim \big(\frac{b}{a}\big)^p$.

also carries some major properties of the original GW distance (Mémoli (2011), Theorem 5.1). The non-negativity and symmetry are obvious, and given $\mathcal{X} \cong \mathcal{Y}$, it becomes 0. Conversely, given a threshold $\tau > 0$, $d_{\text{TGW}}(\mu, \nu) = 0$ implies that the mm spaces $\mathcal{X}$ and $\mathcal{Y}$ are isometric, for $p \in [1, \infty)$. Here, we only define the loss for $p < \infty$ since given $\tau > 1$, the essential norm at $p = \infty$ spoils the thresholding and eventually the robustness. Our goal also lies in avoiding large deviations between TGW from its perturbed GW benchmark, caused solely due to large thresholds. As such, it is not in our interest to check the continuity of $\|\cdot\|_{\mathcal{T}_p(\pi)}$ w.r.t. the weak convergence of $\pi$. Hence, a feasible coupling $\pi \in \Pi(\mu, \nu)$ that realizes the infimum only exists for $p \in [1, \infty)$ (Mémoli (2011), Corollary 10.1[4]). Also, the triangle inequality holds owing to the same property for $\|\cdot\|_{\mathcal{T}_p}$.

**Proposition 1** (Metric properties). *Let $(\mathcal{X}, d_X, \mu_X)$, $(\mathcal{Y}, d_Y, \mu_Y)$ and $(\mathcal{Z}, d_Z, \mu_Z)$ denote arbitrary Polish mm spaces. Then,*

*(i) (Triangle inequality)*
$$d_{TGW}(X, Y) \leq d_{TGW}(X, Z) + d_{TGW}(Z, Y).$$

*(ii) Given $\gamma \in \Pi(\mu_X, \mu_Y)$, for all $1 \leq p \leq p' < \infty$*
$$\left\|\Lambda_{X,Y}\right\|_{\mathcal{T}_p(\gamma \otimes \gamma)} \leq \left\|\Lambda_{X,Y}\right\|_{\mathcal{T}_{p'}(\gamma \otimes \gamma)}.$$

*(iii) For $\tau' > \tau \geq 0$, we have $(p, \tau)$-TGW $\leq (p, \tau')$-TGW.*

The properties altogether make $d_{\text{TGW}}$ a pseudometric over the collection of isomorphism classes of mm spaces. It also limits the corruption due to Huber's $\epsilon$-contamination. For simplicity, let us assume only one of the distributions is contaminated, say $\mu$. As such, one now has observations from $\mu' = (1-\epsilon)\mu + \epsilon\mu_c$ instead, where $\mu_c \in \mathcal{P}_p(\mathcal{X})$. Under this setup, the following result gives the extent to which the population-level loss can propagate.

**Theorem 1.** *Given that the two distributions $\mu, \nu$ belong to the same mm space (i.e. they are namely $(\mathcal{X}, d_X, \mu)$ and $(\mathcal{X}, d_X, \nu)$), if $\mu$ suffers Huber's $\epsilon$-contamination, we have*

$$d_{TGW}(\mu', \nu) \leq \tau \epsilon^{\frac{1}{p}} + W_{\mathcal{T}_p}(\mu, \nu),$$

*where $W_{\mathcal{T}_p} := \inf_{\Pi} \|d_X\|_{\mathcal{T}_p}$ is the $OT_{d_X}$ distance under the $p$-Tukey norm.*

The approach underlying the proof of the result, coupled with the monotonicity of the truncation parameter hints towards the fact that TGW can potentially pose as a provably robust estimate to GW, i.e., $d_{\text{TGW}}(\mu', \nu) - d_{\text{GW}}(\mu, \nu) \leq \tau \epsilon^{1/p}$, if the distributions are supported on the same space. If the bound also holds for the absolute difference, thereby marking a concentration inequality, it becomes instrumental in robust shape-matching and generation problems. We establish the same in Remark 3. Theorem 1 also has interesting consequences under specific assumptions on the contamination model. For example, if there exists $k \geq 1$ such that $W_p(\mu, \mu_c) = kW_p(\mu, \nu)$ (Balaji et al., 2020), we have

$$d_{\text{TGW}}(\mu', \nu) \leq (1 + k\epsilon^{\frac{1}{p}})W_p(\mu, \nu),$$

due to the trivial upper bound on $W_{\mathcal{T}_p}$ by the $p$-Wasserstein distance.

**Remark 1** (Resilience under TGW). *Theorem 1 (see proof, Appendix A) also enables one to discuss the 'resilience' of a distribution $\mu$ under $d_{TGW}$. $\mu \in \mathcal{P}(\mathcal{X})$ is said to be $(\rho, \varepsilon)$-resilient w.r.t. the divergence $d$ if $\forall \tilde{\mu} \in \mathcal{P}(\mathcal{X})$ such that $\tilde{\mu} \leq \frac{1}{1-\varepsilon}\mu$, we have $d(\mu, \tilde{\mu}) \leq \rho$, where $0 \leq \varepsilon < 1$ and $\rho \geq 0$. It boils down to checking the maximum change in $W_{\mathcal{T}_p}$ if a $\varepsilon$-fraction of mass under $\mu$ is deleted and renormalized to form $\tilde{\mu} \leq \frac{1}{1-\varepsilon}\mu$. Nietert et al. (2023) show that $\left|\mathbb{E}_{\tilde{\mu}}[d_X(Y, x_0)^p] - \mathbb{E}_{\mu}[d_X(Z, x_0)^p]\right| \leq \rho$ implies resilience under $W_p$,*

---

[4]The only modification required in the first part of the proof of Corollary 10.1 is the following: Define $f : [0, M] \to \mathbb{R}_+$ as $t \mapsto t^p$ if $t \leq \tau$, and $\tau^p$ if $\tau < t \leq M$, which also becomes Lipschitz with constant $pM^{p-1}$. Observe that, $\tau \geq M = \text{diam}(\mathcal{X}) \vee \text{diam}(\mathcal{Y})$ implies the exact function as in Mémoli (2011).

given $x_0 \in \mathcal{X}$ (Lemma 11). In the process, it is sufficient to assume that $\mu$ has a finite p-moment. Observe that,

$$
\begin{aligned}
\left|\mathbb{E}_{\tilde{\mu}}[\mathcal{T}_p(d_X(Y,x_0))] - \mathbb{E}_\mu[\mathcal{T}_p(d_X(Z,x_0))]\right| &\leq \mathbb{E}_{Y\sim\tilde{\mu},Z\sim\mu}\left|\mathcal{T}_p(d_X(Y,x_0)) - \mathcal{T}_p(d_X(Z,x_0))\right| \\
&\leq \mathbb{E}_{Y\sim\tilde{\mu},Z\sim\mu}\left|d_X(Y,x_0)^p - d_X(Z,x_0)^p\right| \\
&\leq \sqrt{Var_{\tilde{\mu}}[d_X(Y,x_0)^p]} + \sqrt{Var_\mu[d_X(Z,x_0)^p]} + \\
&\qquad \left|\mathbb{E}_{\tilde{\mu}}[d_X(Y,x_0)^p] - \mathbb{E}_\mu[d_X(Z,x_0)^p]\right|,
\end{aligned}
$$

where the first inequality is due to Jensen's inequality. As such, since the variances are finite, the mean resilience also implies the same under $\mathcal{T}_p$. However, in case $\tilde{\mu}$ results in a vastly distinct variance, the associated resilience bound on $W_{\mathcal{T}_p}$ becomes weak.

**Corollary 2** (Nietert et al. (2023)). *Given $Z \sim \mu$ and $x_0 \in \mathcal{X}$, let $\mathbb{E}_\mu[d_X(Z,x_0)^p] \leq \sigma^p$ for some $\sigma \geq 0$. If $\mathcal{T}_p(d_X(Z,x_0))$ is $(\rho,\varepsilon)$-resilient in mean, then $\mu$ is $\left(2\left((\rho^{\frac{1}{p}} + \varepsilon^{\frac{1}{p}}(\sigma \wedge \tau)) \wedge \varepsilon^{\frac{1}{p}}\tau\right), \varepsilon\right)$-resilient w.r.t. $W_{\mathcal{T}_p}$.*

Observe that the bound is non-trivial only when $\sigma \leq \tau$. While a user-defined $\tau$ ensures resilience for distributions having thicker tails, the result hints towards distributions ($\mu$) that imply sharper resilience bounds. One immediate example is the class of sub-Gaussian distributions.

**Remark 2** (Lower bound to ROBOT). *TGW has a surprising relation to existing robust efforts in OT, following from Theorem 1. Given a transportation cost $c : \mathcal{X} \times \mathcal{X} \to \mathbb{R}_+$, Mukherjee et al. (2021) (formulation 2) define the $\lambda$-ROBOT distance between $\mu, \nu \in \mathcal{P}(\mathcal{X})$ as $OT_{c_\lambda}(\mu,\nu)$, where $c_\lambda(x,y) := l_{2\lambda}(c(x,y)) = \min\{c(x,y), 2\lambda\}$. Specifically for $c = d_X$, we write $W_{1,2\lambda}(\mu,\nu)$. It metrizes the underlying class of distributions and was shown earlier to lead to faster cost computation in tasks such as image retrieval (Pele & Werman, 2009). On the other hand, if $p = 1$, the Tukey loss boils down to the exact functional $l_\lambda(x) = \min\{x,\tau\}$, $x > 0$. As such, for any coupling $\pi$, by assuming $\tau = 2\lambda$ we have*

$$
\begin{aligned}
\left\|\min\{2\Lambda_1,\tau\}\right\|_{L^1(\pi\otimes\pi)} &\leq \left\|\min\{d_X(x,y) + d_X(x',y'),\tau\}\right\|_{L^1(\pi\otimes\pi)} \\
&\leq 2\left\|\min\{d_X(x,y),\tau\}\right\|_{L^1(\pi)}.
\end{aligned}
$$

*Taking infimum over $\Pi(\mu,\nu)$ we conclude $(1,2\lambda)$-$d_{TGW} \leq W_{1,2\lambda}$.*

**Remark 3** (Concentration). *In reality, often outlying observations find their way into the pool of samples, which is unlike drawing i.i.d. replicates from a contaminated distribution $\mu' = (1-\epsilon)\mu + \epsilon\mu_c$. Rather, in a set of samples $\{x_i\}_{i=1}^m$, we are left with $|\mathcal{I}|$ i.i.d. observations from $\mu$ and the rest, $|\mathcal{O}| := n - |\mathcal{I}|$ drawn independently from adversaries. If the outliers also follow $\mu_c$ identically and $|\mathcal{O}| = m\epsilon$, it becomes equivalent to Huber's contamination regime in an empirical setup. Lecué & Lerasle (2020) call this the $\mathcal{O} \cup \mathcal{I}$ framework. This is crucial since it allows one to comment on the concentration of the empirical $d_{TGW}$. In such a setup, given samples $\{(x_i,y_j)\}^{m,n}$, we get for $p = 2$*

$$
\left|d_{TGW}^2(\hat{\mu}_m,\hat{\nu}_n) - d_{GW}^2(\hat{\mu}_m^{\mathcal{I}},\hat{\nu}_n^{\mathcal{I}})\right| \leq \left|\frac{|\mathcal{I}^X||\mathcal{I}^Y|}{mn} - 1\right| M^4 + \tau\left(\frac{|\mathcal{O}^X|}{m} + \frac{|\mathcal{O}^Y|}{n} - \frac{|\mathcal{O}^X||\mathcal{O}^Y|}{mn}\right),
$$

*where $\hat{\mu}_m, \hat{\nu}_n$ are the usual empirical distributions based on $\{(x_i,y_j)\}^{m,n}$, and $\hat{\mu}_m^{\mathcal{I}} := |\mathcal{I}^X|^{-1} \sum_{i \in \mathcal{I}^X} \delta_{x_i}$ is the same based on inliers. The same goes for $\hat{\nu}_n^{\mathcal{I}}$ in the other space. Moreover, $\mathcal{O}^X$ and $\mathcal{O}^Y$ denote the set of outliers. As such, given $|\mathcal{O}^X| \vee |\mathcal{O}^Y| = o(m \wedge n)$, obtaining TGW may be done alternatively by calculating GW solely based on the inliers. The associated error becomes arbitrarily small. Moreover,*

$$
\left|\mathbb{E}\left[d_{TGW}^2(\hat{\mu}_m,\hat{\nu}_n)\right] - d_{GW}^2(\mu,\nu)\right| \leq \mathbb{E}\left|d_{TGW}^2(\hat{\mu}_m,\hat{\nu}_n) - d_{GW}^2(\hat{\mu}_m^{\mathcal{I}},\hat{\nu}_n^{\mathcal{I}})\right| + \mathcal{E}_{\mathcal{I}}, \tag{6}
$$

*where, (6) utilizes Jensen's inequality and the estimation error $\mathcal{E}_{\mathcal{I}}$ is bounded from above as the following due to Zhang et al. (2024), Theorem 3*

$$
\mathcal{E}_{\mathcal{I}} := \left|\mathbb{E}[d_{GW}^2(\hat{\mu}_m^{\mathcal{I}},\hat{\nu}_n^{\mathcal{I}})] - d_{GW}^2(\mu,\nu)\right| \lesssim \frac{M^4}{\sqrt{|\mathcal{I}^X| \wedge |\mathcal{I}^Y|}} + (1 + M^4) \bigvee_{\mathcal{I}^X,\mathcal{I}^Y} |\mathcal{I}|^{-\frac{2}{(d \wedge d') \vee 4}} (\log|\mathcal{I}|)^{\mathbb{1}\{d \wedge d' = 4\}}.
$$

*As such, in the $\mathcal{O} \cup \mathcal{I}$ framework, TGW estimates the squared GW distance between inlying distributions asymptotically unbiasedly, and hence robustly.*

The theoretical richness of TGW gives us a solid foundation to search for better approximations using data-dependent thresholding. It is also quite intuitive that a misspecified $\tau > 0$ may lead to heavier penalization than required, generating a large deviation from GW. In practice, even minute fine-tuning errors lead to a significant loss in tail information. A smoother thresholding may achieve a nearer robust approximation without sacrificing favorable properties. This leads us to the Huber 'norm'.

**Definition 2** (Huber loss function). *The Huber loss with threshold $\tau > 0$ is defined as*

$$\mathcal{H}(x) := \begin{cases} x^2/2\tau & \text{if } |x| \leq \tau \\ |x| - \tau/2 & \text{otherwise,} \end{cases}$$

*which induces the corresponding norm,* $\|f\|_{\mathcal{H}(\mu)} = \left( \int_{\mathcal{X}} \mathcal{H}(f(x)) d\mu(x) \right)^{1/2}$.

Observe that, $\mathcal{H}(\cdot)$ is continuously differentiable and given $\tau \simeq 0$, closely approximates the $l_1$ loss. The robust penalization also becomes data-dependent, making it essential in robust M-estimation and regression (Loh, 2017). Following our previous formulation, for $(\mu, \nu) \in \mathcal{P}_2(\mathcal{X}) \times \mathcal{P}_2(\mathcal{Y})$, we define $d_{\mathrm{HGW}}(\mu, \nu) := \inf_{\pi \in \Pi(\mu,\nu)} \frac{1}{2} \|\Lambda_{X,Y}\|_{\mathcal{H}(\pi \otimes \pi)}$, namely the *Huber's GW* (HGW). Addressing the robustness of GW for $p = 2$ in particular, $\|\Lambda_{X,Y}\|_{\mathcal{H}}$ does not admit a monotonic property. However, based on the fact that $\mathcal{H}^{\frac{1}{2}}$ is subadditive (Clarkson & Woodruff, 2014), one can recover a Minkowski-type inequality as in TGW. Moreover, HGW poses as a robust estimate of the corresponding GW value as it follows a property similar to Theorem 1. The result involves defining the Huber version of the modified Wasserstein 'distance' $W_{\mathcal{H}} := \inf_{\Pi} \|d_X\|_{\mathcal{H}}$. Consequently, it promotes resilience to a distribution under it upon mass truncation.

To empirically demonstrate HGW's robustness, we devote the rest of the section to building an algorithm to solve a sample HGW. Given $m$ and $n \in \mathbb{N}_+$ i.i.d. samples from $\mu$ and $\nu$ respectively, let us denote the two pairwise distance matrics (based on $d_X$ and $d_Y$) as $C^X \in \mathbb{R}_+^{m \times m}$ and $C^Y \in \mathbb{R}_+^{n \times n}$. Then, $d_{\mathrm{HGW}}^2$ boils down to solving the familiar non-convex optimization (Peyré et al., 2016)

$$\min_{\pi \in \Pi(\hat{\mu}_m, \hat{\nu}_n)} \sum_{i,i',j,j'} \mathcal{H}(C_{ii'}^X - C_{jj'}^Y) \pi_{ij} \pi_{i'j'} = \min_{\pi \in \Pi(\hat{\mu}_m, \hat{\nu}_n)} \langle \mathcal{H}(C^X - C^Y) \odot \pi, \pi \rangle, \tag{7}$$

where $\hat{\mu}_m, \hat{\nu}_n$ are empirical distributions or rather simplexes and $\Pi \in \mathbb{R}_+^{m \times n}$. Note that $\mathcal{H}(C^X - C^Y) := (\mathcal{H}(C_{ii'}^X - C_{jj'}^Y))_{i,j,i',j'}$ are the pairwise distortions truncated according to the Huber loss. To adapt to the Sinkhorn scaling framework (Cuturi, 2013), we additionally impose an entropic regularization $d_{\mathrm{KL}}(\pi)$ to (7), which at the $k$-th iteration calculates $d_{\mathrm{KL}}(\pi|\pi^k) = \langle \pi, \log \pi - \log \pi^k \rangle$. The resultant Huber's EGW formulation follows Algorithm 1. It is immediately beneficial for computing a robust loss, compared to Unbalanced GW (UGW) or Partial GW (PGW), since it results in marginal distributions and computationally scales with the EGW ($O(m^2 n^2)$) exactly.

While we present a simple working algorithm[5], HEGW also adapts to lower-complexity approximations. In Algorithm 1, computing the cost $\mathcal{C}(\pi)$ alone incurs the high complexity $O(m^2 n^2)$. However, we can write $\mathcal{H}(a - b) = f_1(a) + f_2(b) - h_1(a)h_2(b)$, where given $|a - b| \leq \tau$, $f_1(a) = a^2/2\tau$, $f_2(b) = b^2/2\tau$, $h_1(a) = a/\tau$, $h_2(b) = b$ and if $a - b > \tau$, we have $f_1(a) = a, f_2(b) = -b, h_1(a) = \tau, h_2(b) = 1/2$. As such, the cost computation can be eased down to $O(m^2 n + mn^2)$ (Peyré et al. (2016), Proposition 1) per iteration. This is the best complexity achievable if $\mu, \nu$ are not sliced first or no additional constraint on the matrices satisfying lower ranks is imposed. However, if along with the robust penalization, we identify a set of indices $\mathcal{S} = \{(i,j)\}$ with $|\mathcal{S}| = s$, such that

$$\tilde{\mathcal{H}}(C_{ii'}^X - C_{jj'}^Y) = \begin{cases} \mathcal{H}(C_{ii'}^X - C_{jj'}^Y) & \text{if } (i', j') \in \mathcal{S} \\ 0 & \text{otherwise,} \end{cases}$$

---

[5]This allows seamless integration into existing libraries: https://pythonot.github.io/

---

**Algorithm 1** Huber's Entropic Gromov-Wasserstein

---

**Input:** Initialised distributions $p, q$, regularization parameter $\varepsilon$, number of inner and outer iterations $N_2, N_1$.
**Output:** HEGW

---

Compute pairwise distance matrices $C^X, C^Y$
Initialise $\pi^{(0)} = pq^T$, $a^{(0)} = 1_m$, and $b^{(0)} = 1_n$
**for** $i \in \{0, 1, \cdots, N_1 - 1\}$ **do**
    $\mathscr{C}(\pi^{(i)}) \leftarrow \mathcal{H}(C^X - C^Y) \odot \pi^{(i)}$                                   $\triangleright$ Compute cost matrix
    $K^{(i)} \leftarrow \exp\left(-\frac{\mathscr{C}(\pi^{(i)})}{\varepsilon}\right) \odot \pi^{(i)}$                                $\triangleright$ Compute kernel
    **for** $j \in \{0, 1, \cdots, N_2 - 1\}$ **do**
        $\{a^{(j+1)}, b^{(j+1)}\} \leftarrow \{p \oslash (K^{(i)} b^{(j)}), q \oslash (K^{(i)T} a^{(j+1)})\}$           $\triangleright$ Sinkhorn scaling
    **end for**
    $\pi^{(i+1)} \leftarrow \text{diag}(a^{(N_2)}) K^{(i)} \text{diag}(b^{(N_2)})$
**end for**
Return $\langle \mathscr{C}(\pi^{(N_1)}), \pi^{(N_1)} \rangle$

---

where $(i, j) \in \mathcal{S}$, the modified HEGW problem can be solved with accompanying complexity $O(mn + s^2)$ (Li et al., 2023). While this results in a truncated plan ($\pi$), it effectively thwarts outliers in graphs. In this context, we mention that Rioux et al. (2024) explicitly shows that an $(e^\delta - 1)-$oracle approximation of the optimal plan can be achieved using a Sinkhorn scaling algorithm in finite time under entropic regularization, $\delta > 0$ (see Proposition 8). The number of iterations depends on the two distributions, their joint moments, the parameter associated with entropic regularization, and $\delta$. Note that the result only holds for $p = 2$ (which is commonly used in practice), i.e., when the EGW objective can be broken down into a constrained OT problem. Huber's GW adheres to this, given the definition of the associated norm.

**Experiment: Shape Matching with Outliers**

Shape matching constitutes one of the most natural and direct applications of the GW framework, modeling shapes as objects from mm spaces and seeking mappings that preserve pairwise ambient distances as faithfully as possible (Mémoli, 2011). However, contamination in the underlying shape measures, arising from geometric outliers or noise, can severely distort the computed coupling, leading to inconsistent matchings. We deploy HGW for robust 2D shape matching based on point cloud data (Mroueh & Rigotti, 2020). Observe that the underlying mm spaces are essentially $(\mathbb{R}^2, \|\cdot\|_2)$, endowed with measures $\hat{\mu}_m$ and $\hat{\nu}_n$ respectively. Given two shapes (e.g. cat and heart), we identify one as the target and the other as the source. The contamination regime we follow is the following: for $\alpha \in (0, 1)$, we randomly sample $m\alpha$ observations from the source point cloud and replace them with replicates from an adversary $\mu_c$ (e.g., standard bi-variate Cauchy). For comparison, we use the vanilla GW, FGW (Vayer et al., 2020), PGW (Chapel et al., 2020), and UGW (Séjourné et al., 2021) as baselines under $p = 2$. For the unbalanced methods, we allocate unit mass to each point, and for the rest, we normalize. Each method, at each level of contamination, is repeated 100 times to cover for any variation generated due to optimization. The parameters for each distance are selected based on the recommendations by the respective authors, e.g. in the case of FGW, the mixing coefficient of GW and the OT is kept at 0.5. The regularization parameter for PGW is taken as 0.001. We find that only such small values, chosen judiciously, can strike a balance between adequate penalization and a low enough value of the corresponding loss. In TGW (and HGW), the method's accuracy hinges on the selection of $\tau$. Very low values lead to over-penalization and large deviations from the actual robust benchmark. In our study, we devise a data-driven scheme for selecting $\tau$. Given observations $\{(x_i, y_j)\}^{m,n} \sim \mu \otimes \nu$, we scrutinize the distribution of sample distortion values (say, $J_{X,Y}$) $|\|x_i - x_{i'}\|^2 - \|y_j - y_{j'}\|^2|$. An immediate estimate for a threshold that trims outlying $J_{X,Y}$ values is a higher percentile, e.g., $98\%, 95\%$, which we use as a reference. Our choice of an appropriate $\tau$ becomes $\tilde{m} + 3\tilde{\sigma}$, where $\tilde{m}$ and $\tilde{\sigma}$ are the median and mean absolute deviation about median of $J_{X,Y}$. Ideally, for a standard folded Normal distribution, the value turns out $\approx 2.04$ (see Appendix B, Section A). The method enables a dynamic parameter selection that adjusts according to the proportion of outliers. We present a detailed discussion in Appendix B.

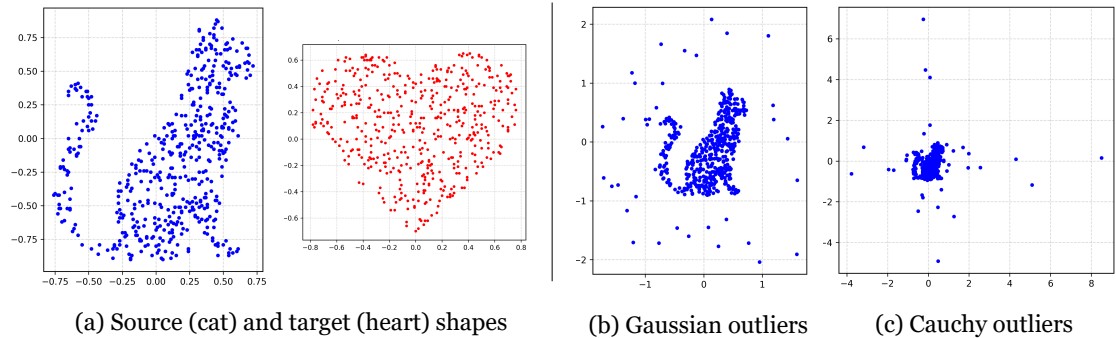

(a) Source (cat) and target (heart) shapes     (b) Gaussian outliers    (c) Cauchy outliers

Figure 2: (a) Point clouds ($m = n = 500$) corresponding to shapes of cat (source) and heart (target). Contaminated source with 20 outliers drawn independently from a standard (b) bivariate Gaussian and (c) bivariate Cauchy.

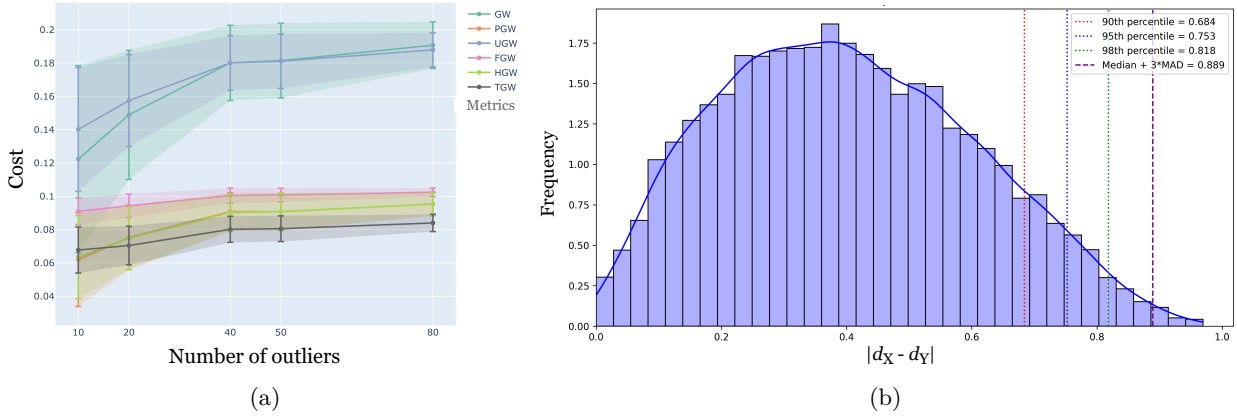

(a)                           (b)

Figure 3: (a) Average loss values under increasing proportion of bi-variate Cauchy outliers $(0.02, 0.04, 0.08, 0.1, 0.16)$ in the source domain. (b) Empirical distribution of deviations between pairwise distances under 80 Cauchy outliers. Realized 95-percentile and $\tilde{m} + 3\tilde{\sigma}$ are 0.753 and 0.889 respectively.

Based on the proposed scheme, the value of $\tau$ is chosen dynamically at each level $\alpha$ for HGW. As a reference, we use the 95-percentile of $J_{X,Y}$ in TGW. The immediate observation is that TGW remains the most stable under increasing contamination. On the other hand, HGW exhibits performance comparable to that of partial mass allocation, as in PGW. Since the threshold only penalizes extreme values of $J_{X,Y}$, pairwise distances between outliers that are similar to those between inliers contribute to the overall loss. This implies a minute increase in HGW. The effect is much pronounced in Gaussian outliers (see Appendix B). The stability of PGW is intuitive since its optimal plan ignores the outliers altogether. Remarkably, HGW simulates the same effect without altering the plan. FGW (with a mixing ratio 0.5) shows elevated fluctuation as its OT component is also vulnerable to contamination. The instability of UGW might stem from mass-splitting and the partial dependence of its plan on outliers (as also pointed out by Bai et al. (2024)). However, despite relying on a full plan connecting outliers to points in the target shape, HGW results in robust estimates of the corresponding loss. In Appendix B, we show the results corresponding to Gaussian contamination and compare the effects due to varying $\tau$.

## 4.2   Local Robustification

Techniques that make the distortion $\Lambda(d_X(x,x'), d_Y(y,y'))$, given $(x,y), (x',y') \in \mu \otimes \nu$, robust to outliers, offer a robust estimate to the extent of deviation from isometry. It is equivalent to selecting a different measurement to compare the two local geometries, however perturbed; for example, TGW. In other words, the penalization applies over the discrepancy in pairwise distances, rather than $d_X(x,x')$ and $d_Y(y,y')$

themselves. This may suffer from inefficiency in information retention, as given only an outlying observation $x' \in \mathcal{X}$, it depreciates the contribution of a 'clean' $d_Y(y, y')$. An earlier-stage robustification of the distances solves this issue. In search of nearer GW approximates, we turn to robust surrogates of $d_X$ and $d_Y$.

For typical choices of metric spaces (for example, $(\mathbb{R}^d, \|\cdot\|)$), it is quite straightforward to retain metric properties of $d_X$ under thresholding (e.g., Tukey) or Winsorization. For example, the *truncated* surrogate $l_\lambda(d_X) = \min\{d_X, \lambda\}$, $\lambda \geq 0$ satisfies non-negativity, symmetry and the triangle inequality. Based on the fact that $\mathcal{T}_2(a - b) \geq \left| l_\lambda(a) - l_\lambda(b) \right|^2$, for $a, b \geq 0$, it immediately improves the corresponding robust GW formulation, evoking a lower bound to Tukey-type distances.

**Lemma 3.** *Given* $(\mu, \nu) \in \mathcal{P}_4(\mathbb{R}^d) \times \mathcal{P}_4(\mathbb{R}^{d'})$, *the* $(2, \lambda)$-*TGW between them satisfies*

$$d_{TGW}(\mu, \nu) \geq \frac{1}{2} \left( \inf_{\pi \in \Pi(\mu,\nu)} \int \int \left| l_\lambda\left( \|x - x'\|^2 \right) - l_\lambda\left( \|y - y'\|^2 \right) \right|^2 d\pi \otimes \pi \right)^{\frac{1}{2}}.$$

We call such formulations, as in the lower bound, *locally robust GW* ($(p, \lambda)$-LRGW, in general). Infima of the corresponding optimization are always realized at relaxed optimal couplings (see Appendix A, Section A). The modification gives greater control over the extent of robustification based on distinct choices of $\lambda, \lambda' \geq 0$ for the two mm spaces. First introduced in Chakrabarty et al. (2025), LRGW also follows most metric properties of GW, particularly non-negativity, symmetry, and triangle inequality. The proofs become similar to showing the same for GW under altered mm spaces of the form $(\mathcal{X}, l_\lambda(d_X), \mu)$, $\lambda > 0$. For completeness, we mention some properties of LRGW, including its dependence on the threshold $\lambda$.

**Proposition 2** (Properties). *Given Polish mm spaces* $(\mathcal{X}, d_X, \mu)$, $(\mathcal{Y}, d_Y, \nu)$; *and* $p \in [0, \infty]$

*(i) for* $\lambda' > \lambda \geq 0$, *we have* $(p, \lambda)$-*LRGW*$(\mu, \nu) \leq (p, \lambda')$-*LRGW*$(\mu, \nu)$. *In fact,*

$$(p, \lambda')\text{-}LRGW(\mu, \nu) - (p, \lambda)\text{-}LRGW(\mu, \nu) \leq \inf_{\pi \in \Pi^\lambda(\mu,\nu)} \left\| l_{\lambda'}(\bar{l}_\lambda(d_X)) - l_{\lambda'}(\bar{l}_\lambda(d_Y)) \right\|_{L^p(\pi \times \pi)},$$

*where* $\bar{l}_\lambda(x) = \max\{x, \lambda\}$ *and* $\Pi^\lambda$ *is the set of couplings optimal for* $(p, \lambda)$-*LRGW*.

*(ii)* $(p, \infty)$-*LRGW*$(\mu, \nu) = p$-*GW*$(\mu, \nu)$.

Observe that, Proposition 2*(ii)* rather holds for any $\lambda \geq M = \operatorname{diam}(\mathcal{X}) \vee \operatorname{diam}(\mathcal{Y})$, given $\operatorname{diam}(\mathcal{X}) = \max_{x,x' \in \mathcal{X}} d_X(x, x')$, which however may become arbitrarily large in the presence of outliers in a sample problem. The result is realized at the limit of truncation since LRGW becomes continuous in $\lambda$ (Chakrabarty et al., 2025). As a consequence of Proposition 2*(ii)*, $\mathcal{X} \cong \mathcal{Y}$ implies that the associated LRGW nullifies. However, the converse does not hold necessarily since, $l_\lambda(d_X(x, x')) = l_\lambda(d_Y(y, y'))$ a.s. does not imply $d_X(x, x') = d_Y(y, y')$ a.s. While this formulation sacrifices non-degeneracy, it preserves geometric sensitivity under appropriately tuned $\lambda$. It delineates an estimated support of the distribution of distances based on inlying observations[6]. Though finer than TGW, such a filtration allows distances corresponding to outlying samples within $\lambda$-radius to each other to pass through. This hints towards addressing contamination due to distribution shifts and mass reallocation as a result. Later in the section, we discuss the origin of local penalization ($l_\lambda$) in a generalized setup. To motivate, we mention that similar costs are often used to devise robust OT-based divergences metrizing $\mathcal{P}(\mathcal{X})$ (see Remark 2). Remarkably, impartially trimmed-$W_p$ due to Czado & Munk (1998) becomes equivalent to trimming the underlying univariate measures (Alvarez-Esteban et al., 2008). In this line, our next result shows that variational representations of LRGW formulations link the alignment problem to certain robust OT costs, relying on trimmed observations instead. Given two mm spaces $(\mathbb{R}^d_{\geq 0}, \|\cdot\|, \mu)$ and $(\mathbb{R}^{d'}_{\geq 0}, \|\cdot\|, \nu)$[7], let us consider the locally robust inner product GW distance (Mémoli, 2011)

$$d_{\text{LRIGW}}(\mu, \nu; \lambda) := \left( \inf_{\pi \in \Pi(\mu,\nu)} \int \int \left| l_\lambda(\langle x, x' \rangle) - l_\lambda(\langle y, y' \rangle) \right|^2 d\pi \otimes \pi(x, y, x', y') \right)^{\frac{1}{2}}.$$

---

[6] Given $x \in \mathcal{X}$, define the map $u_x^\lambda : \mathcal{X} \to [0, \lambda)$ by $u_x^\lambda(x') = l_\lambda(d_X(x, x'))$. Then, $u_{x\#}^\lambda \mu \in \mathcal{P}([0, \lambda))$ denotes the distribution of distances supported on the trimmed interval.

[7] Such subspaces remain Polish equipped with the inherited metric (Fristedt & Gray (2013), Chapter 18.1). We may equivalently choose $[0, 1]^d$, which is sufficient for the IGW formulation. In both cases, the corresponding problem boils down to assessing the alignment between distributions corresponding to non-negative multivariate random variables.

Here, the robustification translates to limiting extreme angular deviation. At $\lambda \to \infty$, the cost circles back to IGW. To state the result, let us first decompose the squared LRIGW cost in the following way:

$$d^2_{\mathrm{LRIGW}}(\mu, \nu; \lambda) = F_1 + F_2,$$

where

$$F_1(\mu, \nu; \lambda) = \int \left| l_\lambda(\langle x, x' \rangle) \right|^2 d\mu \otimes \mu(x, x') + \int \left| l_\lambda(\langle y, y' \rangle) \right|^2 d\nu \otimes \nu(y, y'),$$

$$F_2(\mu, \nu; \lambda) = \inf_{\pi \in \Pi(\mu, \nu)} -2 \int l_\lambda(\langle x, x' \rangle) \, l_\lambda(\langle y, y' \rangle) d\pi \otimes \pi.$$

**Theorem 4** (Locally robust IGW duality)**.** *Given* $(\mu, \nu) \in \mathcal{P}_4(\mathbb{R}^d_{\geq 0}) \times \mathcal{P}_4(\mathbb{R}^{d'}_{\geq 0})$, *define* $M^\lambda_{\mu, \nu} := \sqrt{M_2(\mu; \lambda) M_2(\nu; \lambda)}$, *where for any distribution* $\rho$, $M_2(\rho; \lambda) = \int \left\| l_\lambda(x) \right\|^2 d\rho(x)$, $\lambda \geq 0$. *Then, there exists an upper bound to* $F_2$, *say* $\bar{F}_2$, *satisfying*

$$\bar{F}_2(\mu, \nu; (d \vee d')\lambda^2) = \inf_{\boldsymbol{A} \in \mathcal{D}_{M^\lambda_{\mu, \nu}}} 8\|\boldsymbol{A}\|^2_F + OT_{c^\lambda_{\boldsymbol{A}}}(\mu, \nu),$$

*where* $\mathcal{D}_{M^\lambda_{\mu, \nu}} := [0, M^\lambda_{\mu, \nu}/2]^{d \times d'}$ *and* $c^\lambda_{\boldsymbol{A}} : (x, y) \in \mathbb{R}^d_{\geq 0} \times \mathbb{R}^{d'}_{\geq 0} \mapsto -8l_\lambda(x)^T \boldsymbol{A} l_\lambda(y)$ *denotes the cost function deployed under* $OT_{c^\lambda_{\boldsymbol{A}}}$.

While it is intuitive that a sample-level robustification is sufficient for LR, Theorem 4 additionally provides the deterministic extent to which the associated cost may propagate. The complexity related to computing such a bound shrinks down to that of an OT. To observe that, by denoting $(d \vee d')\lambda^2 = \tilde{\lambda}$, let us write

$$\bar{F}_2(\mu, \nu; \tilde{\lambda}) = \inf_{\mathbf{A} \in \mathcal{D}_{M^\lambda_{\mu, \nu}}} 8\|\mathbf{A}\|^2_F + OT_{c^\lambda_{\mathbf{A}}}(\mu, \nu) = \inf_{\mathbf{A} \in \mathcal{D}_{M^\lambda_{\mu, \nu}}} U^{\mu, \nu}_\lambda(\mathbf{A}).$$

Now, given any other $\tilde{\mu} \in \mathcal{P}_4(\mathbb{R}^d_{\geq 0})$ and $\tilde{\nu} \in \mathcal{P}_4(\mathbb{R}^{d'}_{\geq 0})$, Theorem 4 ensures the existence of $\mathbf{A}, \tilde{\mathbf{A}} \in \mathcal{D}_{M^\lambda_{\mu, \nu}}$ such that $\bar{F}_2(\mu, \nu; \tilde{\lambda}) = U^{\mu, \nu}_\lambda(\mathbf{A})$ and $\bar{F}_2(\tilde{\mu}, \tilde{\nu}; \tilde{\lambda}) = U^{\mu, \nu}_\lambda(\tilde{\mathbf{A}})$. As such, by optimality

$$
\begin{aligned}
\left| \bar{F}_2(\mu, \nu; \tilde{\lambda}) - \bar{F}_2(\tilde{\mu}, \tilde{\nu}; \tilde{\lambda}) \right| &\leq \left| U^{\mu, \nu}_\lambda(\tilde{\mathbf{A}}) - U^{\tilde{\mu}, \tilde{\nu}}_\lambda(\mathbf{A}) \right| + \left| U^{\mu, \nu}_\lambda(\tilde{\mathbf{A}}) - U^{\tilde{\mu}, \tilde{\nu}}_\lambda(\tilde{\mathbf{A}}) \right| \\
&= \left| OT_{c^\lambda_{\mathbf{A}}}(\mu, \nu) - OT_{c^\lambda_{\mathbf{A}}}(\tilde{\mu}, \tilde{\nu}) \right| + \left| OT_{c^\lambda_{\tilde{\mathbf{A}}}}(\mu, \nu) - OT_{c^\lambda_{\tilde{\mathbf{A}}}}(\tilde{\mu}, \tilde{\nu}) \right| \\
&\leq 2 \sup_{\mathbf{A} \in \mathcal{D}_{M^\lambda_{\mu, \nu}}} \left| OT_{c^\lambda_{\mathbf{A}}}(\mu, \nu) - OT_{c^\lambda_{\mathbf{A}}}(\tilde{\mu}, \tilde{\nu}) \right|.
\end{aligned}
\tag{8}
$$

Since $OT_{c^\lambda_{\mathbf{A}}}$ essentially calculates the transportation cost between truncated observations from $\mu$ and $\nu$ — which offers finer control over extreme values — LR turns out to achieve arbitrary accuracy in finding a robust surrogate to the GW cost. By plugging in the empirical distributions $\tilde{\mu} = \hat{\mu}_n$ and $\tilde{\nu} = \hat{\nu}_n$ in the stability bound (8), one can also comment on the sample complexity of $\bar{F}_2$. First, observe that $M^\lambda_{\mu, \nu} \lesssim \lambda^2$ and based on the truncation, the measurable cost $c^\lambda_{\mathbf{A}}$ is absolutely bounded. This narrows down the search for dual potentials $(\phi : \mathbb{R}^d_{\geq 0} \to \mathbb{R})$ corresponding to $OT_{c^\lambda_{\mathbf{A}}}$ to the class $\mathcal{F}_\lambda := \bigcup_{\mathbf{A} \in \mathcal{D}_{M^\lambda_{\mu, \nu}}} \mathcal{F}_{\mathbf{A}, \lambda}$ such that

$$\mathcal{F}_{\mathbf{A}, \lambda} := \left\{ \phi \mid \exists \, \psi : \mathbb{R}^{d'}_{\geq 0} \to \mathbb{R} \ni \phi = \psi^c; \|\phi\|_\infty, \|\psi\|_\infty \lesssim \lambda \right\},$$

where $\psi^c$ (the $c$-transform of $\psi : \mathbb{R}^{d'}_{\geq 0} \to \mathbb{R}$ w.r.t. $c^\lambda_{\mathbf{A}}$) is given by $\psi^c = \inf_y c^\lambda_{\mathbf{A}}(\cdot, y) - \psi(y)$. As such, we can further upper bound (8) to obtain

$$\left| \bar{F}_2(\mu, \nu; \tilde{\lambda}) - \bar{F}_2(\hat{\mu}_n, \hat{\nu}_n; \tilde{\lambda}) \right| \leq 4 \sup_{\phi \in \mathcal{F}_\lambda} \left| \int \phi \, d[\mu - \hat{\mu}_n] \right| + 4 \sup_{\psi \in \mathcal{F}^c_\lambda} \left| \int \psi \, d[\nu - \hat{\nu}_n] \right|,$$

where $\mathcal{F}^c_\lambda := \bigcup_{\mathbf{A} \in \mathcal{D}_{M^\lambda_{\mu, \nu}}} \mathcal{F}^c_{\mathbf{A}, \lambda}$ (Groppe & Hundrieser, 2023). This reduces the problem to controlling the two empirical processes over $\mathcal{F}_\lambda$ and $\mathcal{F}^c_\lambda$. The involvement of raw empirical measures $(\hat{\mu}_n, \hat{\nu}_n)$, susceptible to

outliers, makes further upper-bounding the individual errors in terms of entropy only feasible in an $\mathcal{O} \cup \mathcal{I}$ framework. We identify this as a potential future work since it does not follow directly by adopting the approach of Ma et al. (2023) into the framework of Zhang et al. (2024) (Theorem 3). However, the trivial upper bound (see Remark 2) along with properties such as resilience (Corollary 2) and robust estimation of corresponding GW (Theorem 1) hold as a result of Lemma 3. Nonetheless, it is not apparent how a penalization as $c_{\mathbf{A}}^{\lambda}$ addresses shifts in mass allocation due to contamination. In this line, to motivate our upcoming formulation, let us first unify the underlying spaces under the general framework of *probabilistic mm spaces*.

Probabilistic metric spaces $(\mathcal{X}, p_X)$ are generalizations of typical metric spaces based on the deterministic 'distance' $p_X$ following a modified triangle inequality

$$T\{p_X(x, x')[0, s], p_X(x', x'')[0, t]\} \le p_X(x, x'')[0, s+t],$$

where $T : \mathbb{R}_+ \times \mathbb{R}_+ \to \mathbb{R}_+$ (Bauer et al., 2024), for $x, x', x'' \in \mathcal{X}$ and $s, t \ge 0$. In case $p_X$ is replaced by the distribution function, specific choices of $T$ equate them to Menger spaces (e.g., taking $T = \min$ or $\max$) or Wald spaces ($T = *$, convolution) (Schweizer et al., 1960). Defining a measure on this collection $\mathcal{X}$ completes the triplet $(\mathcal{X}, p_X, \mu)$, which we call a probabilistic mm (pmm) space. In our setup, we particularly choose the pair $(\mathcal{X} = \bar{\mathcal{P}}_p(X), W_p)$, where $\bar{\mathcal{P}}_p \subset \mathcal{P}_p$ is the collection of probability measures with full support on $X \subseteq \mathbb{R}^d$ having finite $p$-moments, $p \ge 1$. This reinforces the notion of generalization based on the fact that given $x, x' \in X$, we have $W_1(\delta_x, \delta_{x'}) = \mathrm{OT}_{d_X}(\delta_x, \delta_{x'}) = d_X(x, x')$. The idea can similarly be extended to an alignment problem between distinct pmm spaces, which brings us back to GW. Observe that, considering a distance $\mathrm{OT}_{l_\lambda(d_X)}$ recovers our earlier LR formulation, as in Lemma 3. We may alternatively choose the Lévy-Prokhorov (LP) metric ($\hat{\rho}$) to construct our pmm space since it ensures that $(\mathcal{X}, \hat{\rho})$ remains Polish, given that $(X, d_X)$ is Polish (Fristedt & Gray (2013), Chapter 18.7).

**Remark 4** (Localization leading to pmm spaces). *Though Dirac measures are the easiest choice to show that individual $x$'s are represented in the pmm space, it is only a special case of a localized measure. Given $\alpha \in \bar{\mathcal{P}}(X)$, a localized measure $m_\alpha^L(x) \in \mathcal{P}(X)$ is tasked with preserving information about the neighborhood of $x \in X$ under the localization operator $L$[8]. Given the choice of the metric as $W_p$ in the pmm space, it implies that $W_p(m_\alpha^L(x), m_\alpha^L(x')) = d_\alpha^L(x, x')$: a generalization over the metric $d_X$. This is the precise reason we name our proposal of a robust GW based on robust $d_X$ 'local robustification'.*

Given two of such pmm spaces $(\mathcal{X}, W_p, \mu)$ and $(\mathcal{Y}, W_p, \nu)$, the GW distance ($\mathcal{Z}$-GW according to Bauer et al. (2024), Section 3.2.5) between them turns out as

$$d_{\mathrm{GW}}(\mu, \nu) := \left( \inf_{\pi \in \Pi(\mu, \nu)} \int_{\mathcal{X} \times \mathcal{Y}} \int_{\mathcal{X} \times \mathcal{Y}} \left| W_p(x, x') - W_p(y, y') \right|^{p'} d\pi \otimes \pi \right)^{\frac{1}{p'}}. \tag{9}$$

For simplicity, we will always assume $p = p'$. However, it is not straightforward to imagine the ambient measures $\mu, \nu$ and the feasible couplings $\Pi$ they form. Let us look at an example that puts the problem in context.

**Example 1** (Gromov-Wasserstein between mixture of Gaussians). *Consider the mm spaces $(\mathcal{N}(\mathbb{R}^d), W_2, \mu)$ and $(\mathcal{N}(\mathbb{R}^{d'}), W_2, \nu)$, where $\mathcal{N}(\mathbb{R}^d)$ is the space of $d$-variate Gaussian distributions. Observe that*

*(I) since Gaussians are exactly identifiable based on their mean and covariance matrix, a finite Gaussian mixture $\in \mathcal{G}(\mathbb{R}^d) := \bigcup_{k \ge 0} \mathcal{G}_k(\mathbb{R}^d)$[9] can be deemed a discrete probability distribution on $\mathcal{N}(\mathbb{R}^d)$.*

*(II) Endowed with $W_2$, $\mathcal{N}(\mathbb{R}^d)$ becomes Polish.*

*(III) Given $\alpha_i = \mathcal{N}(m_i, \Sigma_i)$, $i = \{1, 0\}$ due to Dowson & Landau (1982)*

$$W_2^2(\alpha_0, \alpha_1) = \|m_0 - m_1\|^2 + tr\left[ \Sigma_0 + \Sigma_1 - 2(\Sigma_0^{\frac{1}{2}} \Sigma_1 \Sigma_0^{\frac{1}{2}})^{\frac{1}{2}} \right].$$

---

[8] $L$ maps $\bar{\mathcal{P}}(X)$ to Markov kernels over $X$ (Memoli et al., 2019).
[9] $\mathcal{G}_k(\mathbb{R}^d) :=$ set of Gaussian mixtures with $\le k$ components.

*Hence, for any $\alpha = \sum_{i=1}^{k} a_i \alpha_i \in \mathcal{G}_k(\mathbb{R}^d)$ and $\beta = \sum_{j=1}^{l} b_j \beta_j \in \mathcal{G}_l(\mathbb{R}^{d'})$, there uniquely exist $\mu = \sum_{i=1}^{k} a_i \delta_{\alpha_i} \in \mathcal{P}(\mathcal{N}(\mathbb{R}^d))$ and $\nu = \sum_{j=1}^{l} b_j \delta_{\beta_j} \in \mathcal{P}(\mathcal{N}(\mathbb{R}^{d'}))$ respectively, where $a \coloneqq \{a_i\}_{i=1}^{k} \in \Delta_k$ and $b \coloneqq \{b_j\}_{j=1}^{l} \in \Delta_l$ (using (I)). (III) implies that, under such a $\mu$, $(\mathcal{N}(\mathbb{R}^d), W_2, \mu)$ has finite 2-diameter, i.e. $\int_{\mathcal{N}(\mathbb{R}^d) \times \mathcal{N}(\mathbb{R}^d)} W_2^2(x, x') d\mu(x) d\mu(x') < \infty$ (same holds for $(\mathcal{N}(\mathbb{R}^{d'}), W_2, \nu)$). As such, the corresponding $d_{GW}(\mu, \nu)$ (as in (9), given $p = 2$) exists and follows all metric properties of GW. Salmona et al. (2024) show that solving the problem (9) essentially boils down to*

$$\left( \inf_{\pi \in \Pi(a,b)} \sum_{i,j,s,t} \left| W_2(\alpha_i, \alpha_s) - W_2(\beta_j, \beta_t) \right|^2 \pi_{i,j} \pi_{s,t} \right)^{\frac{1}{2}}, \tag{10}$$

*where $\Pi(a, b)$ is a subset of the simplex $\Delta_{k \times l}$ with marginals $a$ and $b$. This example can be further extended to general distribution classes $\subset \bar{\mathcal{P}}_p(\mathbb{R}^d)$ owing to the fact that $\mathcal{G}(\mathbb{R}^d)$ is dense in $\bar{\mathcal{P}}_p(\mathbb{R}^d)$ for $W_p$, as long as they are complete and separable.*

Evidently, observations from the distributions $\mu$ and $\nu$ can suffer arbitrary corruptions as before. In cases such as Example 1, one or more contaminated individual Gaussian components from either space may contribute to such corruption. To remedy $\epsilon$-contamination in the components, we replace $W_p$ with the smallest cost achieved by 'optimally' removing $\epsilon$-mass from them. This extends our LR framework to alignment models concerning pmm spaces, endowed with $W_p$. Given $\alpha, \beta \in \mathcal{P}(X)$, it is defined as

$$W_p^\epsilon(\alpha, \beta) = \inf_{\substack{\alpha', \beta' \in \mathcal{P}(X): \\ \alpha \in \mathcal{B}_\epsilon(\alpha'), \beta \in \mathcal{B}_\epsilon(\beta')}} W_p(\alpha', \beta'), \tag{11}$$

where $\mathcal{B}_\epsilon(\alpha) \coloneqq \{(1-\epsilon)\alpha + \epsilon\gamma : \gamma \in \mathcal{P}(X)\}$ denotes the $\epsilon$-Huber ball centered at $\alpha$ (Nietert et al., 2022). In this context, $\epsilon \in [0, 1]$ signifies the radius of robustness, which, when chosen distinctly for the two distributions, generalizes the notion (i.e. $W_p^{\epsilon, \epsilon'}$). Remarkably, the dual formulation of $OT_{l_\lambda(d_X)}$ can be derived as a special case of that of (11) (see Appendix A). It also ties the threshold leading to truncation to the underlying optimization. Based on (11), we define the locally robust GW distance between pmm spaces (9) as

$$d_{\text{LRGW}}(\mu, \nu; \epsilon) \coloneqq \left( \inf_{\pi \in \Pi(\mu, \nu)} \int_{\mathcal{X} \times \mathcal{Y}} \int_{\mathcal{X} \times \mathcal{Y}} \left| W_p^\epsilon(x, x') - W_p(y, y') \right|^p d\pi \otimes \pi \right)^{\frac{1}{p}}, \tag{12}$$

which also serves as a robust proxy to $\text{MGW}_2$ (Salmona et al., 2024) or the $\mathcal{Z}$-GW distance. While a generalization is imminent, the asymmetric robustness (only on space $\mathcal{X}$) in (12) is specifically useful in cross-domain generative tasks, e.g., unpaired image-to-image translation.

**Proposition 3** (Dependence on robustness radius). *For any $p \in [1, \infty)$ and $0 \le \epsilon \le \epsilon' \le 1$ we have*

*(i) $d_{LRGW}(\mu, \nu; 0) = d_{GW}(\mu, \nu)$,*

*(ii) $\left| d_{LRGW}(\mu, \nu; \epsilon) - d_{LRGW}(\mu, \nu; \epsilon') \right| \lesssim \left( \frac{\epsilon' - \epsilon}{1 - \epsilon} \right)^{\frac{1}{p}}$.*

The result (i) outlines a necessary condition for the method to be a credible proxy for GW: the LRGW robust surrogate is tight at the uncontaminated limit. On the other hand, part (ii) can be interpreted as: LRGW is Hölder-continuous in the contamination level, with the modulus of continuity controlled by $\left( \frac{\epsilon' - \epsilon}{1 - \epsilon} \right)^{\frac{1}{p}}$, which also reflects the fact that the effective geometry seen by LRGW depends on the residual clean fraction.

**Remark 5** (Local robustness based on Lévy-Prokhorov metric). *The LP distance between $\alpha, \beta \in \mathcal{P}(X)$ is defined as*

$$\hat{\rho}(\alpha, \beta) \coloneqq \inf\{\epsilon > 0 : \beta(A) \le \alpha(A_\epsilon) + \epsilon, \forall \text{ Borel } A \subseteq X\},$$

*where $A_\epsilon = \{x \in X : d_X(x, A) \le \epsilon\}$ is the closed $\epsilon$-ball around $A$. It inherently carries mass allocation robustness, allowing free movement of an $\epsilon$-fraction mass between $\alpha$ and $\beta$. To observe the same, let us write LP in its alternative characterization due to Strassen's theorem (Villani (2021), Section 1.4)*

$$\inf_{\pi \in \Pi(\alpha, \beta)} \left\{ \inf\{\epsilon > 0 : \pi(\{(x, y) : d_X(x, y) > \epsilon\}) \le \epsilon\} \right\}. \tag{13}$$

*As such, it follows that $\left|\hat{\rho}(\alpha,\beta) - \inf\{\epsilon > 0 : W_\infty^{1-\epsilon}(\alpha,\beta) \leq \epsilon\}\right|$ becomes arbitrarily small, given X has unit diameter (Raghvendra et al., 2024). As a result, the feasibility of LR formulations equivalent to (12) based on LP metrics is guaranteed. We show that local robustification using truncation, i.e., $l_\lambda(d_X)$, also extends to pmm spaces under the LP metric. It becomes evident due to the relation between $W_{p,\lambda}$ (ROBOT) and the modified LP.*

**Proposition 4.** *Define $\hat{\rho}_\lambda(\alpha,\beta) = \inf_{\pi \in \Pi(\alpha,\beta)} \left\{ \inf\{\epsilon > 0 : \pi(\{l_\lambda(d_X(x,y)) > \epsilon\}) \leq \epsilon\} \right\}$, for $\lambda > 0$. Then*

$$\frac{1}{1+\lambda}W_{1,\lambda} \leq \hat{\rho}_\lambda \leq \sqrt{W_{1,\lambda}}.$$

The above discussion on LRGW's proficiency at recovering unperturbed GW from a theoretical viewpoint can be empirically supported by its performance on real tasks. Chakrabarty et al. (2025) show that LRGW, being a natural lower bound to TGW, remains consistently stable under increasing proportion of extreme outliers in both domains in a shape matching problem. It also preserves unperturbed interpolations in the form of fixed-support barycenters in $\mathbb{R}^2$. In the following section, we explore its role in cross-domain translations, which in turn reveals intriguing connections to SOTA objectives.

### 4.2.1 Sturm's GW and robust image-to-image translation

Instilling intrinsic robustness to outliers in a measurable map (induced by neural networks) $\mathcal{X} \mapsto \mathcal{Y}$, learned based on contaminated data requires additional regularization. While introducing trimming methods (as in HGW or LRGW) in a GM setup may lead to denoised translations, the approximation capability of the maps thus produced remains shrouded. In this section, we rather connect natural upper bounds of GW to losses that fuel existing I2I models. This way, we ensure robustness guarantees without compromising complexity in an I2I translator.

In our pursuit, let us first define Sturm's GW distance (Sturm, 2006) between the altered spaces $(\mathcal{X}, l_\lambda(d_X), \mu)$ and $(\mathcal{Y}, l_\lambda(d_Y), \nu)$ as $\inf_{\tilde{d},\pi} ||\tilde{d}||_{L^p(\pi)}$. Here, the infimum is over $\pi \in \Pi(\mu,\nu)$ and $\tilde{d} \in \mathscr{D}(l_\lambda(d_X), l_\lambda(d_Y))$, the set of *metric couplings*[10]. Observe that, for $\lambda > 0$, $\mathscr{D}^\lambda := l_\lambda(\mathscr{D}(d_X, d_Y)) \subset \mathscr{D}(l_\lambda(d_X), l_\lambda(d_Y))$. As such,

$$\inf_{\tilde{d} \in \mathscr{D}(l_\lambda(d_X), l_\lambda(d_Y)),\pi} ||\tilde{d}||_{L^p(\pi)} \leq \inf_{\tilde{d} \in \mathscr{D}^\lambda,\pi} ||\tilde{d}||_{L^p(\pi)} =: d_{RSGW}(\mu,\nu), \tag{14}$$

which we call the $(p,\lambda)$-*Robust Sturm's GW*, $p \in [1,\infty)$. The distance essentially embodies a locally robust formulation based on the couplings between $d_X$ and $d_Y$. We refer to the lower bound of (14) as the *lower* RSGW. Complementing the relationship between GW and Sturm's GW, the respective robust formulations follow a similar inequality.

**Theorem 5** (Upper bound to LRGW). *Given $p \in [1,\infty)$ and $\lambda \geq 0$,*

$$(p,\lambda)\text{-}LRGW \leq 2(p,\lambda)\text{-}lRSGW.$$

*Also, for $\delta \in (0, \frac{1}{2}]$, whenever $(p,\lambda)$-LRGW$\leq \delta^5$, we have $(p,\lambda)$-lRSGW$\lesssim \lambda(4\lambda + \delta)^{\frac{1}{p}}$.*

The immediate benefit of such a result is that minimizing a realized loss of the RSGW-type arbitrarily in an I2I setup establishes a near-isometric relation between the two image spaces. Intuitively, this should produce robust translations that preserve geometry. To invoke the notion of an actual architecture, we recall the equivalent formulation of Sturm's GW. In our setup,

$$\begin{aligned} d_{RSGW}(\mu,\nu) &= \inf_{d \in \mathscr{D}(d_X, d_Y),\pi} \left\| l_\lambda(d) \right\|_{L^p(\pi)} \\ &= \inf_{\mathcal{Z},\phi_X,\phi_Y} W_{p,\lambda}(\phi_{X\#}\mu, \phi_{Y\#}\nu), \end{aligned} \tag{15}$$

where the infimum is over all isometric embeddings $\phi_X : \mathcal{X} \to \mathcal{Z}$ and $\phi_Y : \mathcal{Y} \to \mathcal{Z}$ into a *latent space* $\mathcal{Z}$, endowed with the metric $d$ (Sturm (2006), Lemma 3.3). We deliberately bring on the term 'latent space' to

---

[10]$\mathscr{D}(d_X, d_Y) :=$ the set of metrics on $\mathcal{X} \sqcup \mathcal{Y}$ that extend $d_X$ and $d_Y$ (Sturm, 2006).

emphasize the connection to I2I architectures. The formulation also makes it sufficient to embed observations from both spaces into an optimal $\mathcal{Z}$ prior to truncation. Observe that if instead of $W_{p,\lambda}$, we deploy $\hat{\rho}_\lambda$ based on the metric $d$ in (15), we obtain the robust Gromov-Prokhorov (RGP) metric (Blumberg et al. (2014), Section 2.5). By definition, RGP $< \epsilon$ implies the existence of a metric space $\mathcal{Z}$ with embeddings $\phi_X, \phi_Y$ into it, that satisfy $\hat{\rho}_\lambda(\phi_{X\#}\mu, \phi_{Y\#}\nu) < \epsilon$.

**Equivalence of losses:** With the foundation in place, we explore the similarity between the loss (15) and that of I2I translation models such as UNIT (Liu et al., 2017) and GcGAN (Fu et al., 2019). We choose the two models based on their sustained relevance in the domain. However, the equivalence about to be shown can be extended to models that recognize the role of a latent space or deploy a cycle-consistency (CC) loss, such as DistanceGAN (Benaim & Wolf, 2017), StarGAN (Choi et al., 2018), or MUNIT (Huang et al., 2018). The cornerstone of successful I2I learning is inarguably the CC loss. In the population regime, it can be expressed as $W_1(\mu, G \circ F_{\#}\mu)$ for the space $\mathcal{X}$, where $F, G$ are optimized over measure-preserving (transport) maps parametrized using neural networks (NN). It becomes equivalent to optimizing the commonly used $L^1$ norm if $\mu$ possesses a Hölder smooth density (Chakrabarty & Das, 2022).

$$
\begin{array}{c}
(\mathcal{X}, \mu) \xrightleftharpoons[G]{F} (\mathcal{Y}, \nu) \\
\phi_X \quad \phi''_X \quad \phi_Y \quad \phi'_Y \\
(\mathcal{Z}, \omega)
\end{array}
\tag{16}
$$

Now, recognizing the existence of a shared latent space, we may construct $G = \phi''_X \circ \phi_Y$ and $F = \phi'_Y \circ \phi_X$, where $\phi'_Y : \mathcal{Z} \to \mathcal{Y}$ is the left-inverse of $\phi_Y$, and $\phi''_X : \mathcal{Z} \to \mathcal{X}$ is the right-inverse of $\phi_X$. We can assume them to be full functional inverses, as the same applies to isomorphic embeddings, in which case CC is achieved a.s. However, the maps $\phi''_X, \phi'_Y$ may not be measure-preserving in general. Therefore,

$$
\begin{aligned}
W_1(\mu, G \circ F_{\#}\mu) &= \inf_{\pi \in \Pi(\mu, F_{\#}\mu)} \int d_X(x, \phi''_X \circ \phi_Y(y)) \, d\pi(x,y) \\
&= \inf_{\pi \in \Pi(\mu, F_{\#}\mu)} \int d\Big(\phi_X(x), (\phi_X \circ \phi''_X) \circ \phi_Y(y)\Big) \, d\pi(x,y) \tag{17} \\
&= \inf_{\pi \in \Pi(\phi_{X\#}\mu, (\phi_Y \circ \phi'_Y) \circ \phi_{X\#}\mu)} \int d(x,y) \, d\pi(x,y) \\
&= \inf_{\pi \in \Pi(\omega, (\phi_Y \circ \phi'_Y)_{\#}\omega)} \int d(x,y) \, d\pi(x,y) \\
&= W_1(\omega, \phi_Y \circ \phi'_{Y\#}\omega), \tag{18}
\end{aligned}
$$

where $d \in \mathscr{D}(d_X, d_Y)$[11]. We list out some immediate observations from the upper derivation. Firstly, constructing such a chaining ($\mathcal{X} \leftarrow \mathcal{Z} \leftarrow \mathcal{Y}$) reduces the problem of achieving CC in $\mathcal{X}$ to that of ensuring accurate autoencoding of $\omega$ based on the contextual latent law $\nu$ (18). The same choice of $F, G$ also guarantees CC in $\mathcal{Y}$ a.s. Observe that, for any $F$ satisfying $F_{\#}\mu = \nu$, given an optimal $\mathcal{Z}$ and the pair of embeddings into it, (17) equates to SGW. As such, SGW is an upper bound to the optimal CC loss $\inf_{F,G} W_1(\mu, G \circ F_{\#}\mu)$ when $G$ follows our construction optimally, which is rather intuitive. It becomes much simpler if $\mu, \nu \in \mathcal{P}_2^{\mathrm{ac}}(\mathbb{R}^d)$, in which the construction can be made uniquely (see Appendix A, Section D).

The second common loss component between UNIT and GcGAN is the constraint that ensures $F_{\#}\mu = \nu$ and $G_{\#}\nu = \mu$. Typically, the imposition is done using a GAN or WGAN objective. In our framework, a WGAN loss under 1-Lipschitz critics turns out as

$$
\begin{aligned}
W_1(\mu, G_{\#}\nu) &= \inf_{\pi \in \Pi(\mu, \nu)} \int d_X(x, G(y)) \, d\pi(x,y) \\
&= W_1(\phi_{X\#}\mu, \phi_{Y\#}\nu),
\end{aligned}
$$

---

[11]To avoid complications, we do not differentiate the two $W_1$ metrics in terms of notations, which are indeed calculated based on $d_X$ and $d$ respectively.

which, again, at an optimal latent space equals SGW. Similarly, the loss $W_1(F_\#\mu, \nu)$ boils down to solving (18). As such, it is sufficient to optimize SGW between $\mu$ and $\nu$ subject to the autoencoding constraint (18) to solve the UNIT problem.

The only additional term GcGAN employs is namely the geometric-consistency (GC) loss. In the population regime, a $\mathcal{X} \xrightarrow{F} \mathcal{Y}$ translation model incurs a GC loss

$$W_1(F \circ s_{X\#}\mu, s_Y \circ F_\#\mu),$$

where $s_X$ and $s_Y$ are automorphisms in $\mathcal{X}$ and $\mathcal{Y}$ respectively, e.g. rotation. Based on our construction, considering $s_X = \phi''_X \circ \phi_X$ and $s_Y = \phi'_Y \circ \phi_Y = \text{Id}_Y$ meets the constraint. Combining all the above observations gives the clear impression that effectively choosing a latent space $\mathcal{Z}$ — in turn, enabling appropriate construction of $F$ and $G$ — implies consistent I2I translation in UNIT and GcGAN. Remarkably, all the results hold exactly under the altered metric $W_{1,\lambda}$ (also, $W_1^\epsilon$) since the constructions remain same for $l_\lambda(d_X)$ and $l_\lambda(d_Y)$ (15). As such, robustifying UNIT or GcGAN only requires updating their dependence on SGW to one with RSGW.

### Experiment: Style transfer with noise

Style transfer, the task of rendering the content of one image in the visual style of another, was brought into the deep learning era by Gatys et al. (2015), who framed it as matching Gram matrix statistics of deep feature activations. Subsequent work recast this as a distributional alignment problem, drawing on OT. However, when source and target domains belong to structurally incomparable spaces, classical OT is insufficient, and the GW distance (Mémoli, 2011) provides an appropriate generalization, seeking a structure-preserving coupling across spaces often without requiring a shared embedding. Unlike Euclidean OT-based alignment, GW-based style transfer is sensitive to distortions in the pairwise relational structure, not merely marginal shifts, making standard unbalancing strategies inadequate.

The first experiment we conduct tests the denoising capability of a robust GcGAN deploying (11) during I2I style transfer. Despite an overhaul in the optimization, we call our proposed model 'robust GcGAN' for simplicity. Notably, this is the first outlier-robust cross-domain generative model to our knowledge. Based on the dataset 'Apples-Oranges' (Zhu et al., 2017), the underlying task is to translate the visual style of oranges onto apples that are contaminated. Unlike the Huber setup here, standard Gaussian noise is added to the RGB channels of each target sample (apple). The mixing intensity $\alpha$ is kept at 0.2. We present a detailed discussion on the experimental setup in Appendix B, Section B. As discussed in the previous section, it is sufficient to optimize the RSGW loss for a suitable $\mathcal{Z}$, which in this case is the image space itself. As a regularizer, we add the GC loss, taking $s_X, s_Y$ as 90° clockwise rotations in their respective spaces. Model architecture and the choice of the Lagrangian parameter remain similar to that directed by the GcGAN authors. For comparison, the experiment contains three phases. As in the first row of Figure 4, we generate samples using the original GcGAN (without any modifications) on clean observations (control). The second row shows the degradation in translation once noise is added. Finally, applying our robust formulation at $\epsilon = 0.5$ we observe a significant improvement in images, both qualitative and quantitative. We present our parameter selection scheme in Appendix B, Section B, in the form of an ablation study.

The second experiment is in a different spirit, given the dissimilarity between the dimensions of the two spaces, namely the handwritten digit datasets MNIST ($28 \times 28$) and USPS ($16 \times 16$). Our goal is to assess the robust domain-translation capacity of a UNIT architecture reinforced with RSGW. Unlike style transfer, here, samples from MNIST (base distribution) are subjected to Gaussian noise. Keeping the robustness radius at 0.5, the robust UNIT model produces USPS samples with an improved FID score (see Figure 5) compared to vanilla UNIT. Suspecting that the employment of WGAN instead of vanilla GAN regularization may contribute to the heightened image quality, to deconvolute the improvement (of RUNIT) from the refinement due to robustification, we perform the following controlled ablation experiment. Since the underlying robust penalization and the equivalence of losses hinge on the OT argument, we focus on the modified baseline of UNIT, now equipped with WGAN-based regularization. We find that the modification results in comparable image quality (FID = 301.25 vs. the vanilla UNIT baseline producing FID = 304.39). As such, the improvement shown can be solely attributed to the robustification mechanism.

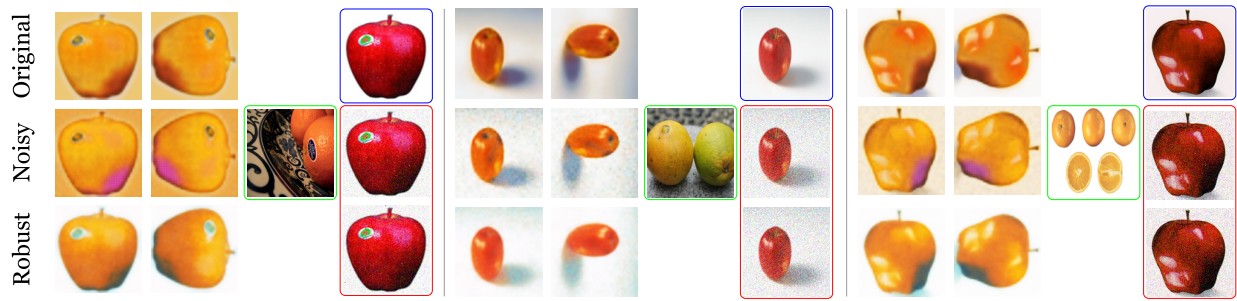

Figure 4: Style transfer performance of robust GcGAN under contamination ($\alpha = 0.2$). Images encircled in 'blue' represent clean target samples, in 'red' are noisy versions of them, and the ones in 'green' act as sources of the style to be transferred. At $\epsilon = 0.5$, the robust translations (third row) maintain sharpness and prevent artifacts from appearing, improving the FID score to 152.65 (compared to 154.74 in the noiseless setting: first row).

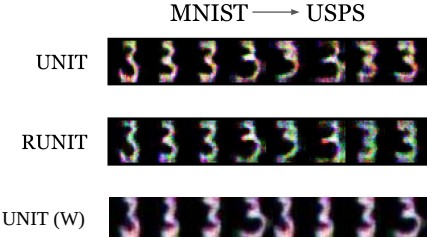

Figure 5: Unpaired translation under contamination ($\alpha = 0.4$) using robust UNIT. At $\epsilon = 0.5$, RUNIT recovers the visual quality of generated USPS samples (FID = 262.48, compared to 304.39 in the case of UNIT (*first row*) and 301.25 as in UNIT with WGAN-based regularization (*bottom row*) under pixel noise.

### 4.3 Plan Robustification

All existing efforts to make GW robust to outliers rely on penalizing the plan $\pi$ based on partial mass transport. Relieving the constraint from aligning all points enables the optimization to filter out outliers as only their contribution to the total mass is ignored. In a GW setup, unbalancing (Séjourné et al., 2021; Tran et al., 2023) in principle achieves the same by relaxing $\Pi(\mu, \nu)$ to $\mathcal{M}_+(\mathcal{X} \times \mathcal{Y})$, i.e. the set of positive Radon measures. However, it does not restrict the amount of mass to be pruned in its imposition of marginal constraints under quadratic $\phi$-divergences. Moreover, it fails to guarantee an optimal plan in the exact sense. While a rescaling afterward produces a joint probability distribution, it only redistributes the leftover mass to inlying points from both spaces uniformly. Using the TV metric instead connects the problem to PGW (Bai et al., 2024). It readily performs the redistribution, thereby linking it to our previous truncation methods (TGW and LRGW). While the redistribution pathway in TGW is not uniform, it is directed toward points that, through their interactions with the other space, result in a $\tau > 0$ distortion. In LR, the redistribution can be interpreted as assigning the truncated mass to points that lie almost $\lambda > 0$ distance apart in their ambient space.

Though imposed on top of an unbalanced GW between surrogates of $\mu$ and $\nu$, Kong et al. (2024) is the only method to date that employs a direct penalization to robustify in the spirit of robust OT (Balaji et al., 2020). Since our motivation lies in constructing a robust I2I translation architecture, we rather prioritize a balanced formulation of the same kind that results in an optimal plan, and eventually maps. As encouragement, we draw an immediate connection between the robust penalization and ROT (Le et al., 2021). Observe that,

the $(4, 2)$-GW distance between Euclidean mm spaces can be fragmented as $d_{\mathrm{GW}}^2(\mu, \nu) = S_1 + S_2$, where

$$S_2(\mu, \nu) = \inf_{\pi \in \Pi(\mu, \nu)} \Gamma(\pi) \coloneqq \inf_{\pi \in \Pi(\mu, \nu)} \int -\|x\|^2 \|y\|^2 \, d\pi(x, y) - 2 \sum_{\substack{1 \leq i \leq d \\ 1 \leq j \leq d'}} \left( \int x_i y_j d\pi(x, y) \right)^2,$$

and $S_1$ depends solely on the marginals $\mu, \nu$ (Zhang et al., 2024). It enables us to define

$$\tilde{S}_2(\mu, \nu) = \inf_{\substack{\tilde{\mu} \in \mathcal{P}(\mathcal{X}) \\ \tilde{\nu} \in \mathcal{P}(\mathcal{Y})}} \min_{\pi \in \Pi(\tilde{\mu}, \tilde{\nu})} \Gamma(\pi) + \lambda_1 D_f(\tilde{\mu}|\mu) + \lambda_2 D_f(\tilde{\nu}|\nu), \tag{19}$$

where $D_f$ is some $f$-divergence and $\lambda_1, \lambda_2 > 0$.

**Lemma 6** (Duality). *Given* $(\mu, \nu) \in \mathcal{P}_4(\mathbb{R}^d) \times \mathcal{P}_4(\mathbb{R}^{d'})$, *if* $D_f \equiv d_{KL}$, *we have*

$$\tilde{S}_2(\mu, \nu) = \inf_{\substack{\tilde{\mu} \in \mathcal{P}(\mathcal{X}) \\ \tilde{\nu} \in \mathcal{P}(\mathcal{Y})}} \inf_{\boldsymbol{A} \in \mathcal{D}_{M_{\tilde{\mu}, \tilde{\nu}}}} 8\|\boldsymbol{A}\|_F^2 + ROT_{c_{\boldsymbol{A}}}(\mu, \nu),$$

*where* $\mathcal{D}_{M_{\tilde{\mu}, \tilde{\nu}}} = [-M_{\tilde{\mu}, \tilde{\nu}}^\infty/2, M_{\tilde{\mu}, \tilde{\nu}}^\infty/2]^{d \times d'}$ *(see Theorem 4) and* $ROT_{c_{\boldsymbol{A}}}(\mu, \nu) = OT_{c_{\boldsymbol{A}}}(\tilde{\mu}, \tilde{\nu}) + \lambda_1 d_{KL}(\tilde{\mu}|\mu) + \lambda_2 d_{KL}(\tilde{\nu}|\nu)$ *is based on the cost* $c_{\boldsymbol{A}}$ *mapping* $(x, y) \mapsto -\|x\|^2 \|y\|^2 - 8x^T \boldsymbol{A} y$.

As such, the optimization underlying robust alignment is essentially a moment-constrained robust OT. If we only regularize based on one marginal (e.g., $\mu$ only), the optimization boils down to solving a semi-constrained problem, namely RSOT. The GW estimate corresponding to such a $\tilde{S}_2(\mu, \nu)$ can be made robust by plugging in the optimal marginals so obtained, $\tilde{\mu}, \tilde{\nu}$ in the functional $S_1$. This marks the potential that the robust penalization (19) has. In search of learnable maps between the spaces, we may narrow down the feasible set of couplings following Hur et al. (2024). Instead of $\Pi(\tilde{\mu}, \tilde{\nu})$, we may choose the path-restricted distributions that follow the *binding constraint*

$$\{\pi : \pi = (\mathrm{Id}, F)_{\#}\tilde{\mu} = (G, \mathrm{Id})_{\#}\tilde{\nu}\} \subset \Pi(\tilde{\mu}, \tilde{\nu}),$$

where $F, G$ are measurable maps between $\mathcal{X}, \mathcal{Y}$ (see illustration 16) and $\tilde{\mu}, \tilde{\nu}$ are supposedly 'clean' marginals. One can also relax this by choosing a larger subclass of $\Pi$ that imposes only $F_{\#}\tilde{\mu} = \tilde{\nu}$ and $G_{\#}\tilde{\nu} = \tilde{\mu}$. In any case, the resultant robust distance— an upper bound to GW under a similar penalization (Hur et al. (2024), Proposition 5.4)— only inculcates maps that transport inlying marginals to those in the other space. In a Huber contamination model, this readily implies that the corresponding set of couplings is non-empty. However, it is in principle different from learning a map possessing a denoising ability in the sense $F_{\#}\mu = \tilde{\nu}$. Maps of the latter kind promote carrying out the optimization over couplings given as

$$\{\pi : \pi = (\tilde{\mathrm{Id}}, F)_{\#}\mu = (G, \tilde{\mathrm{Id}})_{\#}\nu\},$$

where $\tilde{\mathrm{Id}}$ denotes the denoising operation that pushes forward $\mu$ to $\tilde{\mu}$ (also $\nu$ in its ambient space). The set containing such couplings is also non-empty since partial mass transport guarantees the existence of such 'robustifiers' $\tilde{\mathrm{Id}}$, and hence a pair of amenable maps $(F, G)$. Given $\epsilon \in [0, 1)$, let us define the partial couplings

$$\Pi_\epsilon(\mu, \nu) = \{\pi : \pi = (\tilde{\mathrm{Id}}^\epsilon, F)_{\#}\mu = (G, \tilde{\mathrm{Id}}^\epsilon)_{\#}\nu\}, \tag{20}$$

where $\tilde{\mathrm{Id}}^\epsilon_{\#}\alpha \leq (1 - \epsilon)\alpha$, for $\alpha \in \mathcal{P}(\mathcal{X})$. The inequality should be understood setwise. We can also generalize the notion based on distinct mass fractions to be clipped in the two spaces. It is (20) that we base our final proposition on constructing a robust I2I translation model. We call the term $\inf_{\pi \in \Pi_\epsilon(\mu, \nu)} \|d_X - d_Y\|_{L^p(\pi \otimes \pi)}$, the *robust reversible Gromov-Monge* (RRGM) distance. The formulation essentially is a robust surrogate to the RGM distance due to Hur et al. (2024) based on a partial alignment. To favor comparative analysis, we only consider $p = 2$ in our experiments. During transform sampling, RGM uses additional penalization imposing the constraints of measure preservation. The preferred metric for the same is often chosen as Maximum Mean Discrepancy (MMD). To eliminate the critical question of the ideal kernel given an empirical problem, we employ instead $W_1$. Under the same, the RRGM loss[12] can be written in a Lagrangian form

---

[12]The loss (21) relaxes the binding constraint (as in (20)) and only imposes robust measure preservation. As such, it is essentially a lower bound to RRGM.

given as

$$\inf_{F,G} \Big[ \int \Big( d_X(\tilde{\mathrm{Id}}^\epsilon(x), G(y)) - d_Y(\tilde{\mathrm{Id}}^\epsilon(y), F(x)) \Big)^2 d\mu \otimes \nu \Big]^{\frac{1}{2}} + \lambda_1 W_1(\tilde{\mathrm{Id}}^\epsilon_\# \mu, G_\# \nu) + \lambda_2 W_1(F_\# \mu, \tilde{\mathrm{Id}}^\epsilon_\# \nu), \quad (21)$$

where the infimum is over measurable maps and $\lambda_1, \lambda_2 > 0$. We may find a further lower bound to the loss owing to the fact that

$$W_1(\tilde{\mathrm{Id}}^\epsilon_\# \mu, G_\# \nu) \geq W_1^\epsilon(\mu, G_\# \nu), \quad (22)$$

where $W_1^\epsilon$ only carries out a partial transport of $\mu$ asymmetrically (see (11)). The same argument holds for the other term, given an asymmetric $W_1^\epsilon$ employment on the other space. During demanding I2I translations, it is often beneficial to have learnable discriminators over 1-Lipschitz dual maps. The usage of $W_1$ is also advantageous since it enables deploying a larger class of neural network-induced critics. As such, the two added losses can be optimized using WGAN-GP (Gulrajani et al., 2017) architectures. Despite promoting a different redistribution pathway of inlying mass, the sample complexity of transform sampling under $W_1^\epsilon$ should be of a similar order to LR constraints (Nietert et al. (2023), Theorem 4). We note that the convergence rate corresponding to the latter depends only on the inlying sample size (see Appendix A, Section C) if outliers remain bounded in number.

**Remark 6** (Relation to parallel formulations of partial alignment). *Chhoa et al. (2024)'s notion of a relaxed coupling (see Definition 2.2) relates directly to our version of mass trimming during translations (20). Essentially, $\Pi_\epsilon$ can be understood as a path-restricted analogue of $\mathcal{C}_\epsilon$ (due to Chhoa et al. (2024)) under a robust binding constraint. Furthermore, this is equivalent to Chapel et al. (2020)'s notion of partial alignment up to scaling (Corollary 3.14). As such, to put it informally, what reversible Gromov-Monge is to GW, RRGM is to PGW (Chhoa et al. (2024), Definition 2.7). We also note that the distinct operations for relaxing couplings (based on 'mass trimming' (RRGM) and 'mass addition' (PGW)) essentially yield equivalent robustification, given Nietert et al. (2022) (Proposition 1). They also go on to prove the existence of such relaxed optimal couplings (Chhoa et al. (2024), Theorem 2.13), which in turn further ratifies our formulation of partial robustification along transformations. Furthermore, we mention Gong et al. (2025), which adopts Chapel et al.'s notion of partial alignment. For a detailed discussion on the distinction between the two notions, we refer to Section 3.3.1 in Chhoa et al. (2024).*

### Experiment: Image-to-image translation with noise

I2I translation, the task of learning a mapping between visual domains while preserving semantic content, was popularized through paired (Isola et al., 2017) and unpaired (Zhu et al., 2017) adversarial architectures. The unpaired setting, epitomized by CycleGAN, enforces cycle-consistency as a proxy for structural preservation, but remains susceptible to degenerate solutions under large domain gaps or corrupted source data. A crucial theoretical connection was established by Zhang et al. (2022), who showed that I2I translation architectures such as CycleGAN are in fact special cases of Gromov-Monge-like distances. This connection is significant as it directly exposes the vulnerability of such architectures to contamination in the source measure: outlier or corrupted source images destabilize the learned coupling, leading to semantically incoherent translations. While this link has been formally established, existing literature does not provide a pathway to accurate generation under contaminated source data.

We test RRGM in a noisy MNIST↔USPS domain translation experiment. The contamination regime for the experiment remains the same as that in RUNIT. Maintaining $\alpha = 0.4$, we randomly select pixel locations following a Gaussian law and set their values to 1.0 (bright white) for all channels, adding visible outlier points to the image. Since handwritten digit images have all information regarding the numerical in the shape of white pixels, this contamination becomes quite challenging for an I2I model. For example, CycleGAN (Zhu et al., 2017) performs poorly despite employing a cycle-consistency component and generative losses in both directions. In contrast, RRGM (based on (22) with $\epsilon = 0.5$) under $\lambda_i = 0.2$; $i = 1, 2$ generates significantly sharper and denoised samples (Figure 6a). For a fair comparison, we also present both clean and noisy samples to the discriminators in CycleGAN. Even if the discriminators are shown noisy observations only, the generation quality of RRGM surpasses that of CycleGAN. As a reference, we maintain a similar parameter selection for RGM (Hur et al., 2024), which deploys an additional MMD loss to impose the binding

constraint. However, in the absence of dedicated critic modules, it lags behind. Our model outperforms both techniques by a significant margin, in terms of both quantitative and qualitative measures.

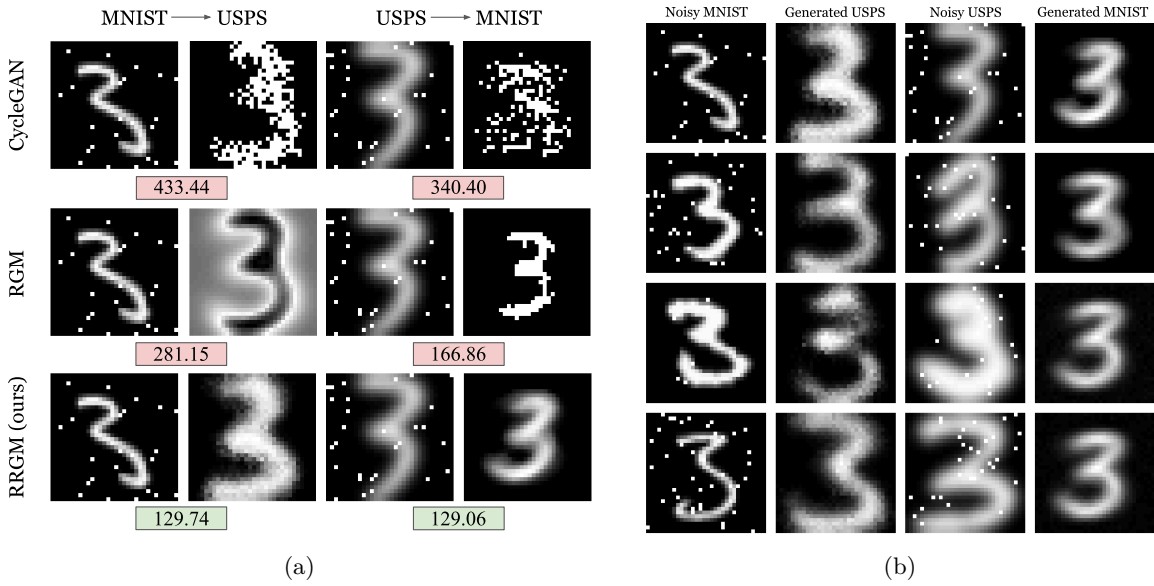

Figure 6: (a) FID scores corresponding to robust cross-domain generations between USPS and MNIST data under Gaussian contamination, using CycleGAN, RGM, and RRGM (ours). (b) Denoised generated samples using RRGM in both domains.

## 5   Discussion

We explore three major possibilities for the robustification problem in a GW setup. Drawing from classical techniques in robust statistics, we propose novel GW surrogate distances TGW (and HGW) and LRGW that limit contamination due to outliers. We study their interrelation and their respective dependence on the truncation parameters. For TGW, we comment on its population-level robust guarantees and the resilience it offers to underlying distributions. Based on a data-dependent parameter selection scheme, we present a working algorithm to solve the HGW distance, which exhibits superior protection against outliers compared to existing methods in shape-matching tasks. On the other hand, solving LRGW-type measures boils down to calculating an OT loss between trimmed samples from the underlying distributions. It also hints at the effective level of trimming necessary ($\lambda$) given a sample problem. To point out an immediate limitation of the two approaches, both TGW and LRGW may suffer 'removable' degeneracies if the truncation parameters = 0. In practice, however, this barely depreciates the applicability of the measures since the thresholding is anyway non-zero. Furthermore, we generalize our notion to probabilistic mm spaces, which allows one to define LR alignment between mixtures of distributions of distinct dimensions. Besides transport-based metrics, the formulation equivalently allows for a novel proposition in establishing robustness, the truncated Lévy-Prokhorov metric. We extend the setup to introduce robust image translation networks that surpass existing benchmarks. We also propose in RRGM a cross-domain transform sampling framework that is robust to outliers. It promotes robust concentration and uniform deviation bounds, besides significantly improving image quality in I2I translations.

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

# A    Technical details and proofs

## Proof of Proposition 1

*(i)* First, let us prove the triangle inequality for the 'norm' $\|\cdot\|_{\mathcal{T}_p}$ (not a norm following the formal definition). Assume $f, g \in L^p(\mu)$, where $\mu \in \mathcal{P}(\mathcal{X})$. Now,

$$
\begin{aligned}
\|f + g\|^p_{\mathcal{T}_p(\mu)} &= \int_{\mathcal{X}} \mathcal{T}_p(f(x) + g(x)) d\mu(x) \\
&= \int_{\mathcal{X}} \left(\mathcal{T}_p(f + g)^{\frac{1}{p}}\right)\left(\mathcal{T}_p(f + g)^{\frac{p-1}{p}}\right) d\mu \\
&\leq \int_{\mathcal{X}} \left(\mathcal{T}_p^{\frac{1}{p}}(f) + \mathcal{T}_p^{\frac{1}{p}}(g)\right)\left(\mathcal{T}_p(f + g)^{\frac{p-1}{p}}\right) d\mu \qquad (23)\\
&= \int_{\mathcal{X}} \mathcal{T}_p^{\frac{1}{p}}(f)\left(\mathcal{T}_p(f + g)^{\frac{p-1}{p}}\right) d\mu + \int_{\mathcal{X}} \mathcal{T}_p^{\frac{1}{p}}(g)\left(\mathcal{T}_p(f + g)^{\frac{p-1}{p}}\right) d\mu \\
&\leq \left[\left(\int_{\mathcal{X}} \mathcal{T}_p(f) d\mu\right)^{\frac{1}{p}} + \left(\int_{\mathcal{X}} \mathcal{T}_p(g) d\mu\right)^{\frac{1}{p}}\right]\left(\int_{\mathcal{X}} \mathcal{T}_p(f + g) d\mu\right)^{\frac{p-1}{p}} \qquad (24)\\
&= \left(\|f\|_{\mathcal{T}_p(\mu)} + \|g\|_{\mathcal{T}_p(\mu)}\right)\|f + g\|^{p-1}_{\mathcal{T}_p(\mu)},
\end{aligned}
$$

where inequality (23) is due to the subadditivity of $\mathcal{T}_p^{\frac{1}{p}}$ (Musco et al. (2021), Lemma C.12) and Hölder inequality implies (24). In the process, we assume that the norm itself is not 0.

Given arbitrary $\varepsilon > 0$, one can obtain feasible optimal couplings $\pi_{XZ} \in \Pi(\mu_X, \mu_Z)$ and $\pi_{ZY} \in \Pi(\mu_Z, \mu_Y)$ that satisfy the following

$$
\frac{1}{2}\|\Lambda_{X,Z}\|_{\mathcal{T}_p(\pi_{XZ} \otimes \pi_{XZ})} + \frac{1}{2}\|\Lambda_{Z,Y}\|_{\mathcal{T}_p(\pi_{ZY} \otimes \pi_{ZY})} = d_{\mathrm{TGW}}(Z, Y) + d_{\mathrm{TGW}}(X, Z) + 2\varepsilon. \qquad (25)
$$

The Gluing Lemma ensures the existence of $\pi \in \mathcal{P}(\mathcal{X} \times \mathcal{Y} \times \mathcal{Z})$ with marginals $\pi_{XZ}$ on $\mathcal{X} \times \mathcal{Z}$ and $\pi_{ZY}$ on $\mathcal{Z} \times \mathcal{Y}$. Also, let $\pi_{XY}$ be the marginal of $\pi$ on $\mathcal{X} \times \mathcal{Y}$. Observe that, given $x, x' \in \mathcal{X}$, $y, y' \in \mathcal{Y}$ and $z, z' \in \mathcal{Z}$, the triangle inequality of $\Lambda_1$ implies

$$
\Lambda_1(d_X(x, x'), d_Y(y, y')) \leq \Lambda_1(d_X(x, x'), d_Z(z, z')) + \Lambda_1(d_Z(z, z'), d_Y(y, y'))
$$

$(\pi \otimes \pi)$-a.e. Now,

$$
\begin{aligned}
d_{\mathrm{TGW}}(X, Y) &\leq \frac{1}{2}\|\Lambda_{X,Y}\|_{\mathcal{T}_p(\pi_{XY} \otimes \pi_{XY})} = \frac{1}{2}\|\Lambda_{X,Y}\|_{\mathcal{T}_p(\pi \otimes \pi)} \\
&\leq \frac{1}{2}\|\Lambda_{X,Z} + \Lambda_{Z,Y}\|_{\mathcal{T}_p(\pi \otimes \pi)} \\
&\leq \frac{1}{2}\|\Lambda_{X,Z}\|_{\mathcal{T}_p(\pi \otimes \pi)} + \frac{1}{2}\|\Lambda_{Z,Y}\|_{\mathcal{T}_p(\pi \otimes \pi)} \qquad (26)\\
&= \frac{1}{2}\|\Lambda_{X,Z}\|_{\mathcal{T}_p(\pi_{XZ} \otimes \pi_{XZ})} + \frac{1}{2}\|\Lambda_{Z,Y}\|_{\mathcal{T}_p(\pi_{ZY} \otimes \pi_{ZY})} \\
&= d_{\mathrm{TGW}}(Z, Y) + d_{\mathrm{TGW}}(X, Z) + 2\varepsilon,
\end{aligned}
$$

where (26) is due to the triangle inequality of the Tukey norm. Since the choice of $\varepsilon > 0$ is arbitrary, this completes the proof.

*(ii)* The proof of this part follows from the monotonicity of $L^p$ norms. Observe that

$$
\|\Lambda_{X,Y}\|_{\mathcal{T}_p(\gamma \otimes \gamma)} = \left\|\mathcal{T}_p^{\frac{1}{p}}(\Lambda_{X,Y})\right\|_{L^p(\gamma \otimes \gamma)} \leq \left\|\mathcal{T}_{p'}^{\frac{1}{p'}}(\Lambda_{X,Y})\right\|_{L^{p'}(\gamma \otimes \gamma)} = \|\Lambda_{X,Y}\|_{\mathcal{T}_{p'}(\gamma \otimes \gamma)}.
$$

*(iii)* The definition of the Tukey loss implies that the corresponding distance is non-decreasing in $\tau$.

**Proof of Theorem 1**

Triangle inequality of $d_X$ implies that given $(x, y), (x', y') \sim \mu' \otimes \nu$

$$2\Lambda_1(d_X, d_X) = \left| d_X(x, x') - d_X(y, y') \right| \leq d_X(x, y) + d_X(x', y').$$

Thus, for a coupling $\pi \in \Pi(\mu', \nu)$, the triangle inequality of the norm $\|\cdot\|_{\mathcal{T}_p}$ implies

$$\|2\Lambda_1\|_{\mathcal{T}_p(\pi \otimes \pi)} \leq 2\|d_X\|_{\mathcal{T}_p(\pi)}. \tag{27}$$

Now,

$$\begin{aligned}
\inf_{\pi \in \Pi} \|d_X\|_{\mathcal{T}_p(\pi)} = \inf_{\pi \in \Pi} \left( \int \mathcal{T}_p(d_X(x, y)) d\pi \right)^{\frac{1}{p}} &= W_{\mathcal{T}_p}(\mu', \nu) \\
&\leq W_{\mathcal{T}_p}((1 - \epsilon)\mu + \epsilon\mu_c, \mu) + W_{\mathcal{T}_p}(\mu, \nu) \tag{28} \\
&\leq W_{\mathcal{T}_p}(\epsilon\mu, \epsilon\mu_c) + W_{\mathcal{T}_p}(\mu, \nu) \tag{29} \\
&\leq \epsilon^{\frac{1}{p}} W_{\mathcal{T}_p}(\mu, \mu_c) + W_{\mathcal{T}_p}(\mu, \nu) \\
&\leq \tau\epsilon^{\frac{1}{p}} + W_{\mathcal{T}_p}(\mu, \nu), \tag{30}
\end{aligned}$$

where (29) is due to the fact that for any $\alpha, \beta \in \mathcal{P}(\mathcal{X})$, we have $\mathrm{OT}_{d_X}(\alpha, \beta) \leq \mathrm{OT}_{d_X}(\alpha - \alpha \wedge \beta, \beta - \alpha \wedge \beta)$ (see Nietert et al. (2022), Proof of Lemma 3). The triangle inequality of $W_{\mathcal{T}_p}$ (28) follows a similar proof to that of the $p$-Wasserstein distance, which we outline as follows. Let $\mu_1, \mu_2, \mu_3 \in \mathcal{P}(\mathcal{X})$, and let $\pi_{12} \in \Pi(\mu_1, \mu_2)$, $\pi_{23} \in \Pi(\mu_2, \mu_3)$ be optimal transport plans. By the Gluing Lemma, there exist a probability measure $\pi \in \mathcal{P}(\mathcal{X} \times \mathcal{X} \times \mathcal{X})$ with marginals $\pi_{12}, \pi_{23}$ and $\pi_{13}$ on $\mathcal{X} \times \mathcal{X}$. As such,

$$\begin{aligned}
W_{\mathcal{T}_p}(\mu_1, \mu_3) \leq \|d_X(x_1, x_3)\|_{\mathcal{T}_p(\pi_{13})} = \|d_X(x_1, x_3)\|_{\mathcal{T}_p(\pi)} \\
\leq \|d_X(x_1, x_2)\|_{\mathcal{T}_p(\pi)} + \|d_X(x_2, x_3)\|_{\mathcal{T}_p(\pi)} \\
= \|d_X(x_1, x_2)\|_{\mathcal{T}_p(\pi_{12})} + \|d_X(x_2, x_3)\|_{\mathcal{T}_p(\pi_{23})} \\
= W_{\mathcal{T}_p}(\mu_1, \mu_2) + W_{\mathcal{T}_p}(\mu_2, \mu_3).
\end{aligned}$$

Inequality (30) is due to the trivial upper bound of the Tukey function. This, along with the previous observation, completes the proof.

**Proof of Corollary 2**

For any $\tilde{\mu} \leq \frac{1}{1-\varepsilon}\mu$, by choosing an appropriate $\beta \in \mathcal{P}(\mathcal{X})$ we can write $\mu = (1 - \varepsilon)\tilde{\mu} + \varepsilon\beta$. Now,

$$\begin{aligned}
W_{\mathcal{T}_p}(\mu, \tilde{\mu}) \leq \varepsilon^{\frac{1}{p}} (W_{\mathcal{T}_p}(\beta, \delta_{x_0}) + W_{\mathcal{T}_p}(\delta_{x_0}, \tilde{\mu})) \\
= \varepsilon^{\frac{1}{p}} \left[ \|d_X(Y, x_0)\|_{\mathcal{T}_p(\beta)} + \|d_X(Y, x_0)\|_{\mathcal{T}_p(\tilde{\mu})} \right] \\
\leq 2\varepsilon^{\frac{1}{p}} \sup_{\alpha \in \mathcal{P}(\mathcal{X}), \alpha \leq \frac{1}{1-\varepsilon'}\mu} \|d_X(Y, x_0)\|_{\mathcal{T}_p(\alpha)}, \tag{31}
\end{aligned}$$

where $\varepsilon' := \varepsilon \vee (1 - \varepsilon)$. Given that the expectation is finite, the definition of $\mathcal{T}_p$ implies $\mathbb{E}_\alpha[\mathcal{T}_p(d_X(Y, x_0))] \leq \tau^p$ for all such $\alpha$ as in (31). Moreover,

$$\begin{aligned}
\mathbb{E}_\alpha[\mathcal{T}_p(d_X(Y, x_0))] \leq \left| \mathbb{E}_\alpha[\mathcal{T}_p(d_X(Y, x_0))] - \mathbb{E}_\mu[\mathcal{T}_p(d_X(Z, x_0))] \right| + \mathbb{E}_\mu[\mathcal{T}_p(d_X(Z, x_0))] \\
\leq \left( 1 \vee \frac{1 - \varepsilon}{\varepsilon} \right) \rho + (\sigma^p \wedge \tau^p), \tag{32}
\end{aligned}$$

where the first inequality is due to the triangle inequality. As such, combining inequality (32) with the bound (31) yields,

$$W_{\mathcal{T}_p}(\mu, \tilde{\mu}) \leq 2 \left( (\rho^{\frac{1}{p}} + \varepsilon^{\frac{1}{p}}(\sigma \wedge \tau)) \wedge \varepsilon^{\frac{1}{p}} \tau \right).$$

## A.1 Breakdown Point (BP) analysis of TGW

It is only fairly recently (Avella Medina & González-Sanz, 2026; Paindaveine & Passeggeri, 2024) that we have come to analyze the Donoho-Huber BP (Donoho & Huber, 1983) of OT maps, whose findings can be generalized for regular convex cost functions (Gonzalez-Sanz & Medina, 2026). Earlier, it was only conjectured that the BP of $W_p^\epsilon$ (as in (12)) can, in principle, be "pushed to its information-theoretic limit of 1/2" (see Nietert et al. (2022), Remark 2). However, extending the truncation technique to GW may render the resulting $GW^\epsilon$ non-informative if $\epsilon \geq 1/3$ (see Gong et al. (2025), Remark 2.5). It can supposedly be improved based on projection estimates of perturbed measures; however, we could only find a limited explanation of the improvement and potential limits. In our work, in contrast, we have already shown the upper bound on the realized bias of TGW while recovering the unperturbed GW under the $\mathcal{O} \cup \mathcal{I}$ framework (see Remark 3). Here, we resort to the more general corruption regime of Huber for BP analysis as follows.

Let $T$ be an estimator based on samples $\{(x_1, y_1), \ldots, (x_n, y_n)\}$ (say, $Z$). 'Corrupted samples' $Z'$ are obtained by replacing any $k$ of the $n$ data points by arbitrary values. For simplicity and to maintain consistency with the standard setup (Avella Medina & González-Sanz, 2026), we only allow perturbations on $\{x_1, \ldots, x_n\}$ (the source sample, with target fixed). Let us define the *maximum bias* caused by $k$ replacements:

$$\text{bias}(k; T, Z) = \sup_{Z'} \left\| T(Z') - T(Z) \right\|,$$

where the supremum is over all such corrupted $Z'$. Now, given that the Tukey loss is bounded, regardless of how many points in $Z$ are replaced and regardless of where they are placed, the corresponding TGW never diverges. This is in contrast with standard GW, where replacing even $k = 1$ point with an arbitrary magnitude forces its transport and diverges the cost. Hence, we characterize how large the bias gets as a function of $k$. Let us consider the contaminated coupling $\pi' = (1 - \frac{k}{n})\pi + \frac{k}{n}\pi_z \in \Pi(\mu', \nu)$, where $\mu'$ is the empirical law corresponding to the corrupted samples $\{x_i\}$ and $\pi \in \Pi(\mu, \nu)$. Expanding the double sum into three parts (according to outlier-outlier, inlier-inlier and inlier-outlier), we get

$$\sup_z \text{TGW}(\mu_c, \nu) = \left(1 - \frac{k}{n}\right)^2 \text{TGW}(\mu, \nu) + \frac{2k}{n}\left(1 - \frac{k}{n}\right)\tau^p + \left(\frac{k}{n}\right)^2 \tau^p.$$

As such,

$$\text{bias}(k; T_{\text{TGW}}, Z) \leq \tau^p \cdot \frac{k(2n - k)}{n^2}, \tag{33}$$

which is increasing in $k$, reaches $\tau^p$ only at $k = n$, and equals 0 at $k = 0$. This is a rather graceful degradation curve, in contrast to the step-function breakdown of GW. Note that the equality holds since we take sup over arbitrary outlying observations. Moreover, it is perhaps irrelevant in the context of BP analysis to consider outlying observations that incur distortions $\leq \tau$ (in outlier-outlier or outlier-inlier interactions) as the effective contribution towards the bias is bounded.

Since the set $\{\frac{k}{n} : \text{bias}(k; T_{\text{TGW}}, Z) = \infty\}$ is empty, we define a more refined notion.

**Definition 3** ($\delta$-Breakdown Point). *The $\delta$-breakdown point is the smallest contamination fraction that can push the bias beyond $\delta\tau^p$:*

$$\varepsilon_n^\delta(T_{TGW}, Z) := \min\left\{\frac{k}{n} \ : \ \text{bias}(k; T_{TGW}, Z) > \delta\,\tau^p\right\}.$$

From (33), we observe (by solving $1 - \left(1 - \frac{k}{n}\right)^2 > \delta$) that $\varepsilon_n^\delta(T_{\text{TGW}}, Z) = 1 - \sqrt{1 - \delta}$. As such, for $\delta = 1/2$: $\varepsilon^{1/2} = 1 - 1/\sqrt{2} \approx 0.293$.

Perhaps the immediate scrutiny of TGW as a robust estimator would be to calculate its Influence function (IF)

$$\text{IF}(z; T, \mu) = \lim_{\epsilon \to 0^+} \frac{T(\mu_\epsilon, \nu) - T(\mu, \nu)}{\epsilon} = V'(0),$$

where $V(\epsilon) = T(\mu_\epsilon, \nu)$ such that $\mu_\epsilon$ is the Huber contaminated counterpart of $\mu$, for $\epsilon \in [0, 1]$. Although the derivative of the Tukey norm (kernel) is uniformly bounded by $p\tau^{p-1}$, note that one requires the dual potentials of TGW to satisfy a similar breakdown to that of GW in order to calculate IF. In the paper, we do not show a duality for TGW, which only appears as an upper bound to LRGW. It is only apparent that the TGW IF is expected to saturate within a finite radius. We believe the problem might benefit from independent future scrutiny.

## A.2 Existence of optimal couplings in LRGW

Given Polish mm spaces $(\mathcal{X}, d_X, \mu)$, $(\mathcal{Y}, d_Y, \nu)$ define the locally robust $p$-distortion realized by a coupling $\pi \in \Pi(\mu, \nu)$ as, $p \in [0, \infty]$

$$J_p^\lambda(\pi) := \left\| l_\lambda(d_X) - l_\lambda(d_Y) \right\|_{L^p(\pi \otimes \pi)},$$

where $\lambda > 0$. Then,

**Lemma 7.** *There exists a coupling $\pi^\lambda \in \Pi(\mu, \nu)$ such that, $(p, \lambda)$-LRGW $= J_p^\lambda(\pi)$.*

The lemma can be proved by extending Corollary 10.1 of Mémoli (2011) for the mm spaces with modified metrics $l_\lambda(d_X)$ and $l_\lambda(d_Y)$. We give a version of the proof for completeness.

*Proof.* First, observe that the set of couplings $\Pi(\mu, \nu)$ is sequentially compact in $\mathcal{P}(\mathcal{X} \times \mathcal{Y})$ (Sturm (2023), Lemma 1.2). Hence, for $1 \leq p < \infty$, it suffices to show that $J_p^\lambda(\pi)$ is continuous on $\Pi(\mu, \nu)$. Let us choose the metric $d^\lambda((x, y), (x', y')) = l_\lambda(d_X(x, x')) + l_\lambda(d_Y(y, y'))$ mapping $\mathcal{X} \times \mathcal{Y}$ to $[0, 2\lambda]$. Given $((x_i, y_i), (x'_i, y'_i)) \sim \pi \otimes \pi$ for $i = 1, 2$, observe that

$$\left| \left| l_\lambda(d_X(x_1, x'_1)) - l_\lambda(d_Y(y_1, y'_1)) \right| - \left| l_\lambda(d_X(x_2, x'_2)) - l_\lambda(d_Y(y_2, y'_2)) \right| \right|$$
$$\leq \left| l_\lambda(d_X(x_1, x'_1)) - l_\lambda(d_X(x_2, x'_2)) + l_\lambda(d_Y(y_2, y'_2)) - l_\lambda(d_Y(y_1, y'_1)) \right|$$
$$\leq \left| l_\lambda(d_X(x_1, x'_1)) - l_\lambda(d_X(x_2, x'_2)) \right| + \left| l_\lambda(d_Y(y_2, y'_2)) - l_\lambda(d_Y(y_1, y'_1)) \right|$$
$$\leq \left[ l_\lambda(d_X(x_1, x_2)) + l_\lambda(d_Y(y_1, y_2)) \right] + \left[ l_\lambda(d_X(x'_1, x'_2)) + l_\lambda(d_Y(y'_1, y'_2)) \right] \tag{34}$$
$$= d^\lambda((x_1, y_1), (x_2, y_2)) + d^\lambda((x'_1, y'_1), (x'_2, y'_2)), \tag{35}$$

which implies the Lipschitz continuity of $f_1 := 2\Lambda_1(l_\lambda(d_X), l_\lambda(d_Y))$ for the metric given by (35). The step (34) follows from the triangle inequalities of $l_\lambda(d_X)$ and $l_\lambda(d_Y)$. Now, consider a function $f_2 : [0, 2\lambda] \to \mathbb{R}_+$ mapping $t \mapsto t^p$, which in turn implies that $f_2 \circ f_1$ is Lipschitz with constant $\leq p(2\lambda)^{p-1}$. Thus, given a sequence $\{\pi_n\}_{n \in \mathbb{N}} \subset \mathcal{P}(\mathcal{X} \times \mathcal{Y})$ such that $\pi_n \xrightarrow{w} \pi$, we have

$$\left| \int \underbrace{\int f_2 \circ f_1 \, \pi_n(d(x, y))}_{:= f_{\pi_n}(x', y')} \pi_n(d(x', y')) - \int \underbrace{\int f_2 \circ f_1 \, \pi(d(x, y))}_{:= f_\pi(x', y')} \pi(d(x', y')) \right|$$
$$\leq \left| \int (f_{\pi_n} - f_\pi) \, \pi_n(d(x', y')) \right| + \left| \int f_\pi \, \pi_n(d(x', y')) - f_\pi \, \pi(d(x', y')) \right|$$
$$\leq \max_{(x', y')} |f_{\pi_n} - f_\pi| + \left| \int f_\pi \, d\pi_n - f_\pi \, d\pi \right|,$$

where the first term on the right-hand side vanishes as $n \to \infty$ due to the uniformly convergent $f_{\pi_n} \to f_\pi$ (based on the point-wise convergence of $f_{\pi_n}$ and the Lipschitz continuity of $f_2 \circ f_1$). The weak convergence of $\pi_n$ ensures that the second term also vanishes. As such, $J_p^\lambda(\pi_n) \to J_p^\lambda(\pi)$ as $n \to \infty$. Hence, the proof.

To show the result for $p = \infty$, first observe that given $\pi$, the sequence $\{J_l^\lambda(\pi)\}$ is non-decreasing in $1 \leq l \leq \infty$ (using Jensen's inequality) and $\lim_{p \to \infty} J_p^\lambda(\pi) \to J_\infty^\lambda(\pi) = \sup\{J_p^\lambda : 1 \leq p < \infty\}$. As such, $J_\infty^\lambda(\pi)$ is lower semi-continuous. Now, invoking the compactness argument proves the infimum is achieved in $\Pi(\mu, \nu)$. $\qquad\square$

**Proof of Proposition 2**

*(i)* Given $\lambda' > \lambda \geq 0$, observe that

$$\left| l_\lambda(d_X) - l_\lambda(d_Y) \right| \leq \left| l_{\lambda'}(d_X) - l_{\lambda'}(d_Y) \right|,$$

a.s. $(\mu \otimes \nu)^{\otimes 2}$, which proves the first claim. Also, for all $a \in \mathbb{R}$ the following equality holds

$$l_{\lambda'}(a) - l_\lambda(a) = \min\{\lambda', \max\{a, \lambda\}\} - \lambda = l_{\lambda'}(\bar{l}_\lambda(a)) - \lambda.$$

We point out that, even a stronger equality holds in general almost surely

$$\left| l_{\lambda'}(d_X) - l_{\lambda'}(d_Y) \right| = \left| l_\lambda(d_X) - l_\lambda(d_Y) \right| + \left| l_{\lambda'}(\bar{l}_\lambda(d_X)) - l_{\lambda'}(\bar{l}_\lambda(d_Y)) \right|. \tag{36}$$

Now, given any $\pi$ that solves the $(p, \lambda)$-LRGW$(\mu, \nu)$ problem

$$
\begin{aligned}
(p, \lambda')\text{-LRGW}(\mu, \nu) - (p, \lambda)\text{-LRGW}(\mu, \nu) &\leq \left\| l_{\lambda'}(d_X) - l_{\lambda'}(d_Y) \right\|_{L^p(\pi \otimes \pi)} - (p, \lambda)\text{-LRGW}(\mu, \nu) \\
&\leq \left\| l_{\lambda'}(\bar{l}_\lambda(d_X)) - l_{\lambda'}(\bar{l}_\lambda(d_Y)) \right\|_{L^p(\pi \otimes \pi)},
\end{aligned}
$$

where the second inequality is due to (36) and the the triangle inequality of $L^p$ norms. The proof follows since the choice of $\pi$ is arbitrary in $\Pi^\lambda$. We also present an upper bound that depends on the spaces' regulated $p$-diameters. First, $\forall a \in \mathbb{R}$ let us write

$$\left| l_{\lambda+\lambda'}(a) - l_{\lambda'}(a) \right| = \bar{l}_0(l_\lambda(a - \lambda')).$$

Now, for any optimal coupling $\pi \in \Pi(\mu, \nu)$ for the $(p, \lambda')$-LRGW problem, we get

$$
\begin{aligned}
(p, \lambda + \lambda')\text{-LRGW}(\mu, \nu) &\leq \left\| l_{\lambda+\lambda'}(d_X) - l_{\lambda+\lambda'}(d_Y) \right\|_{L^p(\pi \otimes \pi)} \\
&\leq \left\| l_{\lambda+\lambda'}(d_X) - l_{\lambda'}(d_X) \right\|_{L^p(\mu \otimes \mu)} + \left\| l_{\lambda'}(d_X) - l_{\lambda'}(d_Y) \right\|_{L^p(\pi \otimes \pi)} \\
&\quad + \left\| l_{\lambda'}(d_Y) - l_{\lambda+\lambda'}(d_Y) \right\|_{L^p(\nu \otimes \nu)} \\
&\leq \left\| \bar{l}_0(l_\lambda(d_X(x, x') - \lambda')) \right\|_{L^p(\mu \otimes \mu)} + (p, \lambda')\text{-LRGW}(\mu, \nu) \\
&\quad + \left\| \bar{l}_0(l_\lambda(d_Y(y, y') - \lambda')) \right\|_{L^p(\nu \otimes \nu)} \\
&= d^{\lambda, \lambda'}_{\mathcal{X}, p} + d^{\lambda, \lambda'}_{\mathcal{Y}, p} + (p, \lambda')\text{-LRGW}(\mu, \nu), \tag{37}
\end{aligned}
$$

where

$$
\begin{aligned}
\left\| l_\lambda(d_X) \right\|_{L^p(\mu \otimes \mu)} &\geq d^{\lambda, \lambda'}_{\mathcal{X}, p} \\
&= \left\| \bar{l}_0(l_\lambda(d_X(x, x') - \lambda')) \right\|_{L^p(\mu \otimes \mu)} \\
&= \left\| \bar{l}_{\lambda'}(l_{\lambda+\lambda'}(d_X(x, x'))) - \lambda' \right\|_{L^p(\mu \otimes \mu)} \\
&\geq \bar{l}_{\lambda'}\left( \left\| l_{\lambda+\lambda'}(d_X(x, x')) \right\|_{L^p(\mu \otimes \mu)} \right) - \lambda' \\
&= \bar{l}_0\left( \left\| l_{\lambda+\lambda'}(d_X) \right\|_{L^p(\mu \otimes \mu)} - \lambda' \right).
\end{aligned}
$$

As such, $(p, \lambda + \lambda')$-LRGW$(\mu, \nu) - (p, \lambda')$-LRGW$(\mu, \nu) \leq 2 \max_{\mathcal{Z} \in (\mathcal{X}, \mathcal{Y})} d^{\lambda, \lambda'}_{\mathcal{Z}, p}$.

*(ii)* Choosing $\lambda' = \infty$ in the first argument of *(i)* proves the result.

**Proof of Theorem 4**

The proof follows the decomposition of the GW cost due to Zhang et al. (2024). Recall the decomposition of the squared LRIGW cost: $d^2_{\text{LRIGW}}(\mu, \nu) = F_1 + F_2$, where $F_2(\mu, \nu; \lambda) = \inf_{\pi \in \Pi(\mu, \nu)} -2 \int l_\lambda(\langle x, x' \rangle) \, l_\lambda(\langle y, y' \rangle) d\pi \otimes \pi(x, y, x', y')$.

Now, for all $x, x' \sim \mu \in \mathcal{P}_4(\mathbb{R}^d_{\geq 0})$

$$l_\lambda(\langle x, x' \rangle) = l_\lambda\Big(\sum_{i=1}^d x_i x_i'\Big) \geq \sum_{i=1}^d l_{\frac{\lambda}{d}}(x_i x_i') \geq \sum_{i=1}^d l_{\sqrt{\frac{\lambda}{d}}}(x_i) l_{\sqrt{\frac{\lambda}{d}}}(x_i') = \Big\langle l_{\sqrt{\frac{\lambda}{d}}}(x), l_{\sqrt{\frac{\lambda}{d}}}(x') \Big\rangle.$$

In the last step, the function $l_\lambda$ applies componentwise. Similarly, the inequality $l_\lambda(\langle y, y' \rangle) \geq \sum_{j=1}^{d'} l_{\sqrt{\lambda/d'}}(y_j) l_{\sqrt{\lambda/d'}}(y_j')$ holds for all $y, y' \sim \nu \in \mathcal{P}_4(\mathbb{R}^{d'}_{\geq 0})$. Let us generalize by defining $M_{\mu,\nu}^{(\lambda,\lambda')} := \sqrt{M_2(\mu;\lambda)M_2(\nu;\lambda')}$, where $M_2(\rho;\lambda) = \int \big\|l_\lambda(x)\big\|^2 d\rho(x)$ for any $\rho$. Also, let $\mathcal{D}_{M_{\mu,\nu}^{(\lambda,\lambda')}} := [0, M_{\mu,\nu}^{(\lambda,\lambda')}/2]^{d \times d'}$. Hence,

$$F_2 \leq \inf_{\pi \in \Pi(\mu,\nu)} -2 \sum_{\substack{1 \leq i \leq d \\ 1 \leq j \leq d'}} \Big( \int l_{\sqrt{\frac{\lambda}{d}}}(x_i) l_{\sqrt{\frac{\lambda}{d'}}}(y_j) d\pi(x,y) \Big)^2 \tag{38}$$

$$= \inf_{\pi \in \Pi(\mu,\nu)} \sum_{\substack{1 \leq i \leq d \\ 1 \leq j \leq d'}} \inf_{0 \leq a_{ij} \leq \frac{M_{\mu,\nu}^{\bar\lambda}}{2}} 8 \Big( a_{ij}^2 - \int a_{ij} l_{\sqrt{\frac{\lambda}{d}}}(x_i) l_{\sqrt{\frac{\lambda}{d'}}}(y_j) d\pi(x,y) \Big) \tag{39}$$

$$= \inf_{\mathbf{A} \in \mathcal{D}_{M_{\mu,\nu}^{\bar\lambda}}} \inf_{\pi \in \Pi(\mu,\nu)} \sum_{\substack{1 \leq i \leq d \\ 1 \leq j \leq d'}} 8 \Big( a_{ij}^2 - \int a_{ij} l_{\sqrt{\frac{\lambda}{d}}}(x_i) l_{\sqrt{\frac{\lambda}{d'}}}(y_j) d\pi(x,y) \Big)$$

$$= \inf_{\mathbf{A} \in \mathcal{D}_{M_{\mu,\nu}^{\bar\lambda}}} 8\|\mathbf{A}\|_F^2 + \inf_{\pi \in \Pi(\mu,\nu)} \int c_{\mathbf{A}}^{\bar\lambda}(x,y) d\pi(x,y),$$

where $\bar\lambda = (\sqrt{\lambda/d}, \sqrt{\lambda/d'})$ and $c_{\mathbf{A}}^{\bar\lambda} : (x,y) \in \mathbb{R}^d_{\geq 0} \times \mathbb{R}^{d'}_{\geq 0} \mapsto -8 l_{\sqrt{\lambda/d}}(x)^T \mathbf{A} l_{\sqrt{\lambda/d'}}(y)$. The optimization in $\mathbf{A}$ can be made unconstrained, however, the optimal $a_{ij}$ in (39) is achieved at $\frac{1}{2} \int l_{\sqrt{\lambda/d}}(x_i) l_{\sqrt{\lambda/d'}}(y_j) d\pi(x,y) \in [0, M_{\mu,\nu}^{\bar\lambda}/2]$ (due to Cauchy-Schwarz inequality), which enables us to restrict the optimization to $\mathcal{D}_{M_{\mu,\nu}^{\bar\lambda}}$.

A simple parametrization can result in uniform $\lambda$-thresholding over the two spaces. Specifically, for the threshold $(d \vee d')\lambda^2$, we achieve the desired upper bound to $F_2$, as in (38), satisfying

$$\bar{F}_2(\mu, \nu; (d \vee d')\lambda^2) = \inf_{\mathbf{A} \in \mathcal{D}_{M_{\mu,\nu}^\lambda}} 8\|\mathbf{A}\|_F^2 + \mathrm{OT}_{c_{\mathbf{A}}^\lambda}(\mu, \nu). \tag{40}$$

Given an optimal coupling $\pi_{\mathbf{A}}^*$ for $\mathrm{OT}_{c_{\mathbf{A}}^\lambda}$, a solution $\mathbf{A}^*$ achieving the infimum in (40) can be expressed as $\mathbf{A}^* = \frac{1}{2} \int l_\lambda(x) l_\lambda(y)^T d\pi_{\mathbf{A}^*}^*(x,y)$. The associated optimal value of the upper bound becomes

$$\bar{F}_2 = -2 \int \Big\langle l_\lambda(x), l_\lambda(x') \Big\rangle \Big\langle l_\lambda(y), l_\lambda(y') \Big\rangle d\pi_{\mathbf{A}^*}^* \otimes \pi_{\mathbf{A}^*}^*(x, y, x', y').$$

### A.3 Relation between $W_p^\epsilon$ and truncated OT

Given $\alpha, \beta \in \mathcal{P}(X)$ such that $X$ is compact, the dual formulation of $W_p^\epsilon$ for $p \in [1, \infty)$ becomes (Nietert et al. (2022), Theorem 2)

$$(1-\epsilon)W_p^\epsilon(\alpha, \beta)^p = \sup_{\phi \in C_b(X)} \int \phi \, d\alpha - \int \phi^c \, d\beta - \epsilon \, \mathrm{Range}(\phi), \tag{41}$$

where $C_b(X) := \{f : X \to \mathbb{R} : f \text{ is continuous}, \|f\|_\infty < \infty\}$ and $\phi^c$ denotes the $c$-transform of $\phi$ w.r.t. the cost $d_X(\cdot, \cdot)^p$. On the other hand, the Kantorovich potential $\phi$ that solves $\mathrm{OT}_{l_\lambda(d_X)}(\alpha, \beta) := \inf_{\pi \in \Pi(\alpha,\beta)} \int_{X \times X} \min\{d_X(x,y), \lambda\}^p d\pi(x,y)$ is a solution to the dual

$$\sup_{\substack{\phi \in C_b(X): \\ \mathrm{Range}(\phi) \leq \lambda}} \int \phi \, d\alpha - \int \phi^c \, d\beta, \tag{42}$$

such that $\phi^c = \phi$ (Ma et al. (2023), Theorem 2.1), $\lambda > 0$. The latter is a constrained formulation of the regularized dual (41). Given any tolerable margin $\lambda < \infty$ on the range of potentials, the solution to (42) satisfies (41).

**Proof of Proposition 3**

*(i)* The result is a direct consequence of the fact $W_p^0 = W_p$.

*(ii)* For $0 \le \epsilon \le \epsilon' \le 1$,

$$d_{\text{LRGW}}(\mu, \nu; \epsilon) = \inf_{\pi \in \Pi(\mu,\nu)} \left\| W_p^\epsilon(x, x') - W_p(y, y') \right\|_{L^p(\pi \otimes \pi)}. \tag{43}$$

Now, for $(x, y), (x', y') \sim \mu \otimes \nu$

$$\left| W_p^\epsilon(x, x') - W_p(y, y') \right| \le \left| W_p^\epsilon(x, x') - W_p^{\epsilon'}(x, x') \right| + \left| W_p^{\epsilon'}(x, x') - W_p(y, y') \right|. \tag{44}$$

Due to the dual form (see, Appendix A.3), given any $\alpha, \beta \in \mathcal{P}(X)$ we may write

$$(1 - \epsilon) W_p^\epsilon(\alpha, \beta)^p = \sup_{\phi \in C_b(X)} \int \phi \, d\alpha - \int \phi^c \, d\beta - \epsilon \, \text{Range}(\phi)$$

$$\le \sup_{\phi \in C_b(X)} \int \phi \, d\alpha - \int \phi^c \, d\beta - \epsilon' \, \text{Range}(\phi) + 2(\epsilon' - \epsilon) \|\phi\|_\infty.$$

Since $\phi \in C_b(X)$, $\exists K > 0$ such that $W_p^\epsilon(\alpha, \beta) - \left(\frac{1-\epsilon'}{1-\epsilon}\right)^{\frac{1}{p}} W_p^{\epsilon'}(\alpha, \beta) \le K \left(\frac{\epsilon'-\epsilon}{1-\epsilon}\right)^{\frac{1}{p}}$. As such,

$$0 \le W_p^\epsilon(x, x') - W_p^{\epsilon'}(x, x')$$

$$\le \left[ \left( \frac{1-\epsilon'}{1-\epsilon} \right)^{\frac{1}{p}} - 1 \right] W_p^{\epsilon'}(x, x') + K \left( \frac{\epsilon' - \epsilon}{1 - \epsilon} \right)^{\frac{1}{p}}$$

$$\le \left[ \left( \frac{1-\epsilon'}{1-\epsilon} \right)^{\frac{1}{p}} - 1 \right] \underline{W}_{p,\mu}^{\epsilon'} + K \left( \frac{\epsilon' - \epsilon}{1 - \epsilon} \right)^{\frac{1}{p}}, \tag{45}$$

where $\underline{W}_{p,\mu}^{\epsilon'} = \inf_{x,x' \sim \mu} W_p^{\epsilon'}(x, x')$, which may only be non-zero given a sample problem. The last inequality, along with (44) and the triangle inequality of $L^p$ norms implies that

$$d_{\text{LRGW}}(\mu, \nu; \epsilon) - d_{\text{LRGW}}(\mu, \nu; \epsilon') \le \left[ \left( \frac{1-\epsilon'}{1-\epsilon} \right)^{\frac{1}{p}} - 1 \right] \underline{W}_{p,\mu}^{\epsilon'} + K \left( \frac{\epsilon' - \epsilon}{1 - \epsilon} \right)^{\frac{1}{p}}.$$

Since (44) holds both ways (in $\epsilon, \epsilon'$), invoking the trivial upper bound to (45) we obtain

$$\left| d_{\text{LRGW}}(\mu, \nu; \epsilon) - d_{\text{LRGW}}(\mu, \nu; \epsilon') \right| \lesssim \left( \frac{\epsilon' - \epsilon}{1 - \epsilon} \right)^{\frac{1}{p}}.$$

**Proof of Proposition 4**

The proof follows as a modification to Huber (1981), Corollary 4.3. Given any $\pi \in \Pi(\alpha, \beta)$, observe that

$$\mathbb{E}_\pi[l_\lambda(d_X(x, y))] \le \varepsilon \, \mathbb{P}(l_\lambda(d_X(x, y)) \le \varepsilon) + \lambda \, \mathbb{P}(l_\lambda(d_X(x, y)) > \varepsilon)$$

$$= \varepsilon + (\lambda - \varepsilon) \, \mathbb{P}(l_\lambda(d_X(x, y)) > \varepsilon).$$

Now, consider $\varepsilon = \hat{\rho}_\lambda(\alpha, \beta)$. As such there exists $\pi$ such that $\pi(\{(x, y) : l_\lambda(d_X(x, y)) > \varepsilon\}) \le \varepsilon$. Thus

$$\mathbb{E}_\pi[l_\lambda(d_X(x, y))] \le \varepsilon + (\lambda - \varepsilon)\varepsilon \le (1 + \lambda)\varepsilon.$$

Taking infimum over all couplings result in $\frac{1}{1+\lambda} W_{1,\lambda} \le \hat{\rho}_\lambda$. To show the upper bound, by using Markov's inequality, we write

$$\mathbb{P}(l_\lambda(d_X(x, y)) > \varepsilon) \le \frac{\mathbb{E}_\pi[l_\lambda(d_X(x, y))]}{\varepsilon},$$

where $\pi \in \Pi(\alpha, \beta)$ is a feasible optimal coupling for $W_{1,\lambda}$. Observe that we can always choose $\varepsilon^2 = W_{1,\lambda}$. As such, $\hat{\rho}_\lambda \le \sqrt{W_{1,\lambda}}$.

**Proof of Theorem 5**

Before proving the first inequality, recall that given two mm spaces $(\mathcal{X}, d_X, \mu)$ and $(\mathcal{Y}, d_Y, \nu)$, a metric $\tilde{d}$ on $\mathcal{X} \sqcup \mathcal{Y}$ (disjoint union) is said to be a coupling of $d_X$ and $d_Y$ if and only if $\tilde{d}(x, x') = d_X(x, x')$ and $\tilde{d}(y, y') = d_Y(y, y')$ hold for all $x, x' \in \mathcal{X}$, $y, y' \in \mathcal{Y}$. Let us denote by $\mathscr{D}(d_X, d_Y)$ the collection of all such couplings.

Now, given $(x, y), (x', y') \sim \mu \otimes \nu$ and $\lambda > 0$, we have $\left| l_\lambda(d_X(x, x')) - l_\lambda(d_Y(y, y')) \right| \leq l_\lambda(\tilde{d}(x, y)) + l_\lambda(\tilde{d}(x', y'))$. Hence, for $p \in [1, \infty)$

$$J_p^\lambda(\pi) := \left\| l_\lambda(d_X) - l_\lambda(d_Y) \right\|_{L^p(\pi \otimes \pi)} \leq 2 \| l_\lambda(\tilde{d}) \|_{L^p(\pi)}$$

hold for all $\pi \in \Pi(\mu, \nu)$. Hence, the inequality. To show the upper bound, we require some additional definitions.

**Definition 4** (Modulus of trimmed mass distribution). *For $\delta \geq 0$, the modulus of $\lambda$-trimmed mass distribution of $\mu$, having full support is defined as*

$$v_\delta^\lambda(\mu) := \inf\{\varepsilon > 0 : \mu(\{x \in \mathcal{X} : \mu(B_X^\lambda(x, \varepsilon)) \leq \delta\}) \leq \varepsilon\},$$

*where $B_X^\lambda(x, \varepsilon) = \{y \in \mathcal{X} : l_\lambda(d_X(x, y)) < \varepsilon\}$ is the open ball of $\lambda$-trimmed radius $\varepsilon > 0$ around $x \in \mathcal{X}$.*

Here, we uniquely consider $\mu$ to be a probability measure. Observe that only when $\varepsilon < \lambda$, we get $B_X^\lambda(x, \varepsilon) = B_X(x, \varepsilon)$: the usual open ball around $x$. Otherwise, the ball becomes the entire $\mathcal{X}$. As such, we only account for the 'thin points' residing in the trimmed support. $v_\delta^\lambda(\mu)$ is essentially equal to $\min\{\lambda, v_\delta(\mu)\}$, where $v_\delta$ is the modulus under the metric $d_X$. This preserves the continuity in the sense that $v_\delta^\lambda(\mu) \xrightarrow{\delta \to 0} 0$ (Greven et al. (2009), Lemma 6.5). The relation also implies that an effective trimming requires $\lambda \leq 1$ for probability measures. The proof for the upper bound requires showing a similar statement to Lemma 10.3 under the altered metrics $l_\lambda(d_X)$ and $l_\lambda(d_Y)$.

**Step 1** *(Construction of $\varepsilon$-nets):* Let $\delta \in (0, \frac{1}{2})$, and $\pi \in \Pi(\mu, \nu)$ such that $J_p^\lambda(\pi) < \delta^5$. Since the altered metric space $(\mathcal{X}, l_\lambda(d_X))$ also contains a maximal $2\varepsilon$-separated net for any $\varepsilon \geq 0$, we have the following statement.

**Lemma 8** (Greven et al. (2009), Lemma 6.9). *Given $\delta > 0$ and $v_\delta^\lambda(\mu) < \varepsilon$, there exist points $\{x_i\}_{i=1}^N \in \mathcal{X}$ with $N \leq \lfloor \frac{1}{\delta} \rfloor$ such that $\mu(B_X^\lambda(x_i, \varepsilon)) > \delta$, and $\mu(\bigcup_{i=1}^N B_X^\lambda(x, 2\varepsilon)) > 1 - \varepsilon$, $\forall\, i = 1, \cdots, N$. Also, for all $i \neq j = 1, \cdots, N$, $l_\lambda(d_X(x_i, x_j)) > \varepsilon$.*

Observe that, since the effective range of permissible $\varepsilon$ remains $(0, \lambda)$, a maximal net of $(\mathcal{X}, d_X)$ may be a feasible candidate satisfying the lemma. Beyond the range, the argument becomes trivial.

Now, set $\varepsilon = 4v_\delta^\lambda(\mu)$. As such, we can find a set of points $\{x_i\}_{i=1}^N \in \mathcal{X}$ which ensures $\mu(\bigcup_{i=1}^N B_X^\lambda(x, \varepsilon)) > 1 - \varepsilon$ with $l_\lambda(d_X(x_i, x_j)) > \varepsilon/2$ for all $i \neq j = 1, \cdots, N$. The set may contain arbitrarily far lying observations, yet the argument holds until $\lambda > \varepsilon/2$. Thus, following Mémoli (2011), Claim 10.2 we argue that $\forall\, i = 1, \cdots, N$ $\exists\, y_i \in \mathcal{Y}$ such that

$$\pi\Big( B_X^\lambda(x_i, \varepsilon) \times B_Y^\lambda(y_i, 2(\varepsilon + \delta)) \Big) \geq (1 - \delta^2)\mu(B_X^\lambda(x_i, \varepsilon)) > \delta(1 - \delta^2). \tag{46}$$

Observe that if $\lambda < 2(\varepsilon + \delta)$, the first inequality holds trivially. We denote by $S = \{(x_i, y_i), i = 1, \cdots, N\} \subset \mathcal{X} \times \mathcal{Y}$ the set of points constructing the nets.

**Step 2** *(Bounding locally robust distortions):* Consider $\{x_i\}_{i=1}^N$ and $\{y_i\}_{i=1}^N$ that satisfy (46) with $\mu(B_X^\lambda(x_i, \varepsilon)) > \delta$. Then, for all $i, j = 1, \cdots, N$

$$\left| l_\lambda(d_X(x_i, x_j)) - l_\lambda(d_Y(y_i, y_j)) \right| \leq 6(\varepsilon + \delta).$$

To prove the claim, let us first assume that there exists a pair $(i, j)$ for which it does not hold. As such, for all $x' \in B_X^\lambda(x_i, \varepsilon)$, $x'' \in B_X^\lambda(x_j, \varepsilon)$, $y' \in B_Y^\lambda(y_i, 2(\varepsilon + \delta))$, and $y'' \in B_Y^\lambda(y_j, 2(\varepsilon + \delta))$ we have

$$
\begin{aligned}
&\left| l_\lambda(d_X(x', x'')) - l_\lambda(d_Y(y', y'')) \right| \\
&\geq \left| l_\lambda(d_X(x_i, x_j)) - l_\lambda(d_Y(y_i, y_j)) \right| - \left| l_\lambda(d_X(x_i, x_j)) - l_\lambda(d_X(x', x'')) \right| \\
&\qquad - \left| l_\lambda(d_Y(y', y'')) - l_\lambda(d_Y(y_i, y_j)) \right| \\
&\geq 6(\varepsilon + \delta) - 3\varepsilon - 4(\varepsilon + \delta) = 2\delta.
\end{aligned}
$$

Then,

$$
\begin{aligned}
J_1^\lambda(\pi) &\geq 2\delta\, \pi\Big( B_X^\lambda(x_i, \varepsilon) \times B_Y^\lambda(y_i, 2(\varepsilon + \delta)) \Big) \pi\Big( B_X^\lambda(x_j, \varepsilon) \times B_Y^\lambda(y_j, 2(\varepsilon + \delta)) \Big) \\
&\geq 2\delta^3(1 - \delta^2)^2 > 2\delta^5,
\end{aligned}
$$

since $\delta \leq \frac{1}{2}$. This contradicts our initial assumption.

**Step 3** (*Constructing a suitable metric S*): Define $\tilde{d}_S^\lambda$ on $\mathcal{X} \sqcup \mathcal{Y}$ as

$$
(x, y) \mapsto \inf_{(x', y') \in S} \left[ l_\lambda(d_X(x, x')) + \left\| l_\lambda(d_X) - l_\lambda(d_Y) \right\|_{L^\infty(S \times S)} + l_\lambda(d_Y(y, y')) \right],
$$

also assuming $\tilde{d}_S^\lambda = l_\lambda(d_X)$ on $\mathcal{X} \times \mathcal{X}$ and $\tilde{d}_S^\lambda = l_\lambda(d_Y)$ on $\mathcal{Y} \times \mathcal{Y}$. Using Step 2, we get

$$
\tilde{d}_S^\lambda(x, y) \leq 2\lambda + 6(\varepsilon + \delta). \tag{47}
$$

However, for $i = 1, \cdots, N$, given $(x, y) \in B_X^\lambda(x_i, \varepsilon) \times B_Y^\lambda(y_i, 2(\varepsilon + \delta))$ we have

$$
\begin{aligned}
\tilde{d}_S^\lambda(x, y) &\leq \varepsilon + 2(\varepsilon + \delta) + \tilde{d}_S^\lambda(x_i, y_i) \\
&\leq \varepsilon + 8(\varepsilon + \delta), \tag{48}
\end{aligned}
$$

where the last inequality is due to Mémoli (2011), Lemma 10.1. Now, in pursuit of fragmenting $\mathcal{X} \times \mathcal{Y}$ based on balls around the points that constitute the maximal net, define $L = \bigcup_{i=1}^N B_X^\lambda(x_i, \varepsilon) \times B_Y^\lambda(y_i, 2(\varepsilon + \delta))$. Hence,

$$
\begin{aligned}
\int_{\mathcal{X} \times \mathcal{Y}} [\tilde{d}_S^\lambda(x, y)]^p d\pi(x, y) &= \int_L [\tilde{d}_S^\lambda(x, y)]^p d\pi(x, y) + \int_{\mathcal{X} \times \mathcal{Y} \backslash L} [\tilde{d}_S^\lambda(x, y)]^p d\pi(x, y) \\
&\leq (9(\varepsilon + \delta))^p + (\varepsilon + \delta)[2\lambda + 6(\varepsilon + \delta)]^p,
\end{aligned}
$$

where the last inequality is due to (47), (48) and the fact that $\pi(\mathcal{X} \times \mathcal{Y} \backslash L) \leq \varepsilon + \delta$ (Mémoli (2011), Claim 10.5), which is obvious if $\lambda < 2(\varepsilon + \delta)$. As such, for $p \in [1, \infty)$

$$
d_{l\mathrm{RSGW}}(\mu, \nu) \leq (4\lambda + \delta)^{\frac{1}{p}} \left( 62\lambda + \frac{15}{2} \right),
$$

since $\varepsilon \leq 4\lambda$ and $\delta \leq \frac{1}{2}$.

## A.4 Sample Complexity of Transform Sampling Using Tukey and LR-guided RGM Under Contamination

While the formulation (21) (see main paper) proposes altering the set of amenable couplings, based on our discussion in the first two sections, it is quite intuitive to think of a formulation that rather penalizes the norms. For example, we may construct a robust transform sampler drawing from both LR and Tukey's robustification techniques as follows:

$$
\inf_{F, G} \int \mathcal{T}_2 \Big( d_X(x, G(y)) - d_Y(y, F(x)) \Big) d\mu \otimes \nu(x, y) + \lambda_1 W_1^\lambda(\mu, G_\# \nu) + \lambda_2 W_1^\lambda(F_\# \mu, \nu), \tag{49}
$$

where the parameter underlying $\mathcal{T}_2$ is $\tau \geq 0$ and $\lambda \geq 0$. Typically, in an empirical problem, both $\tau, \lambda$ need to be tuned, and the infimum is taken over arbitrary measurable maps $F, G$ between spaces $\mathcal{X} \rightleftarrows \mathcal{Y}$. They need not follow the binding constraint, as the Lagrangian conditions ensure their measure preservation only. We assume that they are continuous and component-wise uniformly bounded in the following sense.

**Assumption 1.** *Given that $\mathcal{X} \subseteq \mathbb{R}^d$ and $\mathcal{Y} \subseteq \mathbb{R}^{d'}$, let $F_k(\cdot) : \mathcal{X} \to \mathbb{R}$ (similarly, $G_l(\cdot) : \mathcal{Y} \to \mathbb{R}$) denote the $k$th (and $l$th, $l = 1, \cdots, d$) coordinate of $F$, which is continuous (similarly, $G$), $k = 1, \cdots, d'$. There exists $b > 0$ such that for all $k = 1, \cdots, d'$, and $l = 1, \cdots, d$*

$$\sup_F \|F_k\|_\infty, \sup_G \|G_l\|_\infty \le b.$$

We denote such classes of functions as $\mathcal{F}_b^{\mathcal{X} \to \mathcal{Y}}$ and $\mathcal{F}_b^{\mathcal{Y} \to \mathcal{X}}$ respectively.

### A.4.1 Robust Concentration Inequalities

Given a pair of feasible maps $(F, G) \in \mathcal{F}_b^{\mathcal{X} \to \mathcal{Y}} \times \mathcal{F}_b^{\mathcal{Y} \to \mathcal{X}}$ we observe the non-asymptotic deviation of realized values of (49) from a robust population benchmark. We assume, without loss of generality, that $\lambda_1 = \lambda_2 = 1$. Also, let $x_1, x_2, \cdots, x_m$ are sampled following the $\mathcal{O} \cup \mathcal{I}$ framework with $\mu$ being the inlier distribution. Similarly, we have $y_1, y_2, \cdots, y_n$ from the other space with inliers drawn i.i.d from $\nu$. The inlying (outlying) set of samples are indexed using $\mathcal{I}^X$ and $\mathcal{I}^Y$ ($\mathcal{O}^X$ and $\mathcal{O}^Y$) respectively. Let us denote

$$T(\mu, \nu, F, G) := \int \mathcal{T}_2\Big(d_X(x, G(y)) - d_Y(y, F(x))\Big) d\mu \otimes \nu(x, y),$$

$$L(\mu, \nu, F, G) := W_1^\lambda(\mu, G_\# \nu) + W_1^\lambda(F_\# \mu, \nu).$$

As such, the population loss function in (49) can be written as $C(\mu, \nu, F, G) = T(F, G) + L(F, G)$. The empirical loss under contaminated measures $\hat{\mu}_m, \hat{\nu}_n$ is given as $T(\hat{\mu}_m, \hat{\nu}_n, F, G)$. However, to emphasize the robust translations we write the empirical version of $L(\mu, \nu, F, G)$ as $L(\hat{\mu}_m, \hat{\nu}_n, F, G) := W_1^\lambda(\mu, G_\# \hat{\nu}_n) + W_1^\lambda(F_\# \hat{\mu}_m, \nu)$. The following two results combined, present the concentration of $C(\hat{\mu}_m, \hat{\nu}_n, F, G)$ around $\frac{|\mathcal{I}^X||\mathcal{I}^Y|}{mn} \mathbb{E}_{\mu \otimes \nu}[T(\hat{\mu}_m^{\mathcal{I}}, \hat{\nu}_n^{\mathcal{I}}, F, G)] + \frac{|\mathcal{I}^Y|}{n} \mathbb{E}_\nu[W_1^\lambda(\mu, G_\# \hat{\nu}_n^{\mathcal{I}})] + \frac{|\mathcal{I}^X|}{m} \mathbb{E}_\mu[W_1^\lambda(F_\# \hat{\mu}_m^{\mathcal{I}}, \nu)]$.

**Proposition 5.** *There exists a constant $K > 0$ depending on $\tau^2$ such that for $\delta > 0$*

$$\left| T(\hat{\mu}_m, \hat{\nu}_n, F, G) - \frac{|\mathcal{I}^X||\mathcal{I}^Y|}{mn} \mathbb{E}_{\mu \otimes \nu}[T(\hat{\mu}_m^{\mathcal{I}}, \hat{\nu}_n^{\mathcal{I}}, F, G)] \right| \le K \frac{|\mathcal{I}^X||\mathcal{I}^Y|}{mn} \sqrt{\frac{\ln\left(\frac{4(|\mathcal{I}^X| \vee |\mathcal{I}^Y|)}{\delta}\right)}{|\mathcal{I}^X| \wedge |\mathcal{I}^Y|}} + \tau^2\Big(\frac{|\mathcal{O}^X|}{m} + \frac{|\mathcal{O}^Y|}{n}\Big)$$

*holds with probability at least $1 - \delta$.*

**Proof of Proposition 5**

Let us denote $t_{F,G}(x, y) := \mathcal{T}_2\Big(d_X(x, G(y)) - d_Y(y, F(x))\Big)$. Now, following the $\mathcal{O} \cup \mathcal{I}$ setup

$$T(\hat{\mu}_m, \hat{\nu}_n, F, G) = \frac{1}{mn} \sum_{i=1}^m \sum_{j=1}^n \mathcal{T}_2\Big(d_X(x_i, G(y_j)) - d_Y(y_j, F(x_i))\Big)$$

$$\le \frac{1}{mn} \sum_{i \in \mathcal{I}^X} \sum_{j \in \mathcal{I}^Y} \mathcal{T}_2\Big(d_X(x_i, G(y_j)) - d_Y(y_j, F(x_i))\Big) + \frac{\tau^2}{mn}\Big(|\mathcal{O}^X||\mathcal{O}^Y| + |\mathcal{O}^X||\mathcal{I}^Y| + |\mathcal{I}^X||\mathcal{O}^Y|\Big)$$

$$= \frac{|\mathcal{I}^X||\mathcal{I}^Y|}{mn} T(\hat{\mu}_m^{\mathcal{I}}, \hat{\nu}_n^{\mathcal{I}}, F, G) + \tau^2\Big(\frac{|\mathcal{O}^X|}{m} + \frac{|\mathcal{O}^Y|}{n} - \frac{|\mathcal{O}^X||\mathcal{O}^Y|}{mn}\Big).$$

Also,

$$\frac{|\mathcal{I}^X||\mathcal{I}^Y|}{mn} T(\hat{\mu}_m^{\mathcal{I}}, \hat{\nu}_n^{\mathcal{I}}, F, G) - \tau^2\Big(\frac{|\mathcal{O}^X|}{m} + \frac{|\mathcal{O}^Y|}{n} - \frac{|\mathcal{O}^X||\mathcal{O}^Y|}{mn}\Big) \le T(\hat{\mu}_m, \hat{\nu}_n, F, G).$$

As such, combining the two inequalities we get

$$\left| T(\hat{\mu}_m, \hat{\nu}_n, F, G) - \frac{|\mathcal{I}^X||\mathcal{I}^Y|}{mn} \mathbb{E}_{\mu \otimes \nu}[T(\hat{\mu}_m^{\mathcal{I}}, \hat{\nu}_n^{\mathcal{I}}, F, G)] \right|$$

$$\le \frac{|\mathcal{I}^X||\mathcal{I}^Y|}{mn} \left| T(\hat{\mu}_m^{\mathcal{I}}, \hat{\nu}_n^{\mathcal{I}}, F, G) - \mathbb{E}_{\mu \otimes \nu}[T(\hat{\mu}_m^{\mathcal{I}}, \hat{\nu}_n^{\mathcal{I}}, F, G)] \right| + \tau^2\Big(\frac{|\mathcal{O}^X|}{m} + \frac{|\mathcal{O}^Y|}{n} - \frac{|\mathcal{O}^X||\mathcal{O}^Y|}{mn}\Big). \quad (50)$$

Now, observe that

$$
\left| T(\hat{\mu}_m^{\mathcal{I}}, \hat{\nu}_n^{\mathcal{I}}, F, G) - \mathbb{E}_{\mu \otimes \nu}[T(\hat{\mu}_m^{\mathcal{I}}, \hat{\nu}_n^{\mathcal{I}}, F, G)] \right|
$$

$$
= \left| \frac{1}{|\mathcal{I}^X||\mathcal{I}^Y|} \sum_{i \in \mathcal{I}^X} \sum_{j \in \mathcal{I}^Y} t_{F,G}(x_i, y_j) - \mathbb{E}_{\mu \otimes \nu}[t_{F,G}(x, y)] \right|
$$

$$
\leq \left| \frac{1}{|\mathcal{I}^X|} \sum_{i \in \mathcal{I}^X} \left( \frac{1}{|\mathcal{I}^Y|} \sum_{j \in \mathcal{I}^Y} t_{F,G}(x_i, y_j) - \mathbb{E}_{\nu}[t_{F,G}(x_i, y)] \right) \right| + \left| \frac{1}{|\mathcal{I}^X|} \sum_{i \in \mathcal{I}^X} \mathbb{E}_{\nu}[t_{F,G}(x_i, y)] - \mathbb{E}_{\mu \otimes \nu}[t_{F,G}(x, y)] \right|.
$$

$$
(51)
$$

Recall that $M = \mathrm{diam}(\mathcal{X}) \vee \mathrm{diam}(\mathcal{Y})$. The function $|\mathcal{I}^X|^{-1} \sum_{i \in \mathcal{I}^X} t_{F,G}(x_i, y)$ satisfies the bounded difference inequality with parameter $|\mathcal{I}^X|^{-1}(4M^2 \wedge \tau^2)$. Hence, due to McDiarmid's inequality

$$
\left| \frac{1}{|\mathcal{I}^X|} \sum_{i \in \mathcal{I}^X} \mathbb{E}_{\nu}[t_{F,G}(x_i, y)] - \mathbb{E}_{\mu \otimes \nu}[t_{F,G}(x, y)] \right|
$$

$$
\leq \mathbb{E}_{\nu} \left| \frac{1}{|\mathcal{I}^X|} \sum_{i \in \mathcal{I}^X} t_{F,G}(x_i, y) - \mathbb{E}_{\mu}[t_{F,G}(x, y)] \right| \leq \sqrt{\frac{(4M^2 \wedge \tau^2)^2 \ln(2/\delta)}{2|\mathcal{I}^X|}}
$$

holds with probability at least $1 - \delta$, where the first inequality follows from Jensen's inequality. For the first term in (51), using the union bound over a similar argument, we get

$$
\left| \frac{1}{|\mathcal{I}^X|} \sum_{i \in \mathcal{I}^X} \left( \frac{1}{|\mathcal{I}^Y|} \sum_{j \in \mathcal{I}^Y} t_{F,G}(x_i, y_j) - \mathbb{E}_{\nu}[t_{F,G}(x_i, y)] \right) \right| \leq \sqrt{\frac{(4M^2 \wedge \tau^2)^2 \ln(2|\mathcal{I}^X|/\delta)}{2|\mathcal{I}^Y|}}
$$

with probability at least $1 - \delta$. As such,

$$
\left| T(\hat{\mu}_m^{\mathcal{I}}, \hat{\nu}_n^{\mathcal{I}}, F, G) - \mathbb{E}_{\mu \otimes \nu}[T(\hat{\mu}_m^{\mathcal{I}}, \hat{\nu}_n^{\mathcal{I}}, F, G)] \right| \lesssim \sqrt{\frac{\ln\left( \frac{2(|\mathcal{I}^X| \vee |\mathcal{I}^Y|)}{\delta} \right)}{|\mathcal{I}^X| \wedge |\mathcal{I}^Y|}}
$$

holds with probability $\geq 1 - 2\delta$. Hence, putting this back to (50) proves the result.

The proof also shows that it is always possible to replace the term $\frac{|\mathcal{I}^X||\mathcal{I}^Y|}{mn} \mathbb{E}_{\mu \otimes \nu}[T(\hat{\mu}_m^{\mathcal{I}}, \hat{\nu}_n^{\mathcal{I}}, F, G)]$ by $T(\mu, \nu, F, G)$, only by incurring an additional term on the upper bound of $\mathcal{O}\left( \frac{|\mathcal{O}^X|}{m} + \frac{|\mathcal{O}^Y|}{n} \right)$.

**Proposition 6.** *There exists a constant $\tilde{K} > 0$ depending on $\lambda$ such that for $\delta > 0$*

$$
\left| W_1^{\lambda}(\mu, G_{\#}\hat{\nu}_n) + W_1^{\lambda}(F_{\#}\hat{\mu}_m, \nu) - \frac{|\mathcal{I}^Y|}{n} \mathbb{E}[W_1^{\lambda}(\mu, G_{\#}\hat{\nu}_n^{\mathcal{I}})] - \frac{|\mathcal{I}^X|}{m} \mathbb{E}[W_1^{\lambda}(F_{\#}\hat{\mu}_m^{\mathcal{I}}, \nu)] \right|
$$

$$
\leq \tilde{K}\left( \frac{\sqrt{|\mathcal{I}^X|}}{m} + \frac{\sqrt{|\mathcal{I}^Y|}}{n} \right) \sqrt{\ln(4/\delta)} + \lambda\left( \frac{|\mathcal{O}^X|}{m} + \frac{|\mathcal{O}^Y|}{n} \right)
$$

*holds with probability at least $1 - \delta$.*

**Proof of Proposition 6**
First, let us note that $F_{\#}\hat{\mu}_m = \frac{1}{m} \sum_{i=1}^m \delta_{F(x_i)}$ based on the transformed observations $\{F(x_i)\}_{i=1}^m$. Technically, this is rather the empirical distribution $\widehat{(F_{\#}\mu)}_m$. Our consideration remains valid if $F$ is taken as an information preserving transform (IPT), which is in abundance (e.g., Lipschitz maps) (Chakrabarty et al., 2023). Hence, similar to the proof of Proposition 5, we observe

$$
\left| W_1^{\lambda}(F_{\#}\hat{\mu}_m, \nu) - \frac{|\mathcal{I}^X|}{m} W_1^{\lambda}(F_{\#}\hat{\mu}_m^{\mathcal{I}}, \nu) \right| \leq \frac{\lambda|\mathcal{O}^X|}{m}.
$$

This implies that

$$\left| W_1^\lambda(F_\# \hat\mu_m, \nu) - \frac{|\mathcal{I}^X|}{m} \mathbb{E}[W_1^\lambda(F_\# \hat\mu_m^\mathcal{I}, \nu)] \right| \leq \frac{|\mathcal{I}^X|}{m} \left| W_1^\lambda(F_\# \hat\mu_m^\mathcal{I}, \nu) - \mathbb{E}[W_1^\lambda(F_\# \hat\mu_m^\mathcal{I}, \nu)] \right| + \frac{\lambda|\mathcal{O}^X|}{m}. \tag{52}$$

Now, using the duality of $W_1^\lambda$ (see, Appendix A.3) we may write

$$W_1^\lambda(F_\# \hat\mu_m^\mathcal{I}, \nu) = \frac{1}{|\mathcal{I}^X|} \sup_{\substack{\phi \in C_b(\mathcal{Y}): \\ \mathrm{Range}(\phi) \leq \lambda}} \sum_{i \in \mathcal{I}^X} \phi(F(x_i)) - \mathbb{E}_\nu \phi^c.$$

Due to Assumption 1, the composition $\phi \circ F \in C_{b'}(\mathcal{X})$ for some $b' > 0$. As such, $W_1^\lambda(F_\# \hat\mu_m, \nu)$ satisfies the bounded differences property with upper bound $\mathcal{O}(\frac{1}{|\mathcal{I}^X|})$. Hence,

$$\left| W_1^\lambda(F_\# \hat\mu_m^\mathcal{I}, \nu) - \mathbb{E}[W_1^\lambda(F_\# \hat\mu_m^\mathcal{I}, \nu)] \right| \lesssim \sqrt{\frac{\ln(2/\delta)}{|\mathcal{I}^X|}}$$

holds with probability at least $1 - \delta$. Putting the bound back in (52) yields,

$$\left| W_1^\lambda(F_\# \hat\mu_m, \nu) - \frac{|\mathcal{I}^X|}{m} \mathbb{E}[W_1^\lambda(F_\# \hat\mu_m^\mathcal{I}, \nu)] \right| \leq K' \sqrt{\frac{|\mathcal{I}^X|}{m}} \sqrt{\frac{\ln(2/\delta)}{m}} + \frac{\lambda|\mathcal{O}^X|}{m},$$

that hold with probability at least $1 - \delta$, where $K' > 0$ depends on $\lambda$. Similarly, we can show that there exists $K'' > 0$ such that

$$\left| W_1^\lambda(\mu, G_\# \hat\nu_n) - \frac{|\mathcal{I}^Y|}{n} \mathbb{E}[W_1^\lambda(\mu, G_\# \hat\nu_n^\mathcal{I})] \right| \leq K'' \sqrt{\frac{|\mathcal{I}^Y|}{n}} \sqrt{\frac{\ln(2/\delta)}{n}} + \frac{\lambda|\mathcal{O}^Y|}{n}$$

also holds with probability $\geq 1 - \delta$. Combining the last two bounds proves the result.

**Remark 7** (Uniform Deviations). *The concentration inequalities make it easier to comment on the uniform deviation*
$\sup_{(F,G) \in \mathcal{F}_b^{\mathcal{X} \to \mathcal{Y}} \times \mathcal{F}_b^{\mathcal{Y} \to \mathcal{X}}} \left| T(\hat\mu_m, \hat\nu_n, F, G) - T(\mu, \nu, F, G) \right|$. *Let us assume both $d_X$ and $d_Y$ to be the Euclidean metrics in their respective spaces. Due to (50), the problem boils down to finding*

$$\sup_{(F,G) \in \mathcal{F}_b^{\mathcal{X} \to \mathcal{Y}} \times \mathcal{F}_b^{\mathcal{Y} \to \mathcal{X}}} \left| T(\hat\mu_m^\mathcal{I}, \hat\nu_n^\mathcal{I}, F, G) - \mathbb{E}_{\mu \otimes \nu}[T(\hat\mu_m^\mathcal{I}, \hat\nu_n^\mathcal{I}, F, G)] \right| =: T_{m,n}^\mathcal{I}.$$

*Since the underlying loss depends solely on inlying observations, using Proposition 4.6 of Hur et al. (2024), we obtain for any $\varepsilon > 0$*

$$T_{m,n}^\mathcal{I} \lesssim \sqrt{\frac{\ln\left(\frac{2(|\mathcal{I}^X| \vee |\mathcal{I}^Y|)}{\delta}\right)}{|\mathcal{I}^X| \wedge |\mathcal{I}^Y|}} + \varepsilon + \sqrt{\frac{\sum_{k=1}^{d'} \log N_\infty(\varepsilon, \mathcal{F}_k, |\mathcal{I}^X|) + \sum_{l=1}^{d} \log N_\infty(\varepsilon, \mathcal{G}_l, |\mathcal{I}^Y|)}{|\mathcal{I}^X| \wedge |\mathcal{I}^Y|}},$$

*holds with probability $\geq 1 - \delta$, where $\mathcal{F}_k$ and $\mathcal{G}_l$ are respectively the collections of amenable functions $F_k$ and $G_l$ satisfying Assumption 1. Classes $\mathcal{F}_k$ and $\mathcal{G}_l$ whose metric entropies scale according to $\mathcal{O}(1/\varepsilon)^a$, for some $a > 0$ are abundant. For example, Sobolev or Lipschitz-smooth functions defined on the unit interval $[0,1]^d$. One can similarly derive uniform deviation bounds corresponding to $L(\hat\mu_m, \hat\nu_n, F, G)$ and hence, eventually $C(\hat\mu_m, \hat\nu_n, F, G)$.*

## A.5 Existence of latent chaining

While it is difficult to characterize a suitable $\mathcal{Z}$ that follows the chaining argument given arbitrary $\mu$ and $\nu$, we give examples that conform to our experiments. Assume that $\mu, \nu \in \mathcal{P}_2^{\mathrm{ac}}(\mathbb{R}^d)$ are fully supported. Moreover, $\mathcal{Z}$ is convex, endowed with an absolutely continuous $\omega$ (e.g., Lebesgue). Then, due to Brenier's

polar factorization (Brenier, 1991), any transport map $T : \mathcal{Z} \to \mathcal{X}$ can be decomposed as $T = (\nabla\varphi) \circ s$ a.e., where $\varphi : \mathcal{Z} \to \mathbb{R}$ is convex and $s : \mathcal{Z} \to \mathcal{Z}$ is measure-preserving, both uniquely defined a.e. In fact, $(\nabla\varphi)$ is the unique optimal transport map between $\omega$ and $\mu$ under the Euclidean cost. Hence, we can construct $\phi_X := [(\nabla\varphi) \circ s']^{-1}$, where $s'$ is also bijective (see diagram (16) in the main paper). Similarly, define $\phi_Y' := (\nabla\varrho) \circ s''$, where $(\nabla\varrho)$ is the OT map between $\omega$ and $\nu$, and $s'' : \mathcal{Z} \to \mathcal{Z}$ preserves measure. As such, $F = (\nabla\varrho) \circ s'' \circ [(\nabla\varphi) \circ s']^{-1}$. One can similarly define $G$.

The same can be extended to $\mathcal{Z}$ (Also, $\mathcal{X}$ and $\mathcal{Y}$) being a connected, compact, $C^3$-smooth Riemannian Manifold without boundary (McCann (2001), Theorem 11). Then, any volume-preserving transport $T$ is represented as $\exp(-\nabla\varphi) \circ s$ a.e. Based on this, the rest of the construction follows exactly.

## B    Experiments and implementation details

We refer to the repository https://github.com/Thecoder1012/RCDA for all codes along with execution instructions. All experiments were carried out on an RTX 3090 GPU.

### B.1    Parameter selection in TGW and HGW

As mentioned before, we select $\tau = \tilde{m} + 3\tilde{\sigma}$, where $\tilde{m}$ and $\tilde{\sigma}$ are respectively the median and the mean deviation about median of the deviation values $J_{X,Y} = |d_X - d_Y|$. Observe that, for a univariate standard folded Normal random variable $Z$, $\mathbb{P}(Z \le \tilde{m}) = 2\Phi(\tilde{m}) - 1 = \frac{1}{2}$, where $\Phi(\cdot)$ is the distribution function of $N(0,1)$. As such, $\tilde{m} \approx 0.69$, the third quartile of $N(0,1)$. Now, given $a > 0$,

$$
\begin{aligned}
\tilde{\sigma} = \mathbb{E}\left[|Z - a|\right] &= \int_0^\infty |z - a|\, f(z) dz = \sqrt{\frac{2}{\pi}} \int_0^\infty |z - a|\, e^{-\frac{z^2}{2}} dz \\
&= \sqrt{\frac{2}{\pi}} \Big( \int_0^a (a - z) e^{-\frac{z^2}{2}} dz + \int_a^\infty (z - a) e^{-\frac{z^2}{2}} dz \Big) \\
&= \sqrt{\frac{2}{\pi}} (2e^{-\frac{a^2}{2}} - 1) + 4a\Phi(a) - 3a.
\end{aligned}
$$

As such at $a = 0.69$, we have $\tilde{\sigma} \approx 0.46$. The corresponding estimate for $\tau$ turns out to be $\approx 2.07$, which coincides approximately with the 96-percentile of $Z$. In our experiments, the deviation between pairwise distances under contamination does not follow such a law. Thus, calculating only a certain percentile becomes insufficient. Given a set of observations, we calculate the statistics independently and only use the percentiles as a reference.

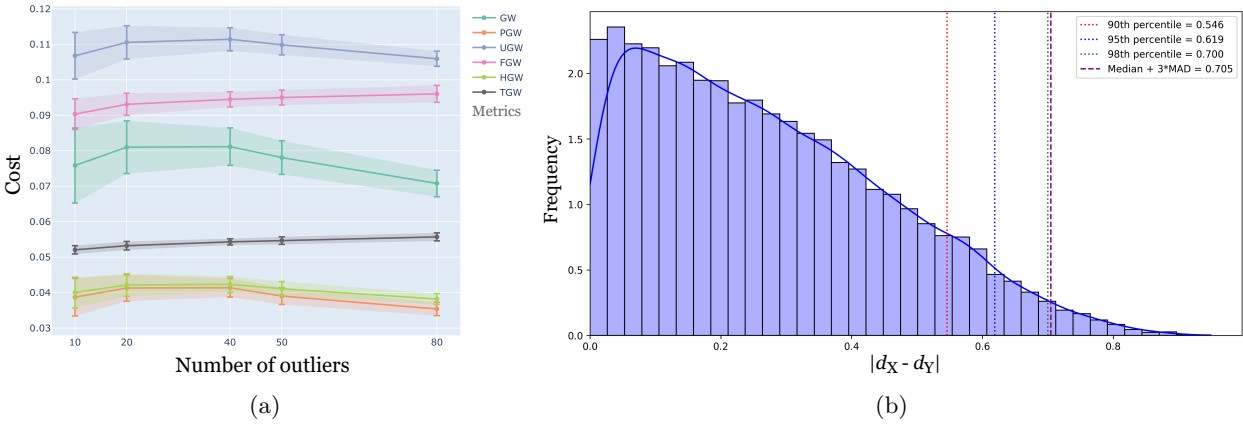

Figure 7:    (a) Average losses under increasing proportion of bi-variate standard Gaussian outliers $(0.02, 0.04, 0.08, 0.1, 0.16)$ in source. (b) Empirical distribution of $J_{X,Y}$ under 80 Gaussian outliers. Realized 95-percentile and $\tilde{m} + 3\tilde{\sigma}$ are 0.619 and 0.705 respectively. TGW follows the 95-percentile selection scheme while HGW is calculated based on $\tilde{m} + 3\tilde{\sigma}$.

Performances of the competing methods alter under a contaminating distribution with a thinner tail. While Cauchy implants vastly outlying observations with higher frequency, Gaussian outliers flock to the immediate neighborhood (see Figure 2(b) in the main paper). As a result, we observe a much more pronounced distortion of the shape instead of extremely large distortion values ($J_{X,Y}$). Moreover, as the number of outliers increases, only 'moderately' high-valued distances ($d_X$) in the source increase (see Figure 8). This noising phenomenon is more difficult to eliminate using our thresholding scheme as 'outlying' $d_X$ values are erroneously considered legitimate. As a result, the vanilla GW value, due to averaging, decreases even at elevated contamination levels. This is misleading since it does not reflect the distortion in local geometry. PGW and HGW perform well, and remarkably, TGW not only stays stable but also exhibits minute increments, indicating intensifying noise. The OT component's increase in FGW also demonstrates the same effect.

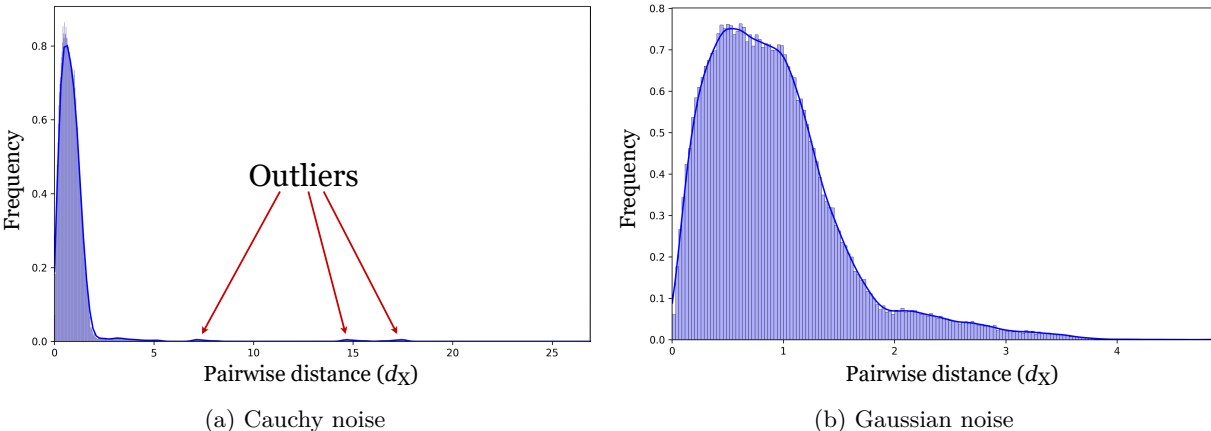

(a) Cauchy noise    (b) Gaussian noise

Figure 8: (a) Empirical density of pairwise distances $d_X(x, x')$ in the source shape (cat) with 40 outliers.

This motivates a rather finer thresholding technique we call local robustification. Instead of trimming extreme distortion values, we shift our focus to individual pairwise distances $d_X$ and $d_Y$.

## B.2 LR translation using GcGAN and UNIT

For images, contamination regimes become more complex than point data. In case there are clear outliers ($n\epsilon$ out of $n$) from other image sources (e.g., MNIST samples in a pool of facial images (Nietert et al., 2023)), the discriminators can still distinguish between them. In contrast, if all images have noise injected in them in a predefined proportion ($\alpha$), the resultant generations get much more affected. This is mainly due to the discriminators misclassifying them. Observe that if $\alpha = 1$, the noisy image from the second regime becomes an outlier from the first case. We maintain a flexible framework, striking a balance between the two.

For the experiment under GcGAN, first, we create a copy of the original image tensor to ensure its data remains unchanged. Next, we generate standard Gaussian noise with the same shape as the tensor, scaling it by $\alpha = 0.2$ to control the noise intensity. Finally, we add this scaled noise to the copied tensor to produce a noisy version of the original.

In the case of UNIT, we inject random bright pixels following a Gaussian law into the random images based on the image ratio. The wrapper supports datasets with or without labels by handling tuple or single-image outputs and seamlessly integrates with existing data loaders.

Regarding the validation setup, for the MNIST $\leftrightarrow$ USPS experiments, we use the standard train/test splits provided by the retrospective benchmark datasets, which are widely used in the literature. For the Apple-to-Orange experiments, we use the split provided in the publicly available dataset (https://www.kaggle.com/datasets/balraj98/apple2orange-dataset), which follows an approximately $80 - 20$ train/test partitioning.

**Ablation study: Parameter selection** Our experimental framework begins with analyzing the generator and discriminator loss propagation (Figure 9) under varying values of $\epsilon$. While smaller values imply weaker protection against noise, larger values tend to degrade discrimination performance (Figure 9b). Coupled with the quantitative scrutiny, we introduce an additional experiment based on the qualitative outcomes as in Figure 10. Here, we observe increased meddling in background color and oversaturation as $\epsilon$ increases. On the other hand, a lower parameter value increases the likelihood of noise being manifested in the resulting images. Based on a trade-off between both examinations, we infer that $\epsilon = 0.5$ consistently demonstrates balanced performance during training, prompting us to an optimal value.

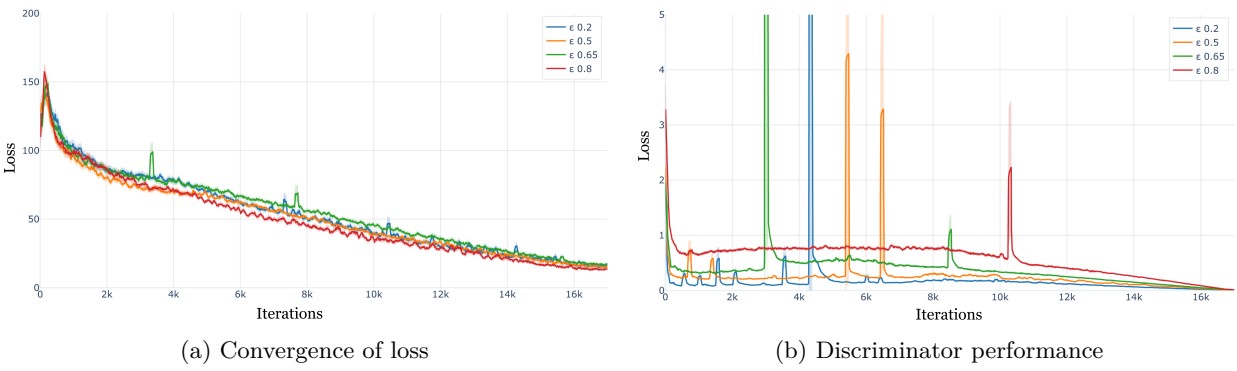

(a) Convergence of loss

(b) Discriminator performance

Figure 9: (a) Realized robust GcGAN loss for varying $\epsilon$ under Gaussian noise ($\alpha = 0.2$). There is no perceptible difference between $\epsilon$ values in this regard. (b) The discriminators also eventually perform similarly.

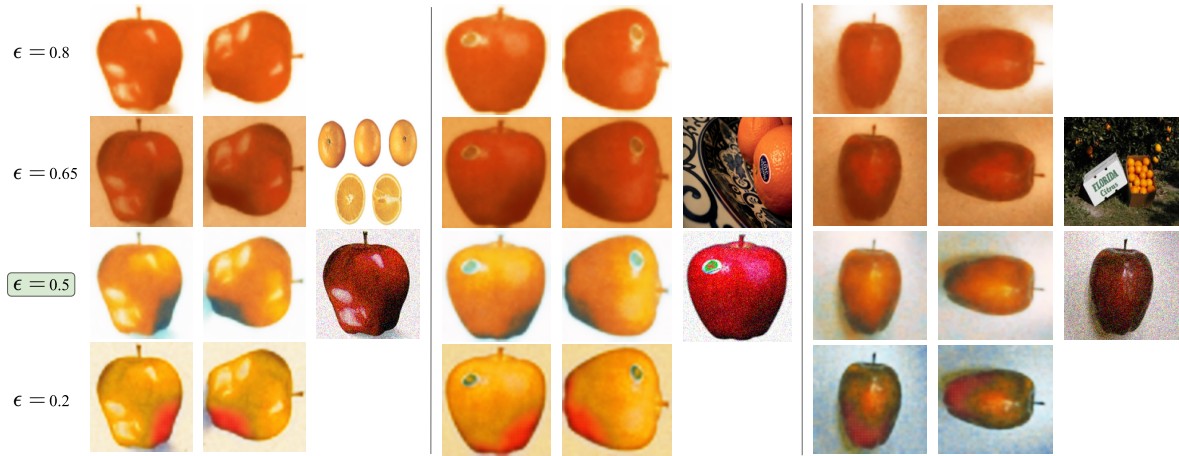

Figure 10: Style transfer performance of robust GcGAN for varying $\epsilon$. While small values ($\epsilon = 0.2$) produce inadequate denoising, high values ($\epsilon = 0.8$) distort the style and oversaturate images.

