# OpenReview forum: "Robust Cross-Domain Alignment"
_TMLR — Accepted by TMLR_

### Review · Reviewer_5aJV · 2026-03-14

**Summary Of Contributions:**

This paper studies robustness in Gromov–Wasserstein alignment under contamination. The central claim is well-motivated: in GW problems, outliers do not only corrupt the measures being coupled, they can also corrupt the pairwise distance structure that defines the alignment cost. The paper then develops three robustification routes. First, it introduces clipping/penalization of distortions, leading to Tukey- and Huber-style GW variants (TGW/HGW). Second, it proposes local robustification by truncating the underlying metrics before comparing pairwise distances, leading to LRGW and extensions to probabilistic metric-measure spaces. Third, it proposes a formulation, in the spirit of unbalanced OT, which relaxes the marginal constraints to f-divergence penalties. The paper combines theory, algorithms, and experiments on shape matching and noisy image translation.

Overall, this is a fairly ambitious paper with a good problem choice and a strong amount of material. At the same time, the manuscript is very dense and not especially well-organized. I found myself moving from proposition to proposition without a strong sense of which method should be preferred in which regime, or how the different parts fit into one clean narrative.

**Audience:**

Yes

**Audience Explanation:**

The paper addresses a notable weakness of standard GW, and it tries to do so in a way that is tailored to alignment, which is a worthwhile contribution.

**Broader Impact Concerns:**

None.

**Claims And Evidence:**

Yes

**Claims Explanation:**

The mathematical statements are given clear proofs and experimental validation is provided.

**Requested Changes:**

The paper needs a clearer structure. It introduces three robustification paradigms, several theoretical detours, and two generative-modeling directions, but it does not provide a simple organizing principle. I recommend adding a comparison table that says, for each method: what is being robustified, what guarantee is proved, what is lost, what objective is actually optimized in practice, and which downstream tasks it is meant for. Without that, the paper feels like a sequence of local results rather than a coherent story.

The computational discussion needs to be much more explicit. The paper correctly notes that GW is NP-hard in general, but it does not explain whether TGW, HGW, LRGW, or the plan-robustified formulations are easier, equally hard, or only tractable after entropic or other relaxations. Since TGW and LRGW reduce back to GW when the truncation parameters become large, and since the sample objectives still look like non-convex quadratic assignment problems, worst-case hardness should not be avoided.

Contributions: why does TGW “extend” ROBOT? It just seems to be a lower bound? Also, “lemma” should be capitalized.

Equations (2) and (4) are incorrect: the infimum over the coupling is missing.

In Proposition 1(ii), is $p \ge 0$ truly allowed? It seems that the Tukey loss was only defined for $p\ge 1$.

After Proposition 1, the text says TGW is a pseudometric on isomorphism classes; isn’t it a genuine metric?

In Remark 3, is LRGW supposed to be TGW?

Figure 4: is $\alpha = 0.2$ (main text) or $\alpha = 0.3$ (caption)?

In the style transfer experiment, it is noted that some of the improved image quality comes from using WGAN instead of vanilla GAN. This suggests that the experiments should be redone but with a fairer baseline comparison.

---

> ### Author Response · Authors · 2026-03-28
>
> We thank the reviewer for the meticulous read and insightful comments. Please find below our response to the reviews.
> ___
> > ### Clearer Structure
> >We appreciate the reviewer for raising this issue. We will include a discussion at the end of Section 4 that lays out the **Organization** of the upcoming analyses. It compares the three principled methods in terms of their objective, the underlying robustification technique based on mass truncation and redistribution, and potential applications where they tend to be beneficial. Since the reader at this point is yet to be acquainted with the propositions and their properties, we aim to keep the discussion simple and steer clear of most technical facts. In the modified **Discussion** section at the end (earlier **Conclusion**), we revisit the comparative analysis from a more theoretical perspective, highlighting the pros and cons. Below, we present a summary table of the discussion. Note that our motivation for introducing *Plan Regularization* lies primarily in developing robust translation architectures. It enforces a relaxed plan in a Reversible Gromov-Monge-type formulation, and we do not explicitly prove its topological properties.
> >___
> >
> >| Criterion | Norm Penalization (TGW/HGW) | Local Robustification (LRGW) | Plan Regularization (RRGM) |
> >|---|---|---|---|
> >| ``` Objective ``` | Deploy an outlier-robust norm (instead of $L^p$) to measure distortion | Replace the endowed metric of each mm space with robust proxies | Impose mass relaxation in the binding constraint corresponding to the (path-restricted) plan $\pi$ |
> >| ``` Robustness Guarantee ``` | Penalizes the contribution of paired observations from either space that causes $\geq \tau$ distortion | Penalizes contributions of observations from each space that lie $\geq \lambda$ apart | Uniformly redistributes relaxed $\epsilon$ mass from each distribution among respective inliers |
> >| ``` Utility ``` | Robust recovery of GW, shape matching | Cross-domain translation (via Sturm's formulation), shape matching, robust recovery of GW barycenters | Cross-domain translation |
> > ___
> >
> >To point out an immediate drawback of the plan regularization scheme, since it readily ties to the partial GW, it runs the risk of significant deviation from the unperturbed value. Meanwhile, both TGW and LRGW may suffer 'removable' degeneracies if the truncation parameters $= 0$.
>
>
> > ### Discussion on Computational Tractability
> As the reviewer has rightly pointed out, the TGW (also HGW) and LRGW formulations from an optimization perspective are **exactly** similar to the original GW. As such, we can only prescribe solving them via state-of-the-art algorithms, such as entropic regularization. The strength of the formulations rather lies in the fact that, besides ensuring outlier-robustness, they do not elevate underlying computational costs and allow seamless integration into existing libraries. As the discussion following **Algorithm 1** clearly outlines (also true for LRGW), they not only adapt to the lower complexity $O(m^2n+mn^2)$ but also conform to further regularization tricks such as Sparsification and Slicing. On the other hand, we develop the RRGM loss as an objective for cross-domain translation models parametrized by feedforward neural networks. As such, the underlying optimization results in an approximation no costlier than the reversible Gromov-Monge.
>
> > ### Extending ROBOT
> The reviewer is correct in pointing out that TGW turns out to be a lower bound of ROBOT. Our aim with the remark was to invoke the idea that TGW *extends* the robustification principle native to ROBOT in the same way GW extends the transportation problem (OT) onto otherwise incomparable spaces (in the form of alignment). We will clarify the same in the paper during revisions.
>
> > ### Soundness of Equations (2) \& (4)
> We would like to clarify that the definitions of OT (Eq. (1)) and GW (Eq. (3)) when plugged into the latter equations ((2) and (4)) make them technically sound.
>
> > ### Correction in Proposition 1(ii)
> We thank the reviewer for pointing this out. It should indeed be restricted to $p \geq 1$. We will rectify the same in the revised paper.
>
> > ### Metricity of TGW
> Observe that the definition of the Tukey norm assumes $\tau \geq 0$ (**Definition 1**). As such, when $\tau = 0$, there may occur a trivial degeneracy of the resultant TGW, i.e., $d_{\textrm{TGW}}(\mu, \nu)=0$ even if $\mathcal{X} \ncong \mathcal{Y}$. Fixing $\tau > 0$ solves the issue. This barely depreciates the applicability of the measure since, in practical scenarios, the thresholding is anyway non-zero.
>
> > ### Corrections in Remark 3 \& Figure 4
> We thank the reviewer again for spotting the lapses. In **Remark 3**, it should indeed be '*TGW*'. As for the caption in **Figure 4**, we have since rechecked our experiments and can confirm that it should also be $\alpha = 0.2$.

---

> ### Author Response · Authors · 2026-03-28
>
> > ### Fairer Baseline in the Style Transfer Experiment
> We appreciate the reviewer's perceptive observation. We have since performed controlled ablation experiments to deconvolute the improvement in our framework (RUNIT) due to the usage of WGAN-based translation from the refinement due to robustification (**Figure 5**). Since the underlying robust penalization and the equivalence of losses hinge on the OT argument, we focus on the modified baseline of UNIT, now equipped with WGAN-based regularization. We find that the modification results in comparable image quality (**FID = 301.25** vs earlier UNIT baseline producing FID = 304.39). As such, the earlier improvement can be solely attributed to the robustification mechanism. In hindsight, perhaps such a result was always likely given the observation by Zhu et al. [1] that replacing $L^1$ objectives in cross-domain translation architectures by Wasserstein losses often leads to similar outcomes (later corroborated theoretically under smoothness assumptions on data densities [2]). We will modify the section accordingly during revisions.
> >___
> >
> >[1] Zhu, J.Y., Park, T., Isola, P. and Efros, A.A. Unpaired Image-to-Image Translation using Cycle-Consistent Adversarial Networks. In *ICCV 2017*.
> >
> >[2] Chakrabarty, A. and Das, S. On Translation and Reconstruction Guarantees of the Cycle-Consistent Generative Adversarial Networks. In *NeurIPS 2022*.
>
> > Adhering to the rebuttal guidelines of TMLR, we will wait till all reviews are posted to revise the paper.

---

> > ### Comment · Reviewer_5aJV · 2026-03-29
> >
> > Thanks for the responses. Just two quick comments:
> >
> > 1. For computational tractability, I think it is important to mention that the cost you mention is the **per-iteration cost** and that the iterations are not guaranteed to converge to the global minimizer.
> >
> > 2. Regarding eq. (2) and (4), I still believe they are not correct as written. EOT is not simply the OT cost plus the entropy of the OT plan; instead, EOT involves **re-optimizing the plan** over an objective which is a sum of the cost + entropy. In particular, the transport cost of the optimal EOT plan is not the same as the OT cost.

---

> > > ### Author Response · Authors · 2026-04-01
> > >
> > > We appreciate the reviewer's prompt response.
> > > ___
> > > > **1.** While we agree that the cost is indeed what is incurred per iteration, perhaps it would be unfair to say that they are not guaranteed to converge. Rioux et al. [3] explicitly show that an $(e^\delta -1)-$oracle approximation of the optimal plan can be achieved using a Sinkhorn scaling (e.g., Algorithm 1) algorithm in finite time under entropic regularization, $\delta > 0$ (see Proposition 8). The number of iterations depends on the *two distributions*, *their joint moments*, *the parameter associated with entropic regularization*, and $\delta$. Note that the result only holds for $p=2$ (which is commonly used in practice), i.e., when the EGW objective can be broken down into a constrained OT problem in the spirit of Theorem 4. Huber's GW adheres to this, given the definition of the associated norm.
> > > > ___
> > > > [3] Rioux, G., Goldfeld, Z. and Kato, K. Entropic Gromov-Wasserstein Distances: Stability and Algorithms. In *JMLR 2024*, 25(363), pp.1-52.
> > >
> > > >**2.** We agree with the reviewer on this and will write the costs explicitly to avoid ambiguity during revisions.

---

### Review · Reviewer_w8aX · 2026-03-18

**Summary Of Contributions:**

The paper presents robust gromov wasserstein formulations with three approaches: (1) by penalizing large distortions to GW loss similar to Tukey & Huber (2) with a relaxed metric (3) by regularizing based on proxy distribution. Overall, the idea seems novel and the paper derives related theoretical guarantees also connecting to the robust OT method, ROBOT. The experiments on cross domain translation & style transfer show the robustness introduced by the proposed formulation.

**Audience:**

Yes

**Audience Explanation:**

Robustness in OT based formulations is an active area of research. While the paper tries to bring in concepts from robust statistics to the GW formulation which would potentially open avenues for further research at the intersection of these two areas.

**Claims And Evidence:**

Yes

**Claims Explanation:**

The paper tries to bring in concepts from robust statistics to the GW formulation & derives related guarantess on (pseudo)metric properties & robustness. However, the contributions related to robust statistics could be improved by including a discussion on
- Bias of the truncated GW loss
- Breakdown point analysis of the robust estimator
- Influence function based analysis.

From robust OT literature point of view, the contributions seem novel & useful.

**Requested Changes:**

I) The experimental evaluation is limited to small datasets but considering the theoretical motivation for the work, I would be okay with that. However, the paper should potentially be compared against the following works

- “Metric properties of partial and robust Gromov-Wasserstein distances” by Jannatul Chhoa, Michael Ivanitskiy, Fushuai Jiang, Shiying Li, Daniel McBride, Tom Needham, and Kaiying O’Hare

- “Robust Alignment via Partial Gromov-Wasserstein Distances” by Xiaoyun Gong, Sloan Nietert, Ziv Goldfeld
There should also be a discussion on the above two papers in the Related Work.

II) Few questions related to the proofs:
- Can the paper cite or prove the triangle inequality property of Tukey norm used in Proposition 1’s proof?
- Please also cite or prove the triangle inequality of truncated OT used in Eq. (28).
- Please also expand on the OT relation used for (29) and followed by the linear scaling property used for truncated OT in the next step

III) The paper downweighs mass-trimming based approaches by mentioning the deviation from the balanced counterpart (“While such a mass-trimming approach penalizes outliers, the resultant distance suffers significant deviations from its balanced counterpart”). However, the mismatch between OT & UOT distance doesn’t seem a reason to discard the UOT based approaches. UOT on its own has good theoretical properties including the metric properties (in contrast to the pseudo-metric property of the approach presented in the paper). So if the robustness literature hinges on UOT based formulations, I do not see a problem with this. May be the paper can explain this point better or just propose the current approach as an alternative to mass-trimming based approaches.

IV) There should be a discussion on how the hyperparameters were chosen for the experiments including the details of the validation set.

Optional V) There could be additional illustrative discussions or toy experiments comparing the three proposed approaches to showcase each one's utility.

---

> ### Author Response · Authors · 2026-04-13
>
> We thank the reviewer for the pertinent comments and constructive suggestions. Please find below our response to the reviews.
> ___
> >### Bias, Breakdown Point \& Influence function
> > This is an interesting point, and we thank the reviewer for raising the same. We would like to preface our answer with the accrued knowledge of breakdown point (BP) analysis from OT and the existing literature on outlier robust GW. One can immediately observe that it is only fairly recently ([I], [II]) that we have come to analyze the Donoho-Huber BP of OT maps, whose findings can be generalized for regular convex cost functions [III]. Earlier, it was only conjectured that the BP of $W^{\epsilon}_{p}$ (as in (12) in our paper) can, in principle, be "*pushed to its information-theoretic limit of 1/2*" (see [IV], Remark 2). However, extending the truncation technique to GW may render the resulting $GW^{\epsilon}$ non-informative if $\epsilon \geq 1/3$ (see [V], Remark 2.5). It can supposedly be improved based on projection estimates of perturbed measures; however, we could only find a limited explanation of the improvement and potential limits.
> >
> > In our work, in contrast, we have already shown the upper bound on the realized bias of TGW while recovering the unperturbed GW under the $\mathcal{O} \cup \mathcal{I}$ framework (see Remark 3). Following the suggestion, we introduce a more general corruption regime for BP analysis as follows. Let $T$ be an estimator based on samples $Z = \\{(x_1,y_1),\ldots,(x_n,y_n)\\}$. *Corrupted samples* $Z'$ are obtained by replacing any $k$ of the $n$ data points by arbitrary values. For simplicity and to maintain consistency with the standard setup [I], we only allow perturbations on $Z = \\{x_1,\ldots,x_n\\}$ (the source sample, with target fixed). Let us define the *maximum bias* caused by $k$ replacements:
> >
> >>$\mathrm{bias}(k; T, Z) = \sup_{Z'} |T(Z') - T(Z)|$,
> >
> >where the supremum is over all such corrupted $Z'$. Observe that, due to the definition of the Tukey norm, the bias associated with TGW never diverges (unlike standard GW, where replacing even $k = 1$ point with magnitude $\to \infty$ forces its transport and diverges the cost). In a discrete setup, since the TGW cost can be broken down into three parts (due to the decomposition of the corrupted coupling as $\gamma' = \frac{n-k}{n}\gamma_{\text{inlier}} + \frac{k}{n}\gamma_{\text{outlier}}$ and by expanding the double integral), we can show that
> >
> >>$\mathrm{bias}(k;T_{\text{TGW}}, Z) \leq \tau^p \frac{k(2n-k)}{n^2}$,
> >
> >which is increasing in $k$, reaches $\tau^p$ only at $k=n$, and equals $0$ at $k=0$. We will include a detailed proof with the necessary discussion in the revised paper. This is a rather graceful degradation curve, in contrast to the step-function breakdown of GW. We may further define a more refined notion of the Donoho-Huber BP to analyze TGW since it is hard-bounded.
> >
> >>The **$\delta$-breakdown point** is the smallest contamination fraction that can push the bias beyond $\delta \tau^p$, defined as
> >$\varepsilon_{n}^{\delta}(T_{\text{TGW}}, Z) = \min\left\\{\frac{k}{n} : \mathrm{bias}(k;T_{\text{TGW}}, Z) > \delta \tau^p\right\\}$.
> >
> >This gives the extent within the effective range to which the degradation reaches due to perturbation. We find that $\varepsilon_{n}^{\delta}(\text{TGW}) = 1 - \sqrt{1-\delta}$. As such, for $\delta = 1/2$: $\varepsilon^{1/2} = 1 - 1/\sqrt{2} \approx 0.293$.
> >
> >To calculate the Influence function (IF) $(\text{IF}(z; T,\mu) = \lim_{\epsilon \to 0^+} \frac{T(\mu_\epsilon, \nu) - T(\mu,\nu)}{\epsilon} = V'(0)$, where $V(\epsilon) = T(\mu_\epsilon, \nu)$ such that $\mu_\epsilon = (1-\epsilon)\mu + \epsilon \delta_z$, for $\epsilon \in [0,1])$ of TGW, the way we see it, one *requires* the dual potentials to satisfy a similar breakdown to that in the case of GW. Note that we do not show a duality for TGW in the paper, which only appears as an upper bound to LRGW. Given that the derivative of the Tukey norm (kernel) is uniformly bounded by $p \tau^{p-1}$, the TGW IF is expected to saturate within a finite radius. Perhaps the discussion would benefit from an independent scrutiny.
>
> ___
> >[I] Avella-Medina, M. and González-Sanz, A., 2024. On the breakdown point of transport-based quantiles. arXiv preprint arXiv:2410.16554.
> >
> >[II] Paindaveine, D. and Passeggeri, R., 2024. On the robustness of semi-discrete optimal transport. arXiv preprint arXiv:2410.19596.
> >
> >[III] Gonzalez-Sanz, A. and Medina, M.A., 2026. Breakdown properties of optimal transport maps: general transportation costs. arXiv preprint arXiv:2603.16005.
> >
> >[IV] Nietert, S., Goldfeld, Z. and Cummings, R. Outlier-robust optimal transport: Duality, structure, and statistical analysis. *In AISTATS 2022*, (pp. 11691-11719). PMLR.
> >
> >[V] Gong, X., Nietert, S. and Goldfeld, Z., 2025. Robust Alignment via Partial Gromov-Wasserstein Distances. arXiv preprint arXiv:2506.21507. (Since been published *In ISIT 2025*.)

---

> ### Author Response · Authors · 2026-04-13
>
> > ### Comparison with Related Works
> >We thank the reviewer for the suggestion. After carefully going through the articles, we come across some remarkable connections between propositions therein and our formulations.
> >
> >___
> > - Chhoa et al.'s [VI] notion of a relaxed coupling (Definition 2.2) relates directly to our version of mass trimming during translations (see (20)). Essentially, $\Pi_{\epsilon}$ can be understood as a path-restricted analogue of $\mathcal{C}_{\epsilon}$ (due to [VI]) under a robust binding constraint. Furthermore, this is equivalent to Chapel et al.'s [VII] notion of partial alignment up to scaling (Corollary 3.14). As such, to put it informally, what reversible Gromov-Monge is to GW, RRGM is to PGW (Definition 2.7 in [VI]). We also note that the distinct operations for relaxing couplings (based on *mass trimming* (RRGM) and *mass addition* (PGW)) essentially yield equivalent robustification, given [IV] (Proposition 1). They also go on to prove the existence of such relaxed optimal couplings (Theorem 2.13 in [VI]), which in turn further ratifies our formulation of partial robustification along transformations.
> >
> > - On the other hand, [V] adopts Chapel et al.'s notion of partial alignment. We refer to Section 3.3.1 in [VI] for a precise distinction between the two notions.
> >
> >___
> > [VI] Chhoa, J., Ivanitskiy, M., Jiang, F., Li, S., McBride, D., Needham, T. and O'Hare, K., 2024. Metric properties of partial and robust Gromov-Wasserstein distances. arXiv preprint arXiv:2411.02198.
> >
> > [VII] Chapel, L., Alaya, M.Z., and Gasso, G. Partial optimal transport with applications on positive-unlabeled learning. *In NeurIPS 2020*, pp.2903-2913.
>
> > ### Questions Related to Proofs
> > - We would like to point out that the proof of Proposition 1(i) opens with proving the triangle inequality of the Tukey norm $\lVert{\cdot}\rVert_{\mathcal{T}_{p}}$. We will highlight the part further during revision.
> >
> > - Observe that since $W_{\mathcal{T}_{p}}(\mu, \nu) = \inf\_{\pi \in \Pi(\mu, \nu)} \lVert d\_{X} \rVert\_{\mathcal{T}\_{p}(\pi)} = \inf\_{\pi \in \Pi(\mu, \nu)} \lVert l\_{\tau}( d\_{X}) \rVert\_{L^{p}(\pi)}$, the triangle inequality and the following step (29) turn out to be mere extensions of similar results from OT, now under the altered cost $l\_{\tau}(d\_{X}) = \min\\{d\_{X}, \tau\\}$, which also becomes a metric if $d\_{X}$ is so.
> >Let $\mu\_1, \mu\_2, \mu\_3 \in \mathcal{P}(\mathcal{X})$, and let $\pi\_{12} \in \Pi(\mu\_1, \mu\_2)$, $\pi\_{23} \in \Pi(\mu\_2, \mu\_3)$ be optimal transport plans. By the Gluing Lemma, there exist a probability measure $\pi \in \mathcal{P}(\mathcal{X} \times \mathcal{X} \times \mathcal{X})$ with marginals $\pi\_{12}$, $\pi\_{23}$ and $\pi\_{13}$ on $\mathcal{X} \times \mathcal{X}$. As such,
> >>$$W\_{\mathcal{T}\_{p}}(\mu\_1, \mu\_3) \leq \lVert d\_{X}(x\_1, x\_3) \rVert\_{\mathcal{T}\_{p}(\pi\_{13})} = \lVert d\_{X}(x\_1, x\_3) \rVert\_{\mathcal{T}\_{p}(\pi)} \leq \lVert d\_{X}(x\_1, x\_2)\rVert\_{\mathcal{T}\_{p}(\pi)} + \lVert d\_{X}(x\_2, x\_3) \rVert_{\mathcal{T}\_{p}(\pi)} = \lVert d\_{X}(x\_1, x\_2) \rVert\_{\mathcal{T}\_{p}(\pi\_{12})} + \lVert d\_{X}(x\_2, x\_3) \rVert\_{\mathcal{T}\_{p}(\pi\_{23})} = W\_{\mathcal{T}\_{p}}(\mu\_1, \mu\_2) + W\_{\mathcal{T}\_{p}}(\mu\_2, \mu\_3).$$
> >>
> > We will include the same in the updated manuscript.
> >
> > - The inequality (29) and the following step, together, are an immediate extension of the first inequality in the Proof of Lemma 3, [IV].
>
> > ### Clarification on UOT
> > We would like to emphasize that the goal of the said section is to contextualize the problem by highlighting the obvious limitations of existing methods. We do this by also acknowledging that perhaps it is unrealistic to expect a unified solution for all underlying tasks where GW-based alignment is relevant. Unbalancing, while being adopted widely as a means to instill robustness, suffers a noticeable drawback in its deviation from the benchmark OT cost. The remark we make in the paper is based on the observation by Nguyen et al. [VIII] (see Section "*Bottleneck of robust OT computation via UOT*"). As such, in tasks where it is crucial to recover the unperturbed OT cost based on a proxy, UOT might not be the ideal choice. We do not prescribe rejecting UOT by any means.
> >
> >___
> [VIII] Nguyen, Q.M., Nguyen, H.H., Zhou, Y., and Nguyen, L.M. On unbalanced optimal transport: Gradient methods, sparsity and approximation error. *In JMLR 2023*, 24(384), pp.1-41.

---

> ### Author Response · Authors · 2026-04-13
>
> > ### Discussion on Hyperparameters
> > We appreciate the reviewer bringing up this issue. Appendix B of the manuscript has a thorough explanation of our hyperparameter selection procedure. In particular, Appendix B discusses the learning rate tuning process for various translation tasks, the parameter selection technique for both the TGW and HGW formulations, and an ablation study examining how sensitive our proposal is to key hyperparameter choices. Together, these provide a thorough account of how the final configurations were determined and justify the settings used in our main experiments.
> >
> >Regarding the validation setup, for the MNIST $\leftrightarrow$ USPS experiments, we use the standard train/test splits provided by the retrospective benchmark datasets, which are widely used in the literature. For the Apple-to-Orange experiments, we use the split provided in the publicly available dataset ([https://www.kaggle.com/datasets/balraj98/apple2orange-dataset](URL)), which follows an approximately $80-20$ train/test partitioning.
>
> > ### Additional Discussion Comparing the Methods
> > We will include a discussion at the end of Section 4, laying out the **Organization** of the upcoming analyses. It compares the three principled methods based on their *motivation*, the underlying *robustification technique based on mass redistribution*, and *potential applications* where they tend to be beneficial. Since the reader at this point is yet to be acquainted with the propositions and their properties, we keep the discussion simple and steer clear of most technical facts. The modified **Discussion** section at the end (earlier **Conclusion**) will revisit the comparative analysis from a more theoretical viewpoint. Below, we present a summary table of the discussion.
> >___
> >
> >| Criterion | Norm Penalization (TGW/HGW) | Local Robustification (LRGW) | Plan Regularization (RRGM) |
> >|---|---|---|---|
> >| ``` Objective ``` | Deploy an outlier-robust norm (instead of $L^p$) to measure distortion | Replace the endowed metric of each mm space with robust proxies | Impose mass relaxation in the binding constraint corresponding to the (path-restricted) plan $\pi$ |
> >| ``` Robustness Guarantee ``` | Penalizes the contribution of paired observations from either space that causes $\geq \tau$ distortion | Penalizes contributions of observations from each space that lie $\geq \lambda$ apart | Uniformly redistributes relaxed $\epsilon$ mass from each distribution among respective inliers |
> >| ``` Utility ``` | Robust recovery of GW, shape matching | Cross-domain translation (via Sturm's formulation), shape matching, robust recovery of GW barycenters | Cross-domain translation |
> >____
>
> > Adhering to the rebuttal guidelines of TMLR, we will wait till all reviews are posted to revise the paper.

---

> ### Comment · Reviewer_w8aX · 2026-05-28
>
> Thank you for clarifying the relation of the proposed formulation to partial/robust GW, relaxed couplings, mass trimming, and plan regularization. This discussion improves the positioning of the paper.
>
> I have some suggestions on the additional part added in Appendix A.1 which do not affect my positive score for the paper. It might be worth for the authors to re-check if Eq. (33) is for the un-rooted $\Phi$ or for the TGW distance and whether the equality statement before Eq. (33) should be an inequality and it's possible consequences in computing the breakdown point.

---

> > ### Author Response · Authors · 2026-06-04
> >
> > We thank the reviewer for the follow-up. Equation (33) (Appendix A.1) is indeed for the TGW distance. However, we are not sure what the reviewer refers to as 'the un-rooted $\Phi$'. As for the equality, since we take $\sup$ over arbitrary outlying observations, the statement holds. Note that it is perhaps irrelevant in the context of BP analysis to consider outlying observations that incur distortions $\leq \tau$ (in outlier-outlier or outlier-inlier interactions), as the distance is anyway non-negative and the effective contribution towards the bias is bounded. As such, the subsequent BP computation remains consistent.

---

### Review · Reviewer_2R4E · 2026-05-05

**Summary Of Contributions:**

This paper studies robustness issues in the Gromov-Wasserstein (GW) distance, a widely used tool for comparing distributions across different metric spaces. The authors argue that existing approaches to robustifying GW are not fully suited to the cross-domain alignment setting.

To address this, the paper introduces three novel robust GW variants derived from principles in robust statistics. For each variant, the authors provide theoretical analysis, including metric properties and robustness guarantees, as well as relationships to classical GW. These methods are further supported by empirical evaluations on several machine learning tasks, where they demonstrate improved robustness to contamination compared to existing approaches.

Overall, the paper combines theoretical contributions with experimental validation, and proposes a new perspective on robustness in GW that departs from standard OT-based techniques.

**Audience:**

Yes

**Audience Explanation:**

The GW distance is an important tool in machine learning, with applications in domain adaptation, generative modeling, and graph analysis. Robustness to contamination is a well-known limitation, and addressing it is of clear interest to the community.

This paper proposes new approaches that go beyond standard OT-inspired techniques, which is likely to be of interest to researchers working on optimal transport, geometric learning, and robustness in ML. In particular, the connection to robust statistics offers a fresh perspective that could inspire future work.

However, the current presentation leans heavily toward theoretical development, with limited emphasis on intuition and practical insights. While this does not diminish the value of the contribution, improving accessibility would broaden its appeal within the TMLR audience.

**Claims And Evidence:**

Yes

**Claims Explanation:**

The paper provides both theoretical and empirical evidence to support its claims.

On the theoretical side, the authors introduce formal results (including several theorems) that motivate the proposed robust GW variants in different contamination settings. These results help clarify the conditions under which each variant is appropriate and how they relate to standard GW.

On the empirical side, the experiments demonstrate improved robustness compared to existing methods such as partial GW and unbalanced GW. In particular, the results in Sections 4.2 and 4.3 illustrate the effectiveness of the proposed approaches on practical image-to-image translation tasks, supporting the claim that they better handle contamination.

That said, while the evidence is generally convincing, clarity could be improved. Some notational choices and definitions are difficult to follow, which makes it harder to fully grasp the theoretical claims. Improving exposition would strengthen the overall persuasiveness of the paper.

Examples: I can't find a definition for $M$ in Remark 3 and following equations (is it the same as the one defined a few pages later, after Proposition 2?), also I don't find the tensor notation $\mathcal{H}(C^X-C^Y)$ very intuitive in eq (7), maybe at least stating what this tensor is could help, but that one may be more of a personal preference.

**Requested Changes:**

- Improve clarity and notation (cf examples above)
- Provide more intuition and high-level insights: The paper is quite mathematically dense, which can make it difficult to extract the main ideas and motivations. Adding more intuition behind the proposed variants (e.g. relying on illustrative examples beyond the experiments) would make the contributions more accessible. Maybe consider including brief summaries or interpretations after key theoretical results.
- Rebalance main text vs. supplementary material: Some of the more technical derivations could be moved to the supplementary material to improve the flow of the main paper. This would allow the main text to focus more on intuition, key results, and practical implications. On the other side, I think experimental sections would benefit from better description of the task at hand and the setup without referring to the supplementary material.

---

> ### Author Response · Authors · 2026-05-12
>
> We thank the reviewer for the encouraging remarks and constructive suggestions. Please find below our response to the questions raised.
> ___
> >### Clarity and Notation
> > The notation $M \coloneqq \textrm{diam}(\mathcal{X}) \vee \textrm{diam}(\mathcal{Y})$ first appears in the footnote in Page 6. We acknowledge that its current placement may be difficult to follow, and we will clearly define it in the **Notations** section during revision. In Eq. (7), $\mathcal{H}(C^{X} - C^{Y}) \coloneqq (\mathcal{H}(C^{X}\_{ii'} - C^{Y}\_{jj'}))\_{i,j,i',j'}$, i.e., pairwise distortions truncated according to the Huber loss.
>
> >### Intuition and High-level insights
> > We thank the reviewer for raising this issue. As `Reviewer w8aX` and `5aJV` have both recommended, we will include a high-level comparison of the proposed methods, outlining their objectives and utility. For reference, kindly see our response, titled '**Clearer Structure**', to `Reviewer 5aJV`. We will also highlight key takeaways from the theoretical results while rewriting.
>
> >### Rebalancing Main text vs. Supplementary Material
> > While we promise to improve the flow of the paper during revisions, we would like to mention that we have only placed technical derivations in the main article that, in our opinion, are crucial to the narrative and add value to the overall understanding, for example, the discussion following Theorem 4. All proofs have been deferred to the Supplementary Material. For the experiments, we will include targeted remarks to better contextualize each task.

---

### Decision · Action_Editor_E22h · 2026-06-11

**Recommendation:** Accept as is

**Additional Comments:**

Overall, the reviewers appreciated the paper's contributions. The authors also resolved many of the concerns with revisions.

To authors: in the camera-ready version, please resolve all editorial / suggested changes.

**Audience:**

Yes

**Audience Explanation:**

The paper addresses robustness of Gromov-Wasserstein alignment, which is relevant to the readers working on optimal transport, geometric ML, domain adaptation, graph/network alignment, and image-to-image translation. The three-way organization in Figure 1 is also useful because it clarifies different ways to make GW-style alignment robust under outliers.

**Claims And Evidence:**

Yes

**Claims Explanation:**

The paper highlights the fragility of standard Gromov-Wasserstein (GW) alignment when exposed to contamination, noting that outliers can easily corrupt the underlying pairwise distance structure (Section 4). To address this vulnerability, the authors propose three distinct robustification strategies like norm penalization, local robustification, and plan regularization. Figure 1 shows a good overview of the different kinds of strategies. While that is standard, further analysis presented by the paper is interesting.

The theoretical evidence is the strongest aspect of the paper. Claims regarding the efficacy of Tukey/Huber losses (TGW/HGW) and local metric truncation (LRGW) are rigorously supported by Proposition 1, Theorem 1, Lemma 3, Proposition 2, and Theorem 4.

The empirical experiments provide a mixed but generally supportive validation of the proposed robustification methods. On the strong end, the shape-matching experiments convincingly demonstrate both the baseline fragility of standard GW alignment and the corrective power of the new loss functions (Figures 2 and 3), while the noisy MNIST-USPS results solidly validate the premise of aligning cleaned proxy distributions rather than contaminated samples (Figure 6). Conversely, the evidence supporting local robustification in the image-translation domain is more limited. Although Figures 4 and 5 offer useful insights, these specific experiments rely on smaller datasets and FID score comparisons.

---

> ### Author Response · Authors · 2026-06-17
>
> We thank the Action Editor and all reviewers for their time and valuable contributions. We have carried out all suggested changes in the camera-ready version and de-anonymized the repository.
>
> Regards,
> Authors